SciPost Physics

Submission

# Graphical model for tensor factorization by sparse sampling

Angelo Giorgio Cavaliere [1] Riki Nagasawa [2] Shuta Yokoi [2]
Tomoyuki Obuchi [3] Hajime Yoshino [1,2*]

**1** D3 Center, Osaka University, Toyonaka, Osaka 560-0043, Japan
**2** Graduate School of Science, Osaka University, Toyonaka, Osaka 560-0043, Japan
**3** Department of Systems Science, Graduate School of Informatics, Kyoto University,
Yoshida Hon-machi, Sakyo-ku, Kyoto-shi, Kyoto 606-8501, Japan
* yoshino.hajime.cmc@osaka-u.ac.jp

October 22, 2025

## Abstract

We consider tensor factorizations based on sparse measurements of the tensor components. The measurements are designed in a way that the underlying graph of interactions is a random graph. The setup will be useful in cases where a substantial amount of data is missing, as in recommendation systems heavily used in social network services. In order to obtain theoretical insights on the setup, we consider statistical inference of the tensor factorization in a high dimensional limit, which we call as dense limit, where the graphs are large and dense but not fully connected. We build message-passing algorithms and test them in a Bayes optimal teacher-student setting. We also develop a replica theory, which becomes exact in the dense limit, to examine the performance of statistical inference.

# 1 Introduction

We consider a reconstruction problem for $N$ $M$-dimensional vectors $\boldsymbol{x}_i \in \mathbb{R}^M$ ($i = 1, 2, \ldots, N$) from observations of $p$-plets $\pi_{i_1, i_2, \ldots, i_p}$ of $\{\boldsymbol{x}_i\}$, which are defined as

$$\pi_{i_1, i_2, \ldots, i_p} = \frac{\lambda}{\sqrt{M}} \sum_{\mu=1}^{M} F_{i_1, i_2, \ldots, i_p, \mu} x_{i_1\mu} x_{i_2\mu} \cdots x_{i_p\mu}, \tag{1}$$

where $\lambda$ controls the signal strength and $F_{i_1, i_2, \ldots, i_p, \mu}$ is the linear coefficient of the observation, which is later assumed to be unity or an independently and identically distributed (i.i.d.) random variable from certain distributions, depending on situations. We will consider the cases of $p = 2, 3, \ldots$ in the present paper, while the special case $p = 1$ with $N = 1$ corresponds to linear estimation problems. To consider general observation processes, the actual observation $y_{i_1, i_2, \ldots, i_p}$ of $\pi_{i_1, i_2, \ldots, i_p}$ is assumed to be generated from the following output distribution:

$$P_{\text{out}}(y_{i_1, i_2, \ldots, i_p} \mid \pi_{i_1, i_2, \ldots, i_p}). \tag{2}$$

This defines the likelihood function for $\boldsymbol{x}$ through the relation (1).

    Similar problem settings have been investigated in the context of tensor/matrix factorization so far [1–10], and the difference of our work from them lies in assuming the limit $N, M \to \infty$ while keeping the relation $N \gg M$, and in assuming the observations are made uniformly randomly over all $p$-plets under the constraint that each vector $\boldsymbol{x}_i$ is observed

$c = \alpha M$ times with $\alpha = O(1)$. Thus the inference is based on very sparse measurements of the tensor: just $O(NM)$ out of $N^{p-1}$ elements of the tensor are measured. We call these assumptions *dense limit* [11,12] hereafter. This is an interesting limit in the context of tensor completion: once we obtain an estimate of the vectors $\boldsymbol{x}_i$ $i = 1, 2, \ldots, N$, we can make an inference of the whole tensor with $O(N^{p-1})$ elements, just based on a vanishing fraction of it.

The aim of this paper is to clarify the performance of the theoretically optimal estimation and also to provide an algorithm achieving optimality under the dense limit. To this end, we work in the framework of the so-called Bayes optimal inference, assuming that the likelihood $P_{\text{out}}$, the linear coefficient $F$, and the prior distribution $P_{\text{pri.}}$ of $\boldsymbol{x}$ are known. The posterior distribution in the Bayes optimal setting is known to give the minimum mean square error (MMSE), and thus serves as a good benchmark for other frameworks and algorithms.

To analyze the Bayes optimal performance, we use the replica method from statistical mechanics. The result reveals how the MMSE depends on the parameters, as well as the existence of phase transitions connected to the computational difficulty of the posterior or the fundamental estimation limit. In addition, we invent an efficient algorithm computing the posterior average based on the framework of generalized approximate message passing (G-AMP) [13]; the associated state evolution (SE) equations are derived and checked to be consistent with the replica result. Numerical experiments on finite-size systems based on G-AMP are also conducted, showing the consistency with the SE/replica predictions.

The consistency of the numerical experiments with the theoretical predictions suggest that our findings are exact in the limit we are currently considering, and, to the best of our knowledge, this is the first result of such precise asymptotics for tensor/matrix factorization where the rank is not $O(1)$ and based on significantly sparse measurements of the tensor. This becomes possible by assuming the dense limit which allows intricate correlations between variables to be ignored. This is an important technical consequence of this paper.

The remainder of this paper is organized as follows. In the next section, we provide a detailed formulation of our problem setting. Sec. 3 explains the analysis using the replica method: we here elaborate how higher-order correlations among variables can be neglected under the dense limit. In sec. 4, the algorithm based on G-AMP and the associated SE equations are derived. In sec. 5, we examine some specific parameter settings and observe the behavior of the MSE, phase transitions, and the performance of the G-AMP algorithm. Sec. 6 concludes the paper with some remarks.

## 2 Formulation and Related Work

### 2.1 Notations, Graphical Models, and Quantities of Interest

Let us consider $N$ $M$-dimensional vectors $\boldsymbol{x}_i = (x_{i1}, x_{i2}, \ldots, x_{iM})^\top \in \mathbb{R}^M$ $(i = 1, 2, \ldots, N)$ and denote the variable index set (identified with the variable set itself) by $\mathtt{V} = \{1, 2, \ldots, N\}$. As mentioned in sec. 1, we make observations over a set of $p$-plets which is uniformly and randomly sampled over all $p$-plets without replacement under the constraint that each vector $\boldsymbol{x}_i$ is observed $c = \alpha M$ $(\alpha = O(1))$ times. The set of actually selected $p$-plets is hereafter denoted as $\mathtt{E}$ and called edge set, with the intention of later associating this with a graphical model. For convenience, we introduce the symbol $\blacksquare$ to index the selected $p$-plets. Among the $p$-plets, the subset that includes site $i(\in \mathtt{V})$ is denoted by $\partial i = \{\blacksquare \in \mathtt{E} \mid i \in \blacksquare\}$: the equality $|\partial i| = c$ holds by the above constraint. In parallel, we express the variable

set belonging to the $p$-plet $\blacksquare$ as $\partial\blacksquare = \{i \in \blacksquare\}$: the equality $|\partial\blacksquare| = p$ holds by definition. The noiseless observation associated to a $p$-plet $\blacksquare$, $\pi_\blacksquare$, is thus rewritten from Eq. (1) to

$$\pi_\blacksquare = \frac{\lambda}{\sqrt{M}} \sum_{\mu=1}^{M} F_{\blacksquare,\mu} \prod_{i \in \partial\blacksquare} x_{i\mu}. \tag{3}$$

Its observation $y_\blacksquare$ is generated from the output distribution $P_{\text{out}}(y_\blacksquare \mid \pi_\blacksquare)$. Under these notations, the posterior distribution given all the observations $\{y_\blacksquare\}_{\blacksquare\in\mathsf{E}}$ is expressed as

$$P_{\text{pos.}}\left(\{\boldsymbol{x}_i\}_{i\in\mathsf{V}} \mid \{y_\blacksquare\}_{\blacksquare\in\mathsf{E}}\right) = \frac{1}{Z(\{y_\blacksquare\}_{\blacksquare\in\mathsf{E}})}\left(\prod_{\blacksquare\in\mathsf{E}} P_{\text{out}}(y_\blacksquare \mid \pi_\blacksquare)\right)\left(\prod_{i\in\mathsf{V}} \prod_{\mu=1}^{M} P_{\text{pri.}}(x_{i\mu})\right), \tag{4}$$

where the prior distribution $P_{\text{pri.}}$ is assumed to be separable over each component and $Z\left(\{y_\blacksquare\}_{\blacksquare\in\mathsf{E}}\right)$ is a normalization constant expressed as

$$Z\left(\{y_\blacksquare\}_{\blacksquare\in\mathsf{E}}\right) = \int \left(\prod_{i\in\mathsf{V}} \prod_{\mu=1}^{M} dx_{i\mu}\right)\left(\prod_{\blacksquare\in\mathsf{E}} P_{\text{out}}(y_\blacksquare \mid \pi_\blacksquare)\right)\left(\prod_{i\in\mathsf{V}} \prod_{\mu=1}^{M} P_{\text{pri.}}(x_{i\mu})\right). \tag{5}$$

In addition, the edge set size is denoted as $N_\blacksquare = |\mathsf{E}|$, and it is related to the other parameters as

$$N_\blacksquare = Nc/p = (\alpha/p)NM \tag{6}$$

Note that this edge set size is much smaller than the number of all $p$-plets $\binom{N}{p} = O(N^p)$. The assumption of this sparse observation with the limit $M, N \to \infty$ keeping $N \gg M$ is the key to making our analysis exact, which has been thus termed *dense limit* as explained in sec. 1. The quantity

$$\gamma = \frac{\alpha}{p} \tag{7}$$

represents the ratio of the number of observations to the number of variables to be estimated, making it an important parameter that characterizes the estimation accuracy.

The above problem setup can be illustrated by a factor graph as shown in Fig. 1: the variable $\boldsymbol{x}_i$ corresponds to a variable node represented by a gray circle, and the likelihood $P_{\text{out}}(y_\blacksquare \mid \pi_\blacksquare)$ is associated to a function node expressed by a black square. The variable node can be decomposed into a collection of sub-variable nodes corresponding to the variable components $x_{i\mu}$, which are represented by green circles. From this graphical representation, the algorithm known as belief propagation (BP) is naturally derived, which will be explained in sec. 4.

### 2.1.1  Technical Assumptions

For each tensor component $\pi_\blacksquare$ to be $O(1)$, the following two properties are hereafter assumed.

1. The prior distribution is zero-mean:

$$\int \mathrm{d}x P_{\text{pri.}}(x)x = 0. \tag{8}$$

2. The linear coefficient $F_{\blacksquare,\mu}$ satisfy one of the following two conditions.

   (a) (deterministic model)

   $$F_{\blacksquare,\mu} = 1. \tag{9}$$

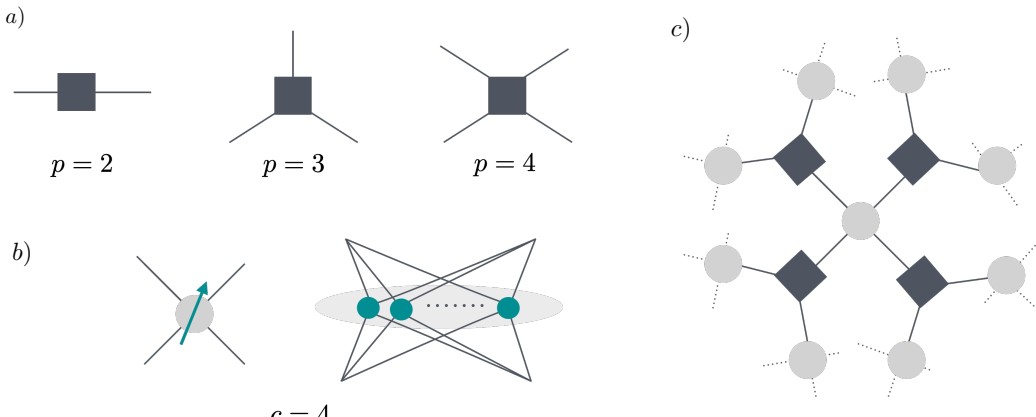

Figure 1: Graphical representations of our model. a) The black-squares ■ represent the function nodes to each of which a $p$-plet $(i_1, i_2, \ldots, i_p)$ is assigned. Each function node has $p$ arms. b) The gray circles represent the variable nodes $(i = 1, 2, \ldots, N)$ to each of which a $M$-component vectorial variable $\boldsymbol{x}_i$ (arrow) is assigned. Each of the variable nodes has $c = \alpha M$ arms. Equivalently, we may also represent a variable node as a collection of sub-variable nodes (green circles) to each of which a component of the vector $x_{i\mu}$ is assigned. c) a factor graph is created by joining the variable and function nodes.

(b) (random model) or (random spreading factor model)

$F_{\blacksquare,\mu}$ is an i.i.d. random variable with zero mean and unit variance. Writing the distribution function of $F$ as $P(F)$,

$$\int dF_{\blacksquare,\mu} P(F_{\blacksquare,\mu}) F_{\blacksquare,\mu} = 0, \qquad \int dF_{\blacksquare,\mu} P(F_{\blacksquare,\mu}) F_{\blacksquare,\mu}^2 = 1. \tag{10}$$

For the deterministic model we can write $P(F_{\blacksquare,\mu}) = \delta(F_{\blacksquare,\mu} - 1)$.

These two properties are crucial for making our analysis exact, which will become clear particularly in sec. 3 for the replica analysis. Furthermore, we remark that the two conditions for the linear coefficient, deterministic and random, yield the identical macroscopic result in the dense limit. From the viewpoint of the tensor/matrix decomposition, the deterministic case $F = 1$ corresponds to the so-called CP (Canonical Polyadic) decomposition and is more natural. These facts might seem to suggest that there is no reason to analyze the random model. We, however, argue that this is not the case. First, the fact that the deterministic and random cases yield the identical result in the limit is itself a nontrivial finding that has been revealed through our analysis. Second, in the message passing algorithms we develop, the random coefficient significantly improves convergence, particularly in the case of $p = 2$; this as a result enhances the consistency between the microscopic behavior of the algorithm and the macroscopic behavior predicted by SE or the replica method, even for relatively small values of $N$ and $M$. This means that the random case also has its own significance.

Although any prior distributions satisfying Eq. (8) can be treated in the presented formulation, we mainly consider the two cases: Ising and Gaussian distributions. Their explicit formulae are

$$P_{\text{pri.}}(x) = \frac{1}{2} \left[ \delta(x - 1) + \delta(x + 1) \right] \tag{11}$$

for the Ising case and

$$P_{\text{pri.}}(x) = \frac{e^{-\frac{1}{2}x^2}}{\sqrt{2\pi}} \tag{12}$$

for the Gaussian case.

Similarly, although our theoretical treatment again allows us to treat generic output distributions as understood from the derivations shown in secs. 3,4, we consider two types of specific distributions for quantitative analysis. The first type treats an additive noise $w$. Namely

$$y = \pi + w. \tag{13}$$

This means the likelihood function can be written as

$$P_{\text{out}}\big(y|\pi\big) \quad = \int dw W(w)\delta(y - (\pi + w)) = W(y - \pi), \tag{14}$$

with $W(w)$ being the noise distribution. A particularly interesting case is the Gaussian noise:

$$W(w) = \mathcal{N}(w \mid 0, \Delta^2). \tag{15}$$

The other type is the sign output function:

$$y = \text{sgn}(\pi). \tag{16}$$

This yields

$$P_{\text{out}}(y|\pi) = \delta(y - 1)\theta(\pi) + \delta(y + 1)\theta(-\pi), \tag{17}$$

where $\theta(x)$ is the Heaviside step function.

### 2.1.2 Averages and Order Parameters

In the present problem, there are several different types of random variables. To discriminate the average over these things, we introduce two different average notations.

The first average is that of $x_{i\mu}$s over the posterior distribution (4) and is denoted as

$$\langle(\cdots)\rangle = \int \left( \prod_{i \in \mathtt{V}} \prod_{\mu=1}^{M} dx_{i\mu} \right) P_{\text{pos.}}\big(\{\boldsymbol{x}_i\}_{i \in \mathtt{V}} \mid \{y_\blacksquare\}_{\blacksquare \in \mathtt{E}}\big)(\cdots). \tag{18}$$

This corresponds to the thermal average in statistical mechanics.

The posterior average depends on the observations $\{y_\blacksquare\}_{\blacksquare \in \mathtt{E}}$ which are random variables themselves, and we denote the average over the $y_\blacksquare$s by

$$\mathbb{E}_y(\cdots) = \int \left( \prod_{\blacksquare \in \mathtt{E}} dy_\blacksquare \right) P(\{y_\blacksquare\}_{\blacksquare \in \mathtt{E}})(\cdots), \tag{19}$$

where $P(\{y_\blacksquare\}_{\blacksquare \in \mathtt{E}})$ is the distribution of $\{y_\blacksquare\}_{\blacksquare \in \mathtt{E}}$, which is computed from the likelihood and the prior distribution as follows:

$$P(\{y_\blacksquare\}_{\blacksquare \in \mathtt{E}}) = \int \left( \prod_{\blacksquare \in \mathtt{E}} \prod_{\mu=1}^{M} dF_{\blacksquare,\mu} P_F(F_{\blacksquare,\mu}) \right) \left( \prod_{i \in \mathtt{V}} \prod_{\mu=1}^{M} dx_{i\mu} P_{\text{pri.}}(x_{i\mu}) \right) \left( \prod_{\blacksquare \in \mathtt{E}} P_{\text{out}}(y_\blacksquare \mid \pi_\blacksquare) \right). \tag{20}$$

This average corresponds to the quenched average in statistical mechanics of disordered systems. Although the graph structure associated to $\mathtt{E}$ is also a random variable in the

current setup, this randomness has no direct consequence on the analysis and we refrain from introducing a symbol to represent it.

Our estimation target is $\{\boldsymbol{x}_i\}_{i \in \mathsf{V}}$. To express this explicitly, we denote the true vectors generated from the prior distribution $P_{\mathrm{pri.}}$ as $\{\boldsymbol{x}_{*,i}\}_{i \in \mathsf{V}}$. Here, the symbol $*$ is interpreted to represent parameters to be estimated. The $p$-plets $\{\pi_{\blacksquare}\}$ are also considered to be the estimation targets and the same subscript rule is applied. Namely, we express the true value of the $\pi_{\blacksquare}$ as

$$\pi_{*,\blacksquare} = \frac{\lambda}{\sqrt{M}} \sum_{\mu=1}^{M} F_{\blacksquare,\mu} \prod_{i \in \partial\blacksquare} x_{*,i\mu}. \tag{21}$$

The linear coefficient $F_{\blacksquare,\mu}$ is assumed to be known and thus has no $*$-subscript. Hereafter, we call the system with these true values *teacher*, while the inference model is termed *student*. This terminology follows the so-called teacher-student setup in generic inference problems [14], which originally comes from neural networks.

As it will be evident in secs. 3 and 4, the following order parameters play the key roles in characterizing the macroscopic behavior:

$$m \quad = \mathbb{E}_y \left[ \tfrac{1}{NM} \sum_{i \in \mathsf{V}} \sum_{\mu=1}^{M} x_{*,i\mu} \langle x_{i\mu} \rangle \right], \tag{22}$$

$$q \quad = \mathbb{E}_y \left[ \tfrac{1}{NM} \sum_{i \in \mathsf{V}} \sum_{\mu=1}^{M} \langle x_{i\mu} \rangle^2 \right]. \tag{23}$$

The parameter $m$ is the overlap which measures the similarity between teacher and student. We may call it as 'magnetization' due to the analogy to spin systems in physics. When the meaningful inference is possible, the inequality $m > 0$ holds. Using the physics terminology, we may call the phase with $m = 0$ as paramagnetic phase (phase where inference is impossible) and $m > 0$ as ferromagnetic or magnetized phase. On the other hand, the parameter $q$ is analogous to the Edwards-Anderson order parameter used in spin-glass physics [15].

These order parameters are directly connected to two standard quantitative indicators of inference accuracy. One is the MSE concerning to the input $\boldsymbol{x}_i$:

$$\frac{1}{NM} \sum_{i \in \mathsf{V}} \sum_{\mu=1}^{M} \mathbb{E}_y \left[ \left( \langle x_{i\mu} \rangle - x_{*,i\mu} \right)^2 \right] = 1 - 2m + q, \tag{24}$$

where we assumed $\mathbb{E}_y x_{*,i\mu}^2 = 1$, which is the case according to our priors (see Eq. (11) and Eq. (12)). The other is the MSE about the output $\pi_{i_1,i_2,\ldots,i_p}$:

$$\binom{N}{p}^{-1} \sum_{1 \le i_1 < i_2 < \cdots i_p \le N} \mathbb{E}_y \left[ \left( \langle \pi_{i_1,i_2,\ldots,i_p} \rangle - \pi_{*,i_1,i_2,\ldots,i_p} \right)^2 \right] = \lambda^2 (1 - 2m^p + q^p).$$

Here we assumed that the posterior distribution is separable and $\frac{1}{N} \sum_{i \in \mathsf{V}} \langle x_{i\mu} \rangle = 0$, $\frac{1}{N} \sum_{i \in \mathsf{V}} \langle x_{i\mu} \rangle \langle x_{i\nu} \rangle = 0$ ($\mu \ne \nu$), both of which asymptotically hold in the current setup, and $\mathbb{E}_y \left( \pi_{*,i_1,i_2,\ldots,i_p}^2 \right) = \lambda^2$ which is the case in our priors. The latter MSE can also be regarded as a generalization error in the context of tensor completion.

In the Bayes-optimal case, one can prove easily that $m = q$ holds [14], which is analogous to what happens along the Nishimori line in spin-glass systems [16,17]. This idealized situation is the problem setup of this study.

## 2.2 Symmetries

There are several global symmetries in our inference problems: the statistical weight of a microscopic state $\{x_{i\mu}\}$ is invariant under the following operations depending on situations.

- **reflection**: In the cases of even $p$, the system is invariant under $x_{i\mu} \to -x_{i\mu}$ for all $i \in V$. This is because we are considering prior distributions $P_{\mathrm{pri.}}(x)$ which are even functions of $x$ (see Eq. (11) and Eq. (12)) .

- **permutation**: In the case $F_{\blacksquare,\mu} = 1$ (deterministic model), the system is invariant under permutations of the index $\mu$ for the components. In the random model (Eq. (10)), this symmetry is suppressed.

- **rotation**: In the special case of $p = 2$ with the Gaussian prior (Eq. (12)), the system with $F_{\blacksquare,\mu} = 1$ (deterministic model, Eq. (9)) is invariant under global rotations $x_{i\mu} = \sum_{\nu=1}^{M} R_{\mu\nu} x_{i\nu}$ with any rotation matrix $R_{\mu\nu}$. In the random model (Eq. (10)), this symmetry is suppressed.

Note that even in the random model (Eq. (10)) the reflection symmetry still remains for even $p$.

The order parameters defined in Eq. (22) and Eq. (23) become 0 under the above symmetry transformations. By the replica theory developed in sec. 3 and applied to specific models in sec 5, we investigate possibilities of spontaneous breakings of the symmetries in the dense limit $\lim_{c \to \infty} \lim_{N \to \infty}$. By the message passing algorithm developed in sec. 4 and applied to specific models in sec 5, the remaining symmetry may be broken dynamically by the algorithm even in systems with finite $N$,$c$ and $M$.

## 2.3 Related Work

Let us sketch below how our problem can be compared with other problems.

### 2.3.1 Matrix/Tensor Factorization

Matrix factorization ($p = 2$) has been extensively studied across various disciplines due to their broad applicability. For instance, dictionary learning seeks to represent signals or data samples as sparse linear combinations of learned basis elements, and it has proven effective in image denoising and compression tasks [18, 19]. Another prominent line of research focuses on low-rank matrix completion, where the goal is to recover a low-rank structure from incomplete observations; a landmark result in this area showed that under certain conditions exact recovery can be achieved through convex optimization [20]. In applications such as array signal processing and compressed sensing, blind calibration has emerged to jointly estimate unknown sensor (or system) parameters and the underlying low-rank or sparse representation of data [21]. Finally, robust PCA addresses the decomposition of data matrices into a low-rank component plus sparse outliers, ensuring stable principal component recovery even in the presence of gross corruptions [22]. The situation is similar for tensor factorization: many applications are found in fields such as signal processing and machine learning [23–25]. These approaches collectively highlight the power of exploiting low-dimensional structures, whether in matrices or higher-order tensors, to solve a range of real-world problems.

It is important to note that the matrix/tensor factorization is performed often in the presence of missing entries [26] which can be substantial in practice. This aligns with our current problem setup with sparse observation. When solving such problems, the estimated matrix/tensor's rank should be effectively low enough compared to its apparent

dimensionality. The magnitude of such effective rank in real-world data varies between different cases, and it is not always very low. For example, estimated tensor-rank can be very large (order $10^2$) for facial images [27] and recommendation systems (50–1000) [28,29], while $N$ is even larger by orders of magnitude. The assumption $N \gg M$ in the dense limit that we consider in this paper is well aligned with this situation, suggesting that the analysis in this paper can provide meaningful insights in such a setting.

### 2.3.2 Statistical Mechanical Approach to Tensor/Matrix Factorization

Low-rank ($M = O(1)$) tensor/matrix factorization have been studied in a number of earlier works from statistical mechanical viewpoints. For example, [1] proposed an iterative estimation algorithm for rank-one matrices under various constraints (e.g., sparsity or non-negativity) and analyzed its performance using SE. [2] combined low-rank matrix estimation with clustering and invented an AMP algorithm, showing that the AMP algorithm can outperform classical algorithms such as Lloyd's K-means in certain regimes. [3] focused on tensor PCA in the rank-one spiked tensor model and conducted a thorough analysis of both statistical and computational thresholds. In particular, this work derived an information-theoretic threshold (below which no estimator can reliably recover the spike) and compared it to the thresholds required by polynomial-time algorithms proposed also in the paper (naive power iteration, unfolding, and an AMP algorithm). The paper also discussed how the maximum likelihood estimation achieves the statistically optimal threshold, whereas practical, polynomial-time methods require a larger signal-to-noise ratio to succeed. This highlights the presence of easy and hard regions for spike recovery. [4] also studied the spiked tensor model with rank $M = O(1)$ in a Bayes-optimal framework, and computed the mutual information, MMSE, and other statistical measures via the replica method. A key finding is that there is an information-theoretic threshold on the signal-to-noise ratio below which accurate recovery is impossible, regardless of computational resources. The paper also proposes an AMP algorithm and examines its performance, showing that while AMP can achieve the MMSE in certain parameter regimes, there remains a hard region where AMP (and presumably any polynomial-time algorithm) cannot match the Bayes-optimal performance.

On the other hand, the high-rank (extensive-rank) case ($M = O(N)$) has been analyzed in some papers but only in the context of matrix factorization or dictionary learning. [5,6] analyzed the reconstruction performance of the dictionaries (matrices) based on the frameworks of the reconstruction error minimization with the $\ell_0$ regularization and the Bayes-optimal one, respectively, using the replica method. The results show that the reconstruction is possible with $O(N)$ training samples both for the error minimization and the Bayes-optimal frameworks, underscoring that feasible sample complexities scale linearly with the dictionary dimension. In the Bayesian framework, both [7,8] used the replica method to provide asymptotically exact descriptions of the MMSE and free-energy, and invented AMP algorithms. [7] mainly treated the problems of dictionary learning and blind calibration while [8] generalized a range of matrix factorization problems (dictionary learning, low-rank matrix completion, blind calibration, robust PCA) into a single Bayesian inference framework. Both established phase diagrams characterizing whether the matrix reconstruction is tractable or not, and clarified how they depend on the problem setting details.

Actually, the high-rank case poses significant challenges. It has become recognized that the above analyses are not, in fact, precise after a while. To overcome the situation, [30] proposed a systematic high-temperature or Plefka–Georges–Yedidia (PGY) expansion to derive the improved (but still approximate) mean-field description, showing that third-order corrections in the expansion lead to better predictions of the reconstruction error,

highlighting the importance of including normally neglected terms. Progress has been made in sub-linearly high-rank cases $M = O(N^a)$ with $0 < a < 1$ with full measurements $O(N^2)$ [10, 31, 32]. Moreover in [33], Barbier and Macris studied dictionary learning by combining random matrix theory with the replica method, introducing a new ansatz called spectral replica method. They derived exact or near-exact characterizations of the mutual information and mean-square errors for the matrix factorization task. In [34], Barbier et al. considered symmetric matrix denoising of a product-form matrix $XX^\top$ under additive Wigner noise, but with a non-rotationally-invariant prior on $X$. Their extensive numerical simulations show a distinct first-order denoising-factorization transition. Below this critical noise level, the rotationally invariant estimator (RIE) remains nearly optimal regardless of the prior. Above the threshold, however, the discrete and factorized structure of the prior strictly outperforms the RIE, though the authors observe that achieving this factorization-based improvement is often algorithmically difficult (due to a glassy posterior landscape).

As shown above, accurately analyzing the high-rank situation is challenging. In the present paper we show it becomes relatively tractable under the dense limit ($N \gg M \gg 1$ with $O(NM)$ measurements) assumed in this paper for $p \geq 2$. By leveraging the well established replica method and AMP within the dense limit, precise asymptotics are obtained. Interestingly, in contrast to [10] where the sub-linearly high-rank case $M = O(N^\alpha)$ with $0 < \alpha < 1$ is treated under full measurements, our problems do not become equivalent to (a bunch of) the rank-one $M = 1$ cases.

### 2.3.3  Vectorial Constraint Satisfaction Problems

Vectorial constraint satisfaction problems (CSP) was studied in [11], and the problem studied in this paper is exactly an inverse problem or *planted version* of it.

Let us briefly summarize the vectorial CSP problem. One considers the $M(\gg 1)$-dimensional vectorial 'spin' variables $\boldsymbol{x}_i$ ($i = 1, 2, \ldots, N$) interacting with each other through $p$-body interactions specified by the following Hamiltonian:

$$H = \sum_{\blacksquare=1}^{N_\blacksquare} V(\delta - \pi_\blacksquare), \qquad \pi_\blacksquare = \frac{1}{\sqrt{M}} \sum_{\mu=1}^{M} F_{\blacksquare,\mu} \prod_{j \in \partial \blacksquare} x_{j\mu}, \tag{25}$$

with a generic potential $V(x)$. As the 'spin' variables, Ising and spherical spins were considered. For the factor $F_{\blacksquare,\mu}$ (denoted as $X_{\blacksquare\mu}$ in [11]), the deterministic, fully disordered, and intermediately disordered cases were considered. Glass transitions (clustering transitions) and jamming (SAT/UNSTAT transition) were studied in detail. Here the number of components $M$ is assumed to be large $M \gg 1$. This approach is complementary to the CSPs of scalar continuous variables [35, 36].

In the case of linear potential

$$V(x) = Jx, \tag{26}$$

the problem was found to become essentially the same as the conventional (scalar $M = 1$) $p$-spin (Ising/spherical) spin glass models with global $c \propto N^{p-1}$ coupling. In the case of $p = 2$, the disordered $F_{\blacksquare,\mu}$ was needed to avoid crystallization. Otherwise, disorder-free glass transitions were found within super-cooled paramagnetic phases.

More interesting cases are of non-linear potentials. In the quadratic potential

$$V(x) = \frac{\epsilon}{2} x^2, \tag{27}$$

continuous replica symmetry breaking (RSB) was found even for the $p = 2$ spherical model. In the case of the hard-core potential

$$e^{-\beta V(x)} = \theta(x), \tag{28}$$

various types of RSB and jamming transitions were found with spherical spins. The universality class of the jamming turned out to be the same as that of hard-spheres [37] for any $p \geq 1$, including $p = 1$ which corresponds to simple perceptrons [38].

### 2.3.4 Error-correcting codes

Our problem is also related to error correcting codes, which is a traditional topic in statistical mechanical informatics. In the Sourlas code [39] (which assumes global coupling $c \propto N^{p-1}$) and in the Kabashima-Saad code [40] (sparse coupling $c = O(1)$), $p$-body products of scalar ($M = 1$) binary (Ising) variables are used to build the codes $\pi_{i_1, i_2, \dots, i_p}$. Both codes attains the Shannon's information theoretical bound [41] in the limit $p \to \infty$. An important difference is that the transmission rate $R = NM/N_{\blacksquare} = Mp/c$ in the thermodynamic limit $N \to \infty$ vanishes in the Sourlas code ($M = 1$) because of the global coupling $c \propto N^{p-1}$, while it can remain finite in the Kabashima-Saad code ($M = 1$) by taking the limit $c \to \infty$, after $N \to \infty$ limit, with the ratio $p/c$ fixed. On the other hand, we consider vectorial variables with large number of components $M = c/\alpha \gg 1$ and intermediately dense coupling $N \gg M \gg 1$. Our model also attains the Shannon's bound with *finite* transmission rate $R = p/\alpha$ by taking the limit $p \to \infty$ with the ratio $p/\alpha$ fixed.

### 2.3.5 Other Related Issues

Linear observation ($p = 1$) problems were widely studied in a number of contexts such as code division multiple access (CDMA) [42, 43], perceptron [44, 45], and compressed sensing [9, 46, 47]. Our model can be regarded as a non-linear ($p > 1$) extension of these problems for a large number of vectors ($N \gg 1$). In the linear estimations ($p = 1$), a 'random spreading code' $F$ corresponding to Eq. (10) is indispensable.

The dense limit is used in this paper to make it possible to neglect higher-order loop effects. The essentially same technique was recently employed in the analysis of deep neural networks (DNNs) [12]. The dense limit serves as a useful asymptotic regime that enables analytical calculations. Beyond DNNs and tensor factorization, there may be other problems where this approach proves beneficial.

## 3 Replica Theory

### 3.1 Basic formalism

The logarithm of the normalization constant $Z$ (see Eq. (5)), or the free energy in statistical mechanics, contains all the information about the system and serves as the central quantity in the discussion here. This quantity is expected to show the self-averaging property and thus converges to its expectation value in our dense limit. To analytically deal with this expectation, the replica method utilizes the following identity:

$$\mathbb{E}_y \ln Z = \lim_{n \to 0} \frac{\partial}{\partial n} \ln \mathbb{E}_y Z^n. \tag{29}$$

The free energy $f$ is thus defined and computed as

$$-f = \lim_{M,N \to \infty} \frac{1}{MN} \mathbb{E}_y \ln Z \left( \{y_{\blacksquare}\} \right) = \lim_{M,N \to \infty} \lim_{n \to 0} \frac{1}{MN} \frac{\partial}{\partial n} \ln \mathbb{E}_y Z^n \left( \{y_{\blacksquare}\} \right). \tag{30}$$

with the partition function $Z$ defined in Eq. (5). In practice, we first set $n$ as a positive integer in order to perform calculations, obtain an analytically continuable expression

under suitable ansatz, and then take the limit $n \to 0$ using this expression. For any positive integer $n$, we may write the replicated partition function $Z^n$ as follows:

$$Z^n(\{y_\blacksquare\}) = \int \prod_{a=1}^{n} \left\{ \left( \prod_{i \in \mathsf{V}} \prod_{\mu=1}^{M} dx_{i\mu}^a \right) \left( \prod_{i \in \mathsf{V}} \prod_{\mu=1}^{M} P_{\mathrm{pri.}}^a(x_{i\mu}^a) \right) \left( \prod_{\blacksquare \in \mathsf{E}} P_{\mathrm{out}}\left(y_\blacksquare | \pi_\blacksquare^a\right) \right) \right\}, \quad (31)$$

where $\pi_\blacksquare^a$ is defined as (see Eq. (3))

$$\pi_\blacksquare^a = \frac{1}{\sqrt{M}} \sum_{\mu=1}^{M} F_{\blacksquare,\mu} \prod_{j \in \partial\blacksquare} x_{j\mu}^a. \quad (32)$$

Recalling how the teacher generates the measurement $y_\blacksquare$ (see Eq. (19)), we find

$$\mathbb{E}_y(\cdots) = \int \left( \prod_{\blacksquare \in \mathsf{E}} dy_\blacksquare \prod_{\mu=1}^{M} dF_{\blacksquare,\mu} P_F(F_{\blacksquare,\mu}) \right)$$
$$\times \int \left( \prod_{i \in \mathsf{V}} \prod_{\mu=1}^{M} dx_{*,i\mu} \right) \left( \prod_{i \in \mathsf{V}} \prod_{\mu=1}^{M} P_{\mathrm{pri.}}(x_{*,i\mu}) \right) \left( \prod_{\blacksquare \in \mathsf{E}} P_{\mathrm{out}}(y_\blacksquare | \pi_{*,\blacksquare}) \right) (\cdots). \quad (33)$$

Assigning the replica index 0 to the teacher or the ground truth as $x_{*,i\mu} = x_{i\mu}^0$, $\pi_* = \pi^0$, we have

$$\mathbb{E}_y Z^n(\{y_\blacksquare\}) = \int \left( \prod_{\blacksquare \in \mathsf{E}} dy_\blacksquare \prod_{\mu=1}^{M} dF_{\blacksquare,\mu} P_F(F_{\blacksquare,\mu}) \right)$$
$$\times \int \prod_{a=0}^{n} \left\{ \left( \prod_{i \in \mathsf{V}} \prod_{\mu=1}^{M} dx_{i\mu}^a \right) \left( \prod_{i \in \mathsf{V}} \prod_{\mu=1}^{M} P_{\mathrm{pri.}}^a(x_{i\mu}^a) \right) \left( \prod_{\blacksquare \in \mathsf{E}} P_{\mathrm{out}}(y_\blacksquare | \pi_\blacksquare^a) \right) \right\}$$
$$= Z_{1+n}[\partial/\partial h] \prod_{\blacksquare \in \mathsf{E}} \left( \int dy_\blacksquare \prod_{a=0}^{n} P_{\mathrm{out}}(y_\blacksquare | h_\blacksquare^a) \right) \Bigg|_{h=0}, \quad (34)$$

where we introduced

$$Z_{1+n}[\partial/\partial h] = \int \left( \prod_{\blacksquare \in \mathsf{E}} \prod_{\mu=1}^{M} dF_{\blacksquare,\mu} P_F(F_{\blacksquare,\mu}) \right)$$
$$\times \int \prod_{a=0}^{n} \left\{ \left( \prod_{i \in \mathsf{V}} \prod_{\mu=1}^{M} dx_{i\mu}^a \right) \left( \prod_{i \in \mathsf{V}} \prod_{\mu=1}^{M} P_{\mathrm{pri.}}^a(x_{i\mu}^a) \right) \right\} e^{\sum_{\blacksquare \in \mathsf{E}} \sum_{a=0}^{n} \pi_\blacksquare^a \frac{\partial}{\partial h_\blacksquare^a}}. \quad (35)$$

The last equation of Eq. (34) can be understood from the following relation:

$$P_{\mathrm{out}}(y | \pi) = e^{\pi \frac{\partial}{\partial h}} P_{\mathrm{out}}(y | h) \big|_{h=0}. \quad (36)$$

In the limit $M \to \infty$ we wish to analyze the order parameters (Eq. (22) and Eq. (23)) which can detect the spontaneous breaking of the global symmetries discussed in sec. 2.2.

To this end we introduce overlaps between replicas including the teacher $a = 0$ and students $a = 1, 2, \ldots, n$:

$$Q_i^{ab} = \frac{1}{M} \sum_{\mu=1}^{M} x_{i\mu}^a x_{i\mu}^b. \tag{37}$$

In order to introduce the order parameters in our analysis we use an identity

$$
\begin{aligned}
1 &= \int \prod_{i \in \mathtt{V}} \prod_{a < b} \left( M \mathrm{d} Q_i^{ab} \delta \Big( M Q_i^{ab} - \sum_{\mu=1}^{M} x_{i\mu}^a x_{i\mu}^b \Big) \right) \\
&= \int \prod_{i \in \mathtt{V}} \prod_{a < b} \left( M \mathrm{d} Q_i^{ab} \frac{d\epsilon_i^{ab}}{2\pi i} \exp \left[ \epsilon_i^{ab} \left( \sum_{\mu=1}^{M} x_{i\mu}^a x_{i\mu}^b - M Q_i^{ab} \right) \right] \right)
\end{aligned} \tag{38}
$$

in the following computation. It is also convenient to introduce the diagonal elements of the overlap matrix $Q_i^{aa}$ ($a = 0, 1, 2, \ldots, n$, $i = 1, 2, \ldots, N$) which reflect normalization of the variables $x_{i\mu}^a$. In the case of the Ising prior (Eq. (11)), $Q_i^{aa} = 1$ holds for any $M$. However, in the case of the Gaussian prior (Eq. (12)), $Q_i^{aa} = 1$ becomes satisfied strictly only in the $M \to \infty$ limit due to the law of large numbers. Thus we introduce $Q_i^{aa}$ as additional order parameter and consider, in addition to Eq. (38),

$$1 = \int \frac{d\epsilon_i^{aa}}{4\pi i} \exp \left[ \frac{\epsilon_i^{aa}}{2} \left( \sum_{\mu=1}^{M} x_{i\mu}^a x_{i\mu}^a - M Q_i^{aa} \right) \right]. \tag{39}$$

The integral will be evaluated by the saddle point method in the $M \to \infty$ limit. We will find that $\epsilon_i^{aa}$ remains arbitrary in the case of the Ising prior.

Using the identities Eq. (38) and Eq. (39) we can rewrite Eq. (35) as,

$$Z_{1+n}[\partial/\partial h] = \int \prod_{i \in \mathtt{V}} \prod_{a < b} (M \mathrm{d} Q_i^{ab})(M \prod_a d Q_i^{aa}) e^{-\hat{F}[Q][\partial/\partial h]}, \tag{40}$$

where

$$e^{-\hat{F}[Q][\partial/\partial h]} = \int_{-i\infty}^{i\infty} \prod_{i \in \mathtt{V}} \prod_{a < b} \frac{d\epsilon_i^{ab}}{2\pi i} \prod_a \frac{d\epsilon_i^{aa}}{4\pi i} e^{-\frac{M}{2} \sum_{i \in \mathtt{V}} \sum_{a,b} \epsilon_i^{ab} Q_i^{ab}} e^{-\hat{G}[\epsilon][\partial/\partial h]}. \tag{41}$$

Here we introduced

$$-\hat{G}[\epsilon][\partial/\partial h] = -G_0[\epsilon] + \ln \left\langle e^{\sum_{\blacksquare \in \mathtt{E}} \sum_{a=0}^{n} \pi_\blacksquare^a \frac{\partial}{\partial h_\blacksquare^a}} \right\rangle_\epsilon, \tag{42}$$

with

$$-G_0[\epsilon] = -\sum_{i \in \mathtt{V}} \sum_{\mu=1}^{M} g_0[\epsilon_i], \quad -g_0[\epsilon_i] = \ln \int \prod_{a=0}^{n} dx^a P_{\mathrm{pri.}}^a(x^a) e^{\frac{1}{2} \sum_{a,b} \epsilon_i^{ab} x^a x^b}, \tag{43}$$

and

$$
\langle (\cdots) \rangle_\epsilon = Z_\epsilon^{-1} \int \left( \prod_{\blacksquare \in \mathrm{E}} \prod_{\mu=1}^{M} dF_{\blacksquare,\mu} P_F(F_{\blacksquare,\mu}) \right)
$$

$$
\times \left( \prod_{i \in \mathrm{V}} \prod_{\mu=1}^{M} \left( \prod_{a=0}^{n} dx_{i\mu}^a \right) P_{\mathrm{pri.}}(x_{i\mu}^a) e^{\frac{1}{2} \sum_{a,b} \epsilon_i^{ab} x_{i\mu}^a x_{i\mu}^b} \right) (\cdots), \tag{44}
$$

$$
Z_\epsilon = \int \left( \prod_{\blacksquare \in \mathrm{E}} \prod_{\mu=1}^{M} dF_{\blacksquare,\mu} P_F(F_{\blacksquare,\mu}) \right) \left( \prod_{i \in \mathrm{V}} \prod_{\mu=1}^{M} \left( \prod_{a=0}^{n} dx_{i\mu}^a \right) P_{\mathrm{pri.}}(x_{i\mu}^a) e^{\frac{1}{2} \sum_{a,b} \epsilon_i^{ab} x_{i\mu}^a x_{i\mu}^b} \right). \tag{45}
$$

Here the subscript 0 is meant to emphasize that the average is took with a 'non-interacting' system.

In the above expressions, $\hat{F}[Q]$ and $\hat{G}[\epsilon]$ are functionals of the order parameters $Q_i^{ab}$ and the conjugated fields $\epsilon_i^{ab}(= \epsilon_i^{ba})$. We will perform the integral over $\epsilon_i^{ab}$ in Eq. (41) by the saddle point method for $M \gg 1$. Then we find,

$$
-\hat{F}[Q] = -\sum_{i \in \mathrm{V}} \frac{M}{2} \sum_{a,b} (\epsilon_i^{ab})^*[Q] \, Q_i^{ab} - \hat{G}[\epsilon], \qquad Q_i^{ab} = \left. \frac{\partial(-\hat{G}[\epsilon])}{\partial \epsilon_i^{ab}} \right|_{\epsilon = \epsilon^*[Q]}, \tag{46}
$$

which can be regarded as a Legendre transformation.

We will evaluate the 2nd term of Eq. (42) perturbatively in a way analogous to the Plefka expansion [48–50]. Introducing a parameter $A$ as a book keeping device to organize an expansion, we will obtain

$$
\hat{G} = G_0 + A\hat{G}_1 + \frac{A^2}{2} \hat{G}_2 + \cdots. \tag{47}
$$

Later we will find that $A = \lambda^2$ is the proper choice as terms proportional to odd powers of $\lambda$ vanish for a symmetry reason. Then the functional $F[Q]$ and the corresponding $\epsilon^*$ can be obtained through Eq. (46) as

$$
\hat{F} = F_0 + A\hat{F}_1 + \frac{A^2}{2} \hat{F}_2 + \cdots, \qquad \epsilon = \epsilon_0 + A\epsilon_1 + \frac{A^2}{2}\epsilon_2 + \cdots, \tag{48}
$$

which yields

$$
\begin{aligned}
-F_0[Q] &= -G_0[\epsilon_0] - \frac{M}{2} \sum_i \sum_{a,b} (\epsilon_0)_i^{ab} Q_i^{ab}, \\
-\hat{F}_1[Q] &= -\hat{G}_1[\epsilon_0], \\
-\hat{F}_2[Q] &= -\hat{G}_2[\epsilon_0] + \frac{(G_0'[\epsilon_0])^2}{G_0''[\epsilon_0]} \qquad \cdots,
\end{aligned} \tag{49}
$$

where $\epsilon_0$ is determined by

$$
Q_i^{ab} = \left. \frac{\partial(-g_0[\epsilon_i])}{\partial \epsilon_i^{ab}} \right|_{\epsilon = \epsilon_0}. \tag{50}
$$

Fortunately we will find that higher order terms $O(A^2)$ vanish in the dense limit $N \gg c \gg 1$ in our models.

Wrapping up the above formal results we can write using Eq. (34), Eq. (35), Eq. (40)

$$\mathbb{E}_y Z^n \left( \{y_\blacksquare\} \right) = \int \prod_{i \in \mathsf{V}} \prod_{a \leq b} M \mathrm{d} Q_i^{ab} e^{-F[Q]}, \tag{51}$$

with $-F[Q]$ defined as

$$-F[Q] = -F_0[Q] - F_{\mathrm{ex}}[Q], \tag{52}$$

which may be regarded as replicated free-energy functional. Here $-F_0[Q]$ can be regarded as entropic part of the free-energy while $-F_{\mathrm{ex}}[Q]$ is the interaction part of the free-energy defined as

$$-F_{\mathrm{ex}}[Q] = \ln \left\{ e^{-\hat{F}_{\mathrm{ex}}[Q][\partial/\partial h]} \prod_{\blacksquare \in \mathsf{E}} \int dy_\blacksquare \prod_{a=0}^n P_{\mathrm{out}}(y_\blacksquare | h_\blacksquare^a) \right\}, \tag{53}$$

with

$$-\hat{F}_{\mathrm{ex}}[Q][\partial/\partial h] = -F_1[Q][\partial/\partial h] - (A^2/2)\hat{F}_2[Q][\partial/\partial h] + \cdots. \tag{54}$$

In the present paper we will consider spatially homogeneous solution:

$$Q_i^{ab} = Q^{ab}, \qquad \epsilon_i^{ab} = \epsilon^{ab}. \tag{55}$$

This is justified since there are no inhomogeneities in the system (e.g. boundary), and thus the dependence on the variable index $i$ of these quantities is hereafter removed. The simplest ansatz for the matrix $Q$ is the replica symmetric one:

$$Q_i^{ab} = (Q - q)\delta_{ab} + q, \qquad a, b \in \{0, 1, 2, \ldots, n\}, \tag{56}$$

which is justified since we will focus on Bayes optimal settings in the present paper. Here $Q$ represents the normalization of the variables $x_{i\mu}^a$ for all replicas $a = 0, 1, 2 \ldots, n$. However we will consider a slightly more general ansatz:

$$\begin{aligned} Q^{ab} &= (Q - q)\delta_{ab} + q, & a, b \in \{1, 2, \ldots, n\}, \\ Q^{0a} &= Q^{a0} = m, & a \in \{1, 2, \ldots, n\}, \\ Q^{00} &= Q_0. \end{aligned} \tag{57}$$

Here $Q$ and $Q_0$ parameterize normalization of the variables $x_{i\mu}^a$ for the students $a = 1, 2 \ldots, n$ and the teacher $a = 0$ respectively.

The parameters $m$ and $q$ are precisely the order parameters introduced in Eq. (22) and Eq. (23). Note that Eq. (56) is recovered assuming $Q = Q_0$ and $m = q$ with $m$ being the overlaps between the teacher (0-th replica) and students and $q$ being the overlaps between the students. In Bayes optimal inferences $m = q$ generally holds much as on the Nishimori line in spinglass models [14, 16, 17].

Correspondingly, for the matrix $\epsilon^{ab}$ we assume the similar form as Eq. (57). We will denote the field conjugated to $m$ as $\psi$ in the Ising prior case:

$$\epsilon^{0a} = \epsilon^{a0} = \psi, \qquad a \in \{1, 2, \ldots, n\}. \tag{58}$$

## 3.2 Entropic part of the free-energy

Let us first compute the entropic part of the free-energy $F_0[Q]$. To this end we have to evaluate $g_0[\epsilon]$ (see Eq. (43)).

### 3.2.1 Ising prior

For the Ising model we have the prior distribution Eq. (11) which yields

$$
\begin{aligned}
-g_0\left[\epsilon\right] &= \ln \prod_{a=0}^{n} \sum_{x^a=\pm 1} e^{\frac{1}{2}\sum_{a,b=0}^{n} \epsilon^{ab} x^a x^b} \\
&= \ln \prod_{a=0}^{n} \sum_{x^a=\pm 1} e^{\frac{1}{2}\sum_{a,b=1}^{n} \epsilon^{ab} \frac{\partial^2}{\partial h^a \partial h^b} + \frac{1}{2}\epsilon^{00} + x^0 \sum_{a=1}^{n} \epsilon^{0a} \frac{\partial}{\partial h^a}} e^{\sum_{a=1}^{n} h^a x^a}\bigg|_{h=0} \\
&= \frac{\epsilon^{00}}{2} + \ln e^{\frac{1}{2}\sum_{a,b=1}^{n} \epsilon^{ab} \frac{\partial^2}{\partial h^a \partial h^b}} \prod_{a=1}^{n}\left(2\cosh\left(h^a + \epsilon^{0a}\right)\right)\bigg|_{h=0}. \quad (59)
\end{aligned}
$$

To derive the last equation we used lemma 1 and $\cosh(h) = \cosh(-h)$.

Then from Eq. (50) we find

$$
Q^{ab} = \frac{e^{\frac{1}{2}\sum_{a,b=1}^{n}(\epsilon_0^*)^{ab}\frac{\partial^2}{\partial h^a \partial h^b}} \prod_{c=1}^{n}(2\cosh(h^c + (\epsilon_0^*)^{0c})) \tanh(h^a + (\epsilon_0^*)^{0a}) \tanh(h^b + (\epsilon_0^*)^{0b})\big|_{h=0}}{e^{\frac{1}{2}\sum_{a,b=1}^{n}(\epsilon_0^*)^{ab}\frac{\partial^2}{\partial h^a \partial h^b}} \prod_{c=1}^{n}(2\cosh(h^c + (\epsilon_0^*)^{0c}))\big|_{h=0}}
$$

$$
(a \neq b, a \geq 1, b \geq 1),
$$

$$
Q^{0a} = Q^{a0} = \frac{e^{\frac{1}{2}\sum_{a,b=1}^{n}(\epsilon_0^*)^{ab}\frac{\partial^2}{\partial h^a \partial h^b}} \prod_{c=1}^{n}(2\cosh(h^c + (\epsilon_0^*)^{0c})) \tanh(h^a + (\epsilon_0^*)^{0a})\big|_{h=0}}{e^{\frac{1}{2}\sum_{a,b=1}^{n}(\epsilon_0^*)^{ab}\frac{\partial^2}{\partial h^a \partial h^b}} \prod_{c=1}^{n}(2\cosh(h^c + (\epsilon_0^*)^{0c}))\big|_{h=0}} \quad (a \geq 1),
$$

$$
Q^{aa} = 1 \quad (a = 0, 1, 2, \ldots, n), \quad (60)
$$

which determines $(\epsilon_0^*)^{ab}$ (see Eq. (50)). To derive the last equation we used $e^{\frac{d^2}{dh^2}}\cosh(h) = \cosh(h)$.

Note that $Q^{aa} = 1$ is satisfied with $\epsilon^{aa}$ remaining arbitrary. Then the 1st equation of Eq. (49) yields

$$
\frac{-F_0[Q]}{M} = -\frac{1}{2}\sum_i \sum_{a,b=0}^{n}(\epsilon_0^*)_i^{ab} Q_i^{ab} + \sum_i \ln e^{\frac{1}{2}\sum_{a,b=1}^{n}(\epsilon_0^*)_i^{ab}\frac{\partial^2}{\partial h_a \partial h_b}} \prod_{c=1}^{n} 2\cosh\left(h_c + (\epsilon_0^*)^{0a}\right)\bigg|_{h=0}. \quad (61)
$$

We now assume the replica symmetric ansatz Eq. (57). For the matrix $\epsilon^{ab}$ we assume the similar form with Eq. (58),

$$
\begin{aligned}
\frac{-F_0(q,m)}{NM} &= -\frac{1}{2}\left[(1+n)\epsilon_0^* + (n^2-n)\epsilon_0^* q + 2n\psi_0^* m\right] + \frac{\epsilon_0^*}{2} + \ln e^{\frac{\epsilon_0^*}{2}\frac{\partial^2}{\partial h^2}}\left(2\cosh(h + \psi_0^*)\right)^n\bigg|_{h=0} \\
&= -\frac{1}{2}\left[n\epsilon_0^* + (n^2-n)\epsilon_0^* q + 2n\psi_0^* m\right] + \ln \int \mathcal{D}z(2\cosh\left(\psi_0^* + \sqrt{\epsilon_0^*}z\right)^n. \quad (62)
\end{aligned}
$$

Note that we assumed the replica symmetric form $(\epsilon_0^*)^{ab} = \epsilon_0^* \ (a, b > 1)$ and for convenience the diagonal components were also set to be equal to the non-diagonal ones $((\epsilon_0^*)^{aa} = \epsilon_0^*)$ using the arbitrariness. $\epsilon_0^*$ and $\psi_0^*$ are to be determined by Eq. (50). Here we introduced a short-hand notation,

$$
\int \mathcal{D}z \ldots = \int_{-\infty}^{\infty} \frac{dz}{\sqrt{2\pi}} e^{-\frac{z^2}{2}} \ldots, \quad (63)
$$

and used lemma 2. We find in $n \to 0$ limit, within the ansatz Eq. (58),

$$q = \int \mathcal{D}z \, \tanh^2(\sqrt{\epsilon_0^*}z + \psi_0^*),$$

$$m = \int \mathcal{D}z \, \tanh\left(\sqrt{\epsilon_0^*}z + \psi_0^*\right). \tag{64}$$

### 3.2.2 Gaussian prior

For the Gaussian model we have the prior distribution Eq. (12) which yields

$$
\begin{aligned}
-g_0(\epsilon) &= \ln \int \prod_{a=0}^{n} dx^a \frac{e^{-\frac{1}{2}x_a^2}}{\sqrt{2\pi}} e^{\frac{1}{2}\sum_{a,b=0}^{n} \epsilon^{ab}x^a x^b} \\
&= \ln \int \prod_{a=0}^{n} \frac{dx^a}{\sqrt{2\pi}} e^{\frac{1}{2}\sum_{a,b=0}^{n}(\epsilon^{ab}-\delta_{ab})x^a x^b} = -\frac{1}{2}\ln \det(-(\hat{\epsilon}-I)). 
\end{aligned}
\tag{65}
$$

Then from Eq. (50) we find

$$Q^{ab} = (-(\hat{\epsilon}_0^* - I))^{-1}, \tag{66}$$

where $I$ is the identity matrix. Finally the 1st equation of Eq. (49) yields,

$$\frac{-F_0[Q]}{M} = -\frac{1}{2}\sum_i \sum_{a=0}^{n} Q_i^{aa} + \frac{nN}{2}\ln(2\pi) + \frac{1}{2}\sum_i \ln \det Q_i^{ab}. \tag{67}$$

In the homogeneous and replica symmetric ansatz Eq. (56), we find using lemma 6,

$$\frac{-F_0(Q,q)}{NM} = \frac{1}{2}\left[\ln(Q+nq) + n\ln(Q-q)\right] - \frac{1+n}{2}Q. \tag{68}$$

In the slightly generalized ansatz Eq. (57) we find using lemma 6 and lemma 7,

$$
\begin{aligned}
\frac{-F_0(Q_0,Q,q,m)}{NM} &= \frac{1}{2}\ln Q_0 + \frac{1}{2}\ln \det((q-m^2/Q_0) + (Q-q)\delta_{ab}) \\
&= \frac{1}{2}\ln Q_0 + \frac{1}{2}\left[\ln\left(Q-q+n(q-m^2/Q_0)\right) + (n-1)\ln(Q-q)\right] \\
&\quad -\frac{1}{2}Q_0 - \frac{n}{2}Q.
\end{aligned}
\tag{69}
$$

We will need derivatives of $\partial_n F_0(q)|_{n=0}$ with respect to $Q,q$ and $m$ which we find as,

$$
\begin{aligned}
\partial_n \frac{\partial}{\partial Q}\frac{-F_0(Q,q,m)}{NM}\Big|_{n=0} &= \frac{1}{2}\frac{Q-2q+m^2/Q_0}{(Q-q)^2} - \frac{1}{2}, \\
\partial_n \frac{\partial}{\partial q}\frac{-F_0(Q,q,m)}{NM}\Big|_{n=0} &= \frac{1}{2}\frac{q-m^2/Q_0}{(Q-q)^2}, \\
\partial_n \frac{\partial}{\partial m}\frac{-F_0(Q,q,m)}{NM}\Big|_{n=0} &= -\frac{m/Q_0}{Q-q}.
\end{aligned}
\tag{70}
$$

It can be noticed that in the Bayes optimal case for which $q = m$ holds, $\partial_n \frac{\partial}{\partial Q}\frac{-F_0(Q,q,m)}{NM}\big|_{n=0} = 0$ provided also that $Q = Q_0 = 1$ holds.

## 3.3   Interaction part of the free-energy

Now we turn to computations of the interaction part of the free-energy $-F_{\text{ex}}[Q]$. To this end we start from the cumulant expansion of the 2nd term of Eq. (42), which allows us to obtain the Plefka expansion of the functional $\hat{G}[\epsilon_0]$ (see Eq. (47)),

$$A(-G_1)[\epsilon_0] + \frac{A^2}{2}(-G_2)[\epsilon_0]\ldots \quad = \quad \ln\langle e^{-\lambda\sum_{\blacksquare}\sum_a \pi_{\blacksquare}^a \frac{\partial}{\partial h_{\blacksquare}^a}}\rangle_\epsilon. \tag{71}$$

Now let us evaluate the last expression by a cumulant expansion,

$$\ln\langle e^{-\lambda\sum_{\blacksquare}\sum_a \pi_{\blacksquare}^a \frac{\partial}{\partial h_{\blacksquare}^a}}\rangle_\epsilon = -\lambda\sum_{\blacksquare}\sum_a \langle \pi_{\blacksquare}^a\rangle_\epsilon \frac{\partial}{\partial h_{\blacksquare}^a}$$

$$+ \quad \frac{\lambda^2}{2}\sum_{\blacksquare_1,\blacksquare_2}\sum_{a,b}\left[\langle \pi_{\blacksquare_1}^a \pi_{\blacksquare_2}^b\rangle_\epsilon - \langle \pi_{\blacksquare_1}^a\rangle_\epsilon\langle \pi_{\blacksquare_2}^b\rangle_\epsilon\right]\frac{\partial^2}{\partial h_{\blacksquare_1}^a \partial h_{\blacksquare_2}^b} + O(\lambda^3)\ldots \tag{72}$$

where we introduced a parameter $\lambda$ to organize a perturbation series. We find below that $A = \lambda^2$ is the proper choice. After extracting $G_1, G_2, \ldots$ through Eq. (71) we will consider the Legendre transformation Eq. (46) order by order in the Plefka expansion.

It is convenient to represent various terms generated by the cumulant expansion Eq. (72) by diagrams as those depicted in Fig. 2. The details of the analysis is presented in appendix B. The key observations in the analysis is the following.

- Disconnected diagrams do not appear in the cumulant expansion meaning that correlation functions which appear in $\hat{G}$ are *connected correlation functions*. On each variable node we should put a connected correlation function like $\langle x_{i\mu}^a x_{i\mu}^b\rangle_{\epsilon,0}^c = \langle x_{i\mu}^a x_{i\mu}^b\rangle_\epsilon - \langle x_{i\mu}^a\rangle_\epsilon\langle x_{i\mu}^b\rangle_\epsilon = Q^{ab}$, $\langle x_{i\mu}^a x_{i\mu}^b x_{i\mu}^c x_{i\mu}^d\rangle_{\epsilon,0}^c$, ... Note also that odd cumulants like $\langle x_{i\mu}^a\rangle_{\epsilon,0}^c$, $\langle x_{i\mu}^a x_{i\mu}^b x_{i\mu}^c\rangle_{\epsilon,0}^c$,... are zero due to the reflection symmetry mentioned in sec. 2.2: $P_{\text{pri.}}^a(x) = P_{\text{pri.}}^a(-x)$ which holds both in the Ising and Gaussian priors. This is the reason why $A = \lambda^2$ becomes the proper choice.

- One particle reducible diagrams, namely diagrams which become disconnected into two parts by cutting connections between two adjacent factor nodes (as those shown in Fig. 2 b)) disappear by the Legendre transform $\hat{G} \to \hat{F}$ Eq. (46) so that only *one particle irreducible (1PI) diagrams* contribute to $\hat{F}$ [51, 52]. They can be classified into two types.

  (A) Diagrams which consist of one factor node as represented by those shown in Fig. 2 a) replicated into $2, 4, 6, 8...$ replicas. Among these, we will find that only 2 replica case is relevant and all others vanish in the dense limit $N \gg c \gg 1$.

  (B) Diagrams which consist of closed loops such as those depicted in Fig. 2 c). We will find that they vanish in the dense limit $N \gg c \gg 1$. However it is not necessarily the case in the case of global coupling $c \propto N^{p-1}$.

Remarkably, as we noted above, we find that the contributions from higher order cumulants $O(A^2), O(A^4)...$ vanish in the dense limit $N \gg c \gg 1$. We explain the details in the appendix sec B where we find an indication that $N$ must grow faster than any polynomial of $c$. As the result we find (see Eq. (263)),

$$\hat{F}_{\text{ex}}[Q][\partial/\partial h_{\blacksquare}^a] = -F_1[Q][\partial/\partial h_{\blacksquare}^a] = \frac{1}{2}\sum_{\blacksquare}\sum_{a,b=0}^n \prod_{j\in\partial\blacksquare} Q_j^{ab}\frac{\partial^2}{\partial h_{\blacksquare}^a \partial h_{\blacksquare}^b}. \tag{73}$$

Using this in Eq. (53) we obtain

$$-F_{\text{ex}}[Q] = \ln \left\{ e^{\frac{1}{2} \sum_{\blacksquare} \sum_{a,b=0}^{n} \prod_{j \in \partial \blacksquare} Q_j^{ab} \frac{\partial^2}{\partial h_{\blacksquare}^a \partial h_{\blacksquare}^b}} \prod_{\blacksquare \in E} \int dy_{\blacksquare} \prod_{a=0}^{n} P_{\text{out}}(y_{\blacksquare}|h_{\blacksquare}^a) \right\}. \quad (74)$$

The vanishment of the higher order contributions is reminiscent of what one encounters in the standard procedure to prove the central limit theorem. Indeed, the above result can be reproduced assuming a Gaussian ansatz as commonly employed in the literature [5,8], in which one explicitly assumes that fluctuation of $\pi$ within the canonical ensemble follows a Gaussian distribution with zero mean and the variance parametrized by $Q_i^{ab}$. Similar situation happens in the structural glass transition of supercooled liquids in the large dimensional limit [53], as well as in the inverse problem of the current problem [11]. However, let us emphasize that the vanishment of the higher order contributions does *not* hold generally, including in the case $c \propto N^{p-1}$. For instance, the matrix factorization problem ($p = 2$) suffers this issue when one considers full rank case $M = O(N)$. This point has been noticed also in the analysis of the TAP equation for the matrix factorization problem [30] and the problem of machine learning by multi-layer perceptrons [12].

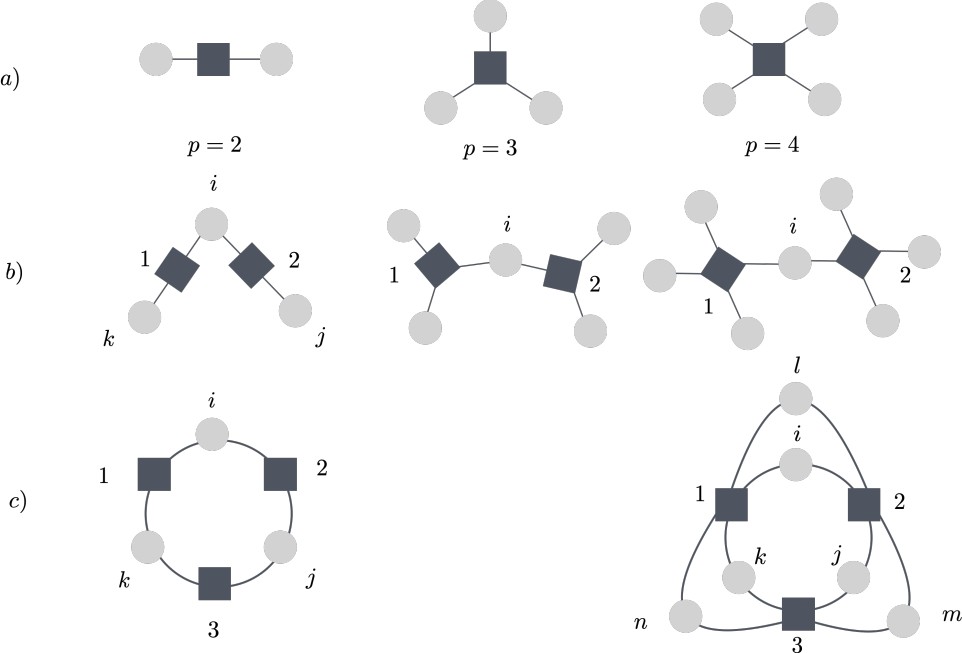

Figure 2: Some representative diagrams which appear in the cumulant expansion.

### 3.3.1 Additive noise

In the case of the additive noise Eq. (14) we have

$$P_{\text{out}}(y|\pi) = W(y - \pi). \quad (75)$$

Using this in Eq. (53) we find

$$-F_{\text{ex}}[Q] = \sum_{\blacksquare} \ln \int dy_{\blacksquare} \exp\left[\frac{\lambda^2}{2}\sum_{a,b=0}^{n}\prod_{j\in\partial\blacksquare}Q_j^{ab}\frac{\partial^2}{\partial h_{\blacksquare}^a \partial h_{\blacksquare}^b}\right]\prod_{a=0}^{n}W(y_{\blacksquare}+h_{\blacksquare}^a)\big|_{\{h_{\blacksquare}^a=0\}} \tag{76}$$

$$= \sum_{\blacksquare}\ln\int dy_{\blacksquare}W(y_{\blacksquare})\exp\left[\frac{\lambda^2}{2}\sum_{a,b=1}^{n}\left(\prod_{j\in\partial\blacksquare}Q_j^{00}-\prod_{j\in\partial\blacksquare}Q_j^{0b}-\prod_{j\in\partial\blacksquare}Q_j^{a0}+\prod_{j\in\partial\blacksquare}Q_j^{ab}\right)\frac{\partial^2}{\partial h_{\blacksquare}^a\partial h_{\blacksquare}^b}\right]$$

$$\prod_{a=1}^{n}W(y_{\blacksquare}+h_{\blacksquare}^a)\big|_{\{h_{\blacksquare}^a=0\}}. \tag{77}$$

To derive the 2nd equation we performed integrations by parts.

In the homogeneous and replica symmetric ansatz Eq. (56) we find from the 1st equation of Eq. (77),

$$\frac{-F_{\text{ex}}(Q,q)}{NM} = \frac{\alpha}{p}\ln\int dy\, e^{\frac{\lambda^2}{2}q^p\frac{\partial^2}{\partial h^2}}\left(e^{\frac{\lambda^2}{2}(Q-q^p)\frac{\partial^2}{\partial h^2}}W(y+h)\right)^{1+n}\Bigg|_{h=0}, \tag{78}$$

where we used $N_{\blacksquare} = NM(\alpha/p)$ (see Eq. (6) and Eq. (7)). In the slightly generalized ansatz Eq. (57) we find from the 2nd equation of Eq. (77),

$$\frac{-F_{\text{ex}}(Q_0,Q,q,m)}{NM} = \frac{\alpha}{p}\ln\int dy\,W(y)e^{\frac{\lambda^2}{2}(Q_0^p-2m^p+q^p)\frac{\partial^2}{\partial h^2}}\left(e^{\frac{\lambda^2}{2}(Q^q-q^p)\frac{\partial^2}{\partial h^2}}W(y+h)\right)^n\Bigg|_{h=0}$$

$$= \frac{\alpha}{p}\ln\int dy\,W(y)\int\mathcal{D}z_0\left(\int\mathcal{D}z_1W(y+\sqrt{Q_0^p-2m^p+q^p}z_0+\sqrt{Q^p-q^p}z_1)\right)^n\Bigg|_{h=0}, \tag{79}$$

where we used the short-hand notation Eq. (63) and lemma 2.

We will need derivatives of $\partial_n F_{\text{ex}}(Q_0,Q,q,m)|_{n=0}$ with respect to $Q$, $q$ and $m$ which we find as

$$\partial_n\frac{\partial}{\partial Q}\frac{-F_{\text{ex}}(Q_0,Q,q,m)}{NM}\Big|_{n=0} = \frac{\lambda^2}{2}\alpha Q^{p-1}\theta_1(\lambda,Q_0,Q,m,q),$$

$$\partial_n\frac{\partial}{\partial q}\frac{-F_{\text{ex}}(Q_0,Q,q,m)}{NM}\Big|_{n=0} = -\frac{\lambda^2}{2}\alpha q^{p-1}\theta_0(\lambda,Q_0,Q,m,q),$$

$$\partial_n\frac{\partial}{\partial m}\frac{-F_{\text{ex}}(Q_0,Q,q,m)}{NM}\Big|_{n=0} = \lambda^2\alpha m^{p-1}\left(\theta_0(\lambda,m,q)-\theta_1(\lambda,Q_0,Q,m,q)\right), \tag{80}$$

with

$$\theta_0(\lambda,Q_0,Q,m,q) = \int dy\,W(y)e^{\frac{\lambda^2}{2}(Q_0^p-2m^p+q^p)\frac{\partial^2}{\partial h^2}}\left(\frac{e^{\frac{\lambda^2}{2}(Q^p-q^p)\frac{\partial^2}{\partial h^2}}\frac{d}{dh}W(y+h)}{e^{\frac{\lambda^2}{2}(Q^p-q^p)\frac{\partial^2}{\partial h^2}}W(y+h)}\right)^2\Bigg|_{h=0}$$

$$= \int dy\,W(y)\int\mathcal{D}z_0\left(\frac{\int\mathcal{D}z_1(d/d\Xi)W(\Xi)}{\int\mathcal{D}z_1W(\Xi)}\right)^2,$$

$$\theta_1(\lambda,Q_0,Q,m,q) = \int dy\,W(y)e^{\frac{\lambda^2}{2}(Q_0^p-2m^p+q^p)\frac{\partial^2}{\partial h^2}}\frac{e^{\frac{\lambda^2}{2}(Q^p-q^p)\frac{\partial^2}{\partial h^2}}\frac{d^2}{dh^2}W(y+h)}{e^{\frac{\lambda^2}{2}(Q^p-q^p)\frac{\partial^2}{\partial h^2}}W(y+h)}\Bigg|_{h=0}$$

$$= \int dy\,W(y)\int\mathcal{D}z_0\frac{\int\mathcal{D}z_1(d^2/d\Xi^2)W(\Xi)}{\int\mathcal{D}z_1W(\Xi)}, \tag{81}$$

and

$$\Xi = y - \lambda\sqrt{Q_0^p - 2m^p + q^p}z_0 - \lambda\sqrt{Q^p - q^p}z_1. \tag{82}$$

In the Bayes optimal case, for which $m = q$ holds, one finds $\theta_1 = 0$ as shown in Lemma 4 provided also that $Q_0 = Q$. The latter certainly holds for the Ising prior.

More specifically in the case of Gaussian noise

$$W(w) = \frac{e^{-\frac{w^2}{2}}}{\sqrt{2\pi}}, \tag{83}$$

we find

$$\theta_0(\lambda, Q_0, Q, m, q) = \frac{\lambda^2(Q_0^p - 2m^p + q^p) + 1}{(\lambda^2(Q^p - q^p) + 1)^2},$$

$$\theta_1(\lambda, Q_0, Q, m, q) = \theta_0(\lambda, Q_0, Q, m, q) - \frac{1}{\lambda^2(Q^p - q^p) + 1}. \tag{84}$$

In the Bayes-optimal case, for which $m = q$ must hold, we find again that $\theta_1 = 0$ holds provided $Q_0 = Q$ also holds.

### 3.3.2 Sign output

In the case of the sign output we have

$$P_{\text{out}}(y|\pi) = \theta(y\pi) = \delta(y - 1)\theta(\pi) + \delta(y + 1)\theta(-\pi). \tag{85}$$

Thus

$$\prod_a P_{\text{out}}(y|\pi^a) = \prod_a \theta(y\pi^a) = \delta(y - 1)\prod_a \theta(\pi^a) + \delta(y + 1)\prod_a \theta(-\pi^a), \tag{86}$$

since $\theta(x)\theta(-x) = 0$.

From the above result we find

$$
\begin{aligned}
-F_{\text{ex}}[Q] &= \sum_\blacksquare \ln \int dy_\blacksquare \exp\left[\frac{\lambda^2}{2}\sum_{a,b=0}^n \prod_{j\in\partial\blacksquare} Q_j^{ab}\frac{\partial^2}{\partial h_\blacksquare^a \partial h_\blacksquare^b}\right] \\
&\qquad \prod_{a=0}^n \left[\delta(y-1)\prod_a \theta(h^a) + \delta(y+1)\prod_a \theta(-h^a)\right]\Bigg|_{\{h_\blacksquare^a=0\}} \\
&= \sum_\blacksquare \ln\left[\exp\left[\frac{\lambda^2}{2}\sum_{a,b=0}^n \prod_{j\in\partial\blacksquare} Q_j^{ab}\frac{\partial^2}{\partial h_\blacksquare^a \partial h_\blacksquare^b}\right]\prod_{a=0}^n \theta(h_\blacksquare^a)|_{\{h_\blacksquare^a=0\}}\right] \\
&\quad + N_\blacksquare \ln 2. 
\end{aligned} \tag{87}
$$

In the following we will omit the last constant term.

In the homogeneous and replica symmetric ansatz Eq. (56) we find,

$$\frac{-F_{\text{ex}}(Q, q)}{NM} = \frac{\alpha}{p}\ln e^{\frac{\lambda^2}{2}q^p\frac{\partial^2}{\partial h^2}}\left(e^{\frac{\lambda^2}{2}(Q^p-q^p)\frac{\partial^2}{\partial h^2}}\theta(h)\right)^{1+n}\Bigg|_{h=0}. \tag{88}$$

In the slightly generalized ansatz Eq. (57) we can write

$$Q_{ab}^p = m^p + (Q_0^p - m^p)\delta_{a,0}\delta_{b,0} + I(a \geq 1)I(b \geq 1)[(q^p - m^p) + (Q^p - q^p)\delta_{a,b}], \tag{89}$$

where $I(..)$ is the indicator function. Then we find,

$$
\frac{-F_{\text{ex}}(Q_0, Q, q, m)}{NM} = \frac{\alpha}{p} \ln e^{\frac{\lambda^2}{2} m^p \frac{\partial^2}{\partial h^2}} \left\{ \left( e^{\frac{\lambda^2}{2}(Q_0^p - m^p)\frac{\partial^2}{\partial h^2}} \theta(h) \right) e^{\frac{\lambda^2}{2}(m^p - q^p)\frac{\partial^2}{\partial h^2}} \left( e^{\frac{\lambda^2}{2}(Q^p - q^p)\frac{\partial^2}{\partial h^2}} \theta(h) \right)^n \right\} \Bigg|_{h=0}
$$

$$
= \frac{\alpha}{p} \ln \int \mathcal{D}z_0 \left\{ \int \mathcal{D}z_3 \theta(\lambda\sqrt{m^p} z_0 + \lambda\sqrt{Q_0^p - m^p} z_3) \right.
$$

$$
\left. \int \mathcal{D}z_1 \left( \int \mathcal{D}z_2 \theta(\lambda\sqrt{m^p} z_0 + \lambda\sqrt{q^p - m^p} z_1 + \lambda\sqrt{Q^p - q^p} z_2) \right)^n \right\}
$$

$$
= \frac{\alpha}{p} \ln \int \mathcal{D}z_0 H(X_{\text{teacher}}) \int \mathcal{D}z_1 H^n(X_{\text{student}}), \tag{90}
$$

where we used the short-hand notation Eq. (63) and lemma 2. We also introduced a function $H(x)$,

$$
H(x) = \frac{1}{2}\text{erfc}\left(\frac{x}{\sqrt{2}}\right) = \int_x^\infty \frac{dt}{\sqrt{2\pi}} e^{-\frac{t^2}{2}}, \tag{91}
$$

and also

$$
X_{\text{teacher}} = -\frac{\sqrt{m^p}}{\sqrt{Q_0^p - m^p}} z_0, \qquad X_{\text{student}} = -\frac{\sqrt{m^p} z_0 + \sqrt{q^p - m^p} z_1}{\sqrt{Q^p - q^p}}. \tag{92}
$$

We will need derivatives of $\partial_n F_{\text{ex}}(Q_0, Q, q, m)|_{n=0}$ with respect to $Q$, $q$ and $m$ which we find as,

$$
\partial_n \frac{\partial}{\partial Q} \frac{-F_{\text{ex}}(Q_0, Q, q, m)}{NM}\Big|_{n=0} = \lambda^2 \alpha Q^{p-1} e^{\frac{\lambda^2}{2} m^p \frac{\partial^2}{\partial h^2}} \left\{ \left( e^{\frac{\lambda^2}{2}(Q_0^p - m^p)\frac{\partial^2}{\partial h^2}} \theta(h) \right) \right.
$$

$$
\left. e^{\frac{\lambda^2}{2}(m^p - q^p)\frac{\partial^2}{\partial h^2}} \left( \frac{\frac{\partial^2}{\partial h^2} e^{\frac{\lambda^2}{2}(Q^p - q^p)\frac{\partial^2}{\partial h^2}} \theta(h)}{e^{\frac{\lambda^2}{2}(Q^p - q^p)\frac{\partial^2}{\partial h^2}} \theta(h)} \right) \right\} \Bigg|_{h=0}
$$

$$
= \frac{\lambda^2 \alpha Q^{p-1}}{Q^p - q^p} \int \mathcal{D}z_0 H(X_{\text{teacher}}) \int \mathcal{D}z_1 \frac{H''(X_{\text{student}})}{H(X_{\text{student}})}, \tag{93}
$$

$$
\partial_n \frac{\partial}{\partial q} \frac{-F_{\text{ex}}(Q_0, Q, q, m)}{NM}\Big|_{n=0} = -\lambda^2 \alpha q^{p-1} e^{\frac{\lambda^2}{2} m^p \frac{\partial^2}{\partial h^2}} \left\{ \left( e^{\frac{\lambda^2}{2}(Q_0^p - m^p)\frac{\partial^2}{\partial h^2}} \theta(h) \right) \right.
$$

$$
\left. e^{\frac{\lambda^2}{2}(m^p - q^p)\frac{\partial^2}{\partial h^2}} \left( \frac{\frac{\partial}{\partial h} e^{\frac{\lambda^2}{2}(Q^p - q^p)\frac{\partial^2}{\partial h^2}} \theta(h)}{e^{\frac{\lambda^2}{2}(Q^p - q^p)\frac{\partial^2}{\partial h^2}} \theta(h)} \right)^2 \right\} \Bigg|_{h=0}
$$

$$
= -\frac{\lambda^2 \alpha q^{p-1}}{Q^p - q^p} \int \mathcal{D}z_0 H(X_{\text{teacher}}) \int \mathcal{D}z_1 \left( \frac{H'(X_{\text{student}})}{H(X_{\text{student}})} \right)^2, \tag{94}
$$

$$\partial_n \frac{\partial}{\partial m} \frac{-F_{\text{ex}}(Q_0, Q, q, m)}{NM}\Big|_{n=0} = \lambda^2 \alpha m^{p-1} e^{\frac{\lambda^2}{2} m^p \frac{\partial^2}{\partial h^2}} \left\{ \left( \frac{\partial}{\partial h} e^{\frac{\lambda^2}{2}(Q_0^p - m^p)\frac{\partial^2}{\partial h^2}} \theta(h) \right) \right.$$

$$\left. e^{\frac{\lambda^2}{2}(m^p - q^p)\frac{\partial^2}{\partial h^2}} \frac{\frac{\partial}{\partial h} e^{\frac{\lambda^2}{2}(Q^p - q^p)\frac{\partial^2}{\partial h^2}} \theta(h)}{e^{\frac{\lambda^2}{2}(Q^p - q^p)\frac{\partial^2}{\partial h^2}} \theta(h)} \right\}\Bigg|_{h=0}$$

$$= \frac{\lambda^2 \alpha m^{p-1}}{Q_0^p - m^p} \int \mathcal{D}z_0 H'(X_{\text{teacher}}) \int \mathcal{D}z_1 \frac{H'(X_{\text{student}})}{H(X_{\text{student}})}. \quad (95)$$

Here we used $e^{\frac{\lambda^2}{2} m^p \frac{\partial^2}{\partial h^2}} e^{\frac{\lambda^2}{2}(Q_0^p - m^p)\frac{\partial^2}{\partial h^2}} \theta(h) = e^{\frac{\lambda^2}{2}\frac{\partial^2}{\partial h^2}} \theta(h) = \int_0^\infty \frac{dy}{\sqrt{2\pi\lambda^2}} e^{-\frac{y^2}{2\lambda^2}} = 1/2$.

### 3.4 Total free-energy and equation of states

#### 3.4.1 In the absence of students

In the absence of students $n \to 0$ we find Eq. (74) becomes

$$\lim_{n \to 0} -F_{\text{ex}}[Q] = 0 \quad (96)$$

because of the normalization of the output function $\int dy P_{\text{out}}(y|h) = 1$. Then in the case of Gaussian prior we find the total free-energy, using Eq. (69), in the $n \to 0$ limit as

$$\lim_{n \to 0} \frac{-F(Q_0, Q, q, m)}{NM} = \frac{1}{2} \ln Q_0 - \frac{1}{2} Q_0. \quad (97)$$

Taking derivative with respect to $Q_0$ we find $0 = \lim_{n \to 0} \frac{\partial}{\partial Q_0} \frac{-F_0(Q_0, Q, q, m)}{NM}$ is satisfied with $Q_0 = 1$, as expected because of the law of large numbers. Thus we assume $Q_0 = 1$ in the following.

#### 3.4.2 Ising prior and additive noise

Using Eq. (62) and Eq. (79) together with $Q_0 = Q = 1$ which holds in the Ising case, we obtain the total replicated free-energy as

$$\frac{-F(q, m)}{NM} = -\frac{1}{2} \left[ n\epsilon_0^* + (n^2 - n)\epsilon_0^* q + 2n\psi_0^* m \right] + \ln \int \mathcal{D}z (2\cosh(\psi_0^* + \sqrt{\epsilon_0^*} z))^n$$

$$+ \frac{\alpha}{p} \ln \int dy W(y) \int \mathcal{D}z_0 \left( \int \mathcal{D}z_1 W(y + \lambda\sqrt{1 - 2m^p + q^p} z_0 + \lambda\sqrt{1 - q^p} z_1) \right)^n \quad (98)$$

from which we find the free-energy Eq. (30) as

$$-f = -\frac{1}{2} \left[ \epsilon_0^* - \epsilon_0^* q + 2\psi_0^* m \right] + \int \mathcal{D}z \ln\left( 2\cosh(\psi_0^* + \sqrt{\epsilon_0^*} z) \right)$$

$$+ \frac{\alpha}{p} \int dy W(y) \int \mathcal{D}z_0 \ln\left( \int \mathcal{D}z_1 W(y + \lambda\sqrt{1 - 2m^p + q^p} z_0 + \lambda\sqrt{1 - q^p} z_1) \right) \quad (99)$$

Extremizing the free-energy with respect to $q$ and $m$ we find, using Eq. (80)

$$\epsilon_0^* = \lambda^2 \alpha q^{p-1} \theta_0(\lambda, m, q)$$
$$\psi_0^* = \lambda^2 \alpha m^{p-1} (\theta_0(\lambda, m, q) - \theta_1(\lambda, m, q)) \quad (100)$$

Then the equations of state (saddle point equations) for the order parameters $q$ and $m$ are found using Eq. (64) as,

$$
\begin{aligned}
q &= \int \mathcal{D}z \tanh^2(\sqrt{\lambda^2 \alpha q^{p-1}\theta_0(\lambda,m,q)}z + \lambda^2 \alpha m^{p-1}(\theta_0(\lambda,m,q) - \theta_1(\lambda,m,q))), \\
m &= \int \mathcal{D}z \tanh\left(\sqrt{\lambda^2 \alpha q^{p-1}\theta_0(\lambda,m,q)}z + \lambda^2 \alpha m^{p-1}(\theta_0(\lambda,m,q) - \theta_1(\lambda,m,q))\right) \quad (101)
\end{aligned}
$$

where $\theta_0$ and $\theta_1$ are given by Eq. (81) with Eq. (82).

In the Bayes optimal case $m = q$, we have already shown using Lemma 4 that $\theta_1 = 0$. Then with $A \equiv \alpha \lambda^2 m^{p-1}\theta_0$ we find the two equations Eq. (101) become the same,

$$
m - q = \int \mathcal{D}z \tanh\left(A + \sqrt{A}z\right) - \int \mathcal{D}z \tanh^2(A + \sqrt{A}z) =
$$

$$
= \frac{e^{-\frac{A}{2}}}{\sqrt{2\pi A}} \int \mathrm{d}u\, e^{-\frac{u^2}{2A}} \tanh u\, \frac{1}{\cosh u} = 0, \qquad \forall A \qquad (102)
$$

where we have used a change of variable $u = A + \sqrt{A}z$ to recognize in the integrand an odd function of $u$.

Specifically we will analyze the Bayes-optimal case $m = q$ with Gaussian noise Eq. (83). In this case one finds, using Eq. (84),

$$
m = \int \mathcal{D}z \tanh\left(A + \sqrt{A}z\right), \qquad A = \frac{\alpha \lambda^2 m^{p-1}}{1 + \lambda^2(1 - m^p)}, \qquad (103)
$$

For this case, the free-energy Eq. (99) -Eq. (101) becomes,

$$
f(m,\alpha,\lambda) = \frac{1}{2}A(m,\alpha,\lambda)(1 + m) - \int \mathcal{D}z \ln \cosh\left(A(m,\alpha,\lambda) + z\sqrt{A(m,\alpha,\lambda)}\right)
$$

$$
+ \frac{\alpha}{2p}\ln\left(1 + \lambda^2(1 - m^p)\right). \qquad (104)
$$

### 3.4.3  Gaussian prior and additive noise

Using Eq. (69) and Eq. (79) together with $Q_0 = 1$ (which we found in sec. 3.4.1), we obtain the total replicated free-energy as

$$
\begin{aligned}
\frac{-F(Q,q,m)}{NM} &= \frac{1}{2}\left[\ln\left(Q - q + n(q - m^2)\right) + (n-1)\ln(Q - q)\right] - \frac{1}{2} - \frac{n}{2}Q \\
&+ \frac{\alpha}{p}\int \mathrm{d}y W(y) \int \mathcal{D}z_0 \left(\int \mathcal{D}z_1 W(y + \sqrt{1 - 2m^p + q^p}z_0 + \sqrt{Q^p - q^p}z_1)\right)^n \quad (105)
\end{aligned}
$$

from which we find the free-energy Eq. (30) as

$$
\begin{aligned}
-f &= \frac{1}{2}\left[\frac{q - m^2}{Q - q} + \ln(Q - q)\right] - \frac{Q}{2} \\
&+ \frac{\alpha}{p}\ln \int \mathrm{d}y W(y) \int \mathcal{D}z_0 \ln\left(\int \mathcal{D}z_1 W(y + \sqrt{1 - 2m^p + q^p}z_0 + \sqrt{Q^p - q^p}z_1)\right) \quad (106)
\end{aligned}
$$

Extremizing the free-energy with respect to $Q$, $q$ and $m$ we find, using Eq. (70) and Eq. (80),

$$
\begin{aligned}
0 &= \frac{Q - 2q - m^2}{(Q - q)^2} - 1 - \lambda^2 \alpha Q^{p-1}\theta_1(\lambda,1,Q,m,q)), \\
0 &= \frac{q - m^2}{(Q - q)^2} - \lambda^2 \alpha q^{p-1}\theta_0(\lambda,1,Q,m,q), \\
0 &= -\frac{m}{Q - q} + \lambda^2 \alpha m^{p-1}(\theta_0(\lambda,m,q) - \theta_1(\lambda,1,Q,m,q)). \quad (107)
\end{aligned}
$$

In the Bayes-optimal case, $m = q$ must hold. As noted before in sec 3.3.1 we find $\theta_1 = 0$ if $m = q$ and $Q_0 = Q$. In this case the first equation of Eq. (107) become satisfied and the last two equations of Eq. (107) become the same. Thus by solving the latter we find a self-consistent solution with $q = m$ and $Q = Q_0 = 1$.

We will analyze specifically the Bayes-optimal case with Gaussian noise. In this case one finds, using Eq. (84),

$$m = \frac{\alpha\lambda^2 m^{p-1}}{1 + \lambda^2(1 - m^p) + \alpha\lambda^2 m^{p-1}}. \tag{108}$$

For this case, the free-energy Eq. (106) becomes,

$$-f(m, \alpha, \lambda) = \frac{1}{2}\left[m + \ln(1 - m)\right] + \frac{\alpha}{p}\ln\int dy W(y)\ln\left(\int \mathcal{D}z W(y + \sqrt{1 - m^p}z)\right). \tag{109}$$

### 3.4.4 Gaussian prior and sign output

Using Eq. (69) and Eq. (90) together with $Q_0 = 1$ (which we found in sec. 3.4.1) we obtain the total replicated free-energy as

$$
\begin{aligned}
\frac{-F(Q, q, m)}{NM} &= \frac{1}{2}\left[\ln\left(Q - q + n(q - m^2)\right) + (n - 1)\ln(-q)\right] - \frac{1}{2} - \frac{n}{2}Q \\
&\quad + \frac{\alpha}{p}\ln\int \mathcal{D}z_0 H(X_{\text{teacher}})\int \mathcal{D}z_1 H^n(X_{\text{student}}),
\end{aligned}
$$

with $H(x)$ defined in Eq. (91) and $X_{\text{teacher}}$ and $X_{\text{student}}$ defined in Eq. (92). Then we find the free-energy Eq. (30) as

$$-f = \frac{1}{2}\left[\frac{q - m^2}{Q - q} + \ln(Q - q)\right] - \frac{Q}{2} + \frac{\alpha}{p}\ln\int \mathcal{D}z_0 H(X_{\text{teacher}})\int \mathcal{D}z_1 \ln H(X_{\text{student}}). \tag{110}$$

Extremizing the free-energy with respect to $Q$, $q$ and $m$ we find, using Eq. (70) Eq. (93), Eq. (94) and Eq. (95),

$$
\begin{aligned}
0 &= \frac{Q - 2q - m^2}{(Q - q)^2} - 1 + \frac{\lambda^2\alpha Q^{p-1}}{Q^p - q^p}\int \mathcal{D}z_0 H(X_{\text{teacher}})\int \mathcal{D}z_1 \frac{H''(X_{\text{student}})}{H(X_{\text{student}})}, \\
0 &= \frac{q - m^2}{(Q - q)^2} - \frac{2\lambda^2\alpha q^{p-1}}{Q^p - q^p}\int \mathcal{D}z_0 H(X_{\text{teacher}})\int \mathcal{D}z_1 \left(\frac{H'(X_{\text{student}})}{H(X_{\text{student}})}\right)^2, \\
0 &= -\frac{m}{1 - q} + \frac{\lambda^2\alpha m^{p-1}}{1 - m^p}\int \mathcal{D}z_0 H'(X_{\text{teacher}})\int \mathcal{D}z_1 \frac{H'(X_{\text{student}})}{H(X_{\text{student}})}. 
\end{aligned}
\tag{111}
$$

We find the 1st equation becomes satisfied in the Bayes optimal case $q = m$ with $Q_0 = Q$.

In the Bayes optimal case $q = m$ must hold. Assuming also $Q = 1$ we find the last two equations shown above become the same,

$$0 = -\frac{m}{1 - m} + \frac{\lambda^2\alpha m^{p-1}}{1 - m^p}\int \mathcal{D}z_0 \frac{(H'(X))^2}{H(X)}, \qquad X = -\frac{\sqrt{m^p}}{\sqrt{1 - m^p}}z. \tag{112}$$

By solving this equation we find a self-consistent solution with $q = m$ and $Q = Q_0 = 1$.

In the Bayes optimal case $q = m$ and assuming $Q = 1$, the free-energy Eq. (110) becomes,

$$-f = \frac{1}{2}\left[\frac{q - m^2}{1 - m} + \ln(1 - m)\right] + \frac{\alpha}{p}\ln\int \mathcal{D}z H(X)\ln H(X), \qquad X = -\frac{\sqrt{m^p}}{\sqrt{1 - m^p}}z. \tag{113}$$

## 3.5 Error correcting code

Similarly to the Sourlas code [39], our inference problem with the Ising prior and additive Gaussian noise can be viewed as an error correcting code. The teacher (encoder) is trying to send $NM$ bits (ground truth) encoded into $N_\blacksquare$ data and student (decoder) is trying to infer the ground truth given the data. The transmission rate is given by

$$R = \frac{NM}{N_\blacksquare} = \frac{p}{\alpha}. \tag{114}$$

Now let us consider to take $p \to \infty$, $\alpha \to \infty$ with the ratio $R$ fixed. In this limit Eq. (103) admits only two solutions

$$m = 0, 1. \tag{115}$$

Thus in this limit the inference completely fails $m = 0$ or succeed perfectly $m = 1$. From Eq. (104) we find the free-energy associated with the two solutions as,

$$-f(m = 0) = \ln 2 - \frac{1}{2R} \left[ 1 + \ln(2\pi) + \ln\left(1 + \lambda^2\right) \right], \tag{116}$$

$$-f(m = 1) = -\frac{1}{2R} \left[ 1 + \ln(2\pi) \right]. \tag{117}$$

From this we find there is a 1st order transition at $\lambda_c$ given by

$$R = \frac{1}{2} \ln_2(1 + \lambda_c^2), \tag{118}$$

or

$$\lambda_c = \sqrt{2R \log 2 - 1}. \tag{119}$$

For $\lambda < \lambda_c$ the paramagnetic solution $m = 0$ is the thermodynamically relevant one which is replaced by the ferromagnetic solution $m = 1$ for $\lambda > \lambda_c$.

On the other hand the capacity of Gaussian channel is given by

$$C = \frac{1}{2} \ln_2(1 + \lambda^2), \tag{120}$$

thus we find

$$C(\lambda_c) = R. \tag{121}$$

This means that the Shannon's bound [41]

$$R \leq C(\lambda) \tag{122}$$

is satisfied for $\lambda \geq \lambda_c$ and the lower bound is attained in the limit $\lambda \to \lambda_c^+$.

Thus our code can attain the Shannon bound taking $p \to \infty$ limit similarly to the Sourlas code. However the transmission rate $R$ can remain finite by considering $p, \alpha \to \infty$ with the ratio $R = p/\alpha$ being fixed. On the other hand $R \propto 1/N^{p-1}$ in the Sourlas code so that $R \to 0$ in the thermodynamic limit $N \to \infty$.

## 3.6 Some remarks on non-linearity of effective potentials

Each of our inference problems can be viewed as a system in which $x_{i\mu}$s are interacting with each other as described by an effective Hamiltonian,

$$H = \sum_\blacksquare V(\pi_\blacksquare) \qquad V(\pi_\blacksquare) = -\ln P_{\text{out}}(y_\blacksquare | \pi_\blacksquare) \tag{123}$$

Here we omitted local potential terms due to $P_{\mathrm{pri}}(x_{i\mu})$. In the cases that we considered in the present paper, the effective potentials $V(\pi_\blacksquare)$ are non-linear functions of $\pi_\blacksquare$. Importantly our problems cannot be linealized and this point brings about non-trivial consequences as we discuss below.

In sec. 2.3.3 we mentioned that our present problem is exactly the inverse problem or *planted version* of the vectorial constraint satisfaction problems (CSP) studied in [11]. In the case of 'linear potential' $V(x) = Jx$ (see sec. Eq. (26)), it was found that the vectorial CSP problems in the dense limit become essentially equivalent to the $M = 1$ problem which are the usual mean-field $p$-spin spinglass models. However such correspondences disappear for non-linear potentials. For instance, in the case of quadratic potential $V(x) = \frac{\epsilon}{2}x^2$ Eq. (27), full RSB was found even for $p = 2$ *spherical* model (continuous $x_{j\mu}$ with the constraint $\sum_{\mu=1}^{M} x_{j\mu}^2 = M$) while such RSB is absent in the case of linear potential for the $p = 2$ spherical model (spherical SK model [54]).

Similarly we believe that our vectorial inference problem do *not* reduce to rank-one $M = 1$ problem in general except for the special case of linear potential $V(x) = Jx$, i.e. $\ln P_{\mathrm{out}}(y_\blacksquare|\pi_\blacksquare) \propto -\pi_\blacksquare$. Indeed in the case of the Ising model with additive Gaussian noise $\ln P_{\mathrm{out}}(y_\blacksquare|\pi_\blacksquare) \propto -(y_\blacksquare - \pi_\blacksquare)^2$, which can be viewed as a system with quadratic potential $V(x) = \frac{\epsilon}{2}x^2$, we obtained the equation of state Eq. (103) which has a term $\lambda^2(1 - m^p)$ in the denominator. Such a term is absent in the case of the Sourlas code which is a rank-one $M = 1$ problem with additive Gaussian noise. As the result some properties of our model are *qualitatively* different from those of the Sourlas code. Most importantly the phase behavior in the $p = 2$ case: mixture of 2nd and 1st order transitions as discussed in sec. 5.1.1. Such a feature is absent in the case of the Sourlas code.

Let us discuss more closely the case of the quadratic effective potential,

$$
\begin{aligned}
H &= -\frac{1}{2}\sum_\blacksquare \left((\pi_\blacksquare^* + w_\blacksquare) - \pi_\blacksquare\right)^2 & (124) \\
&= \sum_\blacksquare (\pi_\blacksquare^* + w_\blacksquare)\pi_\blacksquare - \frac{1}{2}\sum_\blacksquare (\pi_\blacksquare)^2 + \mathrm{const}, & (125)
\end{aligned}
$$

where $w_\blacksquare$ is a quenched Gaussian random number with zero mean and unit variance and

$$
\pi_\blacksquare = \frac{\Lambda}{\sqrt{c}}\sum_{\mu=1}^{M} F_{\blacksquare\mu}\prod_{j\in\partial\blacksquare} x_{j\mu}. \tag{126}
$$

In the last equation of Eq. (125), "const" represents a irrelevant constant term which does not involve $\pi_\blacksquare$.

1. Globally coupled systems $c \propto N^{p-1}$

   In this case $\pi_\blacksquare$ scales as $\pi_\blacksquare = O((MN^{1-p})^{1/2})$ with $p \geq 2$. Here we assumed that the sum $\sum_{\mu=1}^{M}\cdots$ scales as $O(M^{1/2})$. This is valid as long as correlation between different components $\mu$ can be neglected. Thus for $M = N^a$ we find $\pi_\blacksquare \to 0$ in $N \to \infty$ limit if the exponent $a$ is small enough. This applies to the case of the Soulas code ($M = 1$) and sub-linearly high-rank cases if $0 < a < p - 1$. In such a situation the 2nd term $\pi_\blacksquare^2$ in the last equation Eq. (125) can be safely dropped in $N \to \infty$ and the problem reduces to models linear in $\pi_\blacksquare$.

   Indeed the Sourlas code can be mapped exactly to the ferromagnetically biased $p$-spin Ising spin-glass model [55–57]. More recent findings that sub-linearly high-rank models with global coupling [10, 31, 32] become essentially equivalent to rank-one $M = 1$ cases is consistent with the above observation.

The situation is different if the exponent $a$ is not small enough, including the case of full rank $M = O(N)$ case ($a = 1$) in matrix factorization ($p = 2$) [5,6]. In such cases the problems *cannot* be reduced to models linear in $\pi_\blacksquare$.

2. Our model $c \propto M$ ($\Lambda = \lambda/\sqrt{\alpha}$)

   The situation is very different in this case: $\pi = O(1)$ so that the 2nd term $\pi_\blacksquare^2$ in the last equation Eq. (125) cannot be dropped. Thus our problem cannot be reduced to models linear in $\pi_\blacksquare$.

## 3.7   Independence on choices of factor $F$

In the results presented in this section one can notice that expressions for the macroscopic quantities like the order-parameters, equation of states, free-energy... are the same for the deterministic model with $F = 1$ Eq. (9) and the disordered model with random $F$ Eq. (10). Note however that the remaining global symmetries (see sec 2.2) are different between the two cases.

# 4 Message Passing Algorithms

In this section we develop a message passing approach for the present problem.

## 4.1 Belief Propagation (BP)

Based on the graphical model in Fig. 1, we here introduce the well-known belief propagation (BP) algorithm. The graph composed of gray circles and squares corresponds to the random and sparse observations, and it forms a sparse random graph. Therefore, the BP algorithm that uses the messages for $\boldsymbol{x}_i = (x_{i1}, x_{i2}, \ldots, x_{iM})^\top$ corresponding to the gray circles is expected to yield asymptotically exact results. Fortunately, in our dense limit, a further simplification is possible. Specifically, even if we consider messages and BP for each variable component $x_{i\mu}$ corresponding to the green circles, the results are still expected to be asymptotically exact. This is precisely the same situation in which BP becomes asymptotically exact in fully connected models. In summary, for two reasons–the randomness and sparsity of observations and the divergence of $M$–the use of BP for each $x_{i\mu}$ is justified.

The BP equations for the messages of $x_{i\mu}$ can be written formally as

$$\tilde{\phi}^t_{\blacksquare \to i\mu}(x_{i\mu}) \qquad = \tfrac{1}{\tilde{z}^t_{\blacksquare \to i\mu}} \left\langle P_{\text{out}}\big(y_\blacksquare | \pi_\blacksquare\big) \right\rangle^{\backslash i\mu}_t, \tag{127}$$

$$\phi^{t+1}_{i\mu \to \blacksquare}(x_{i\mu}) = \tfrac{1}{z^{t+1}_{i\mu \to \blacksquare}} P_{\text{pri.}}(x_{i\mu}) \prod_{\blacksquare' \in \partial i \backslash \blacksquare} \tilde{\phi}^t_{\blacksquare' \to i\mu}(x_{i\mu}), \tag{128}$$

where we introduced an average defined as

$$\langle \cdots \rangle^{\backslash i\mu}_t = \int \left( \prod_{j \in \partial \blacksquare \backslash i} \prod_{\nu=1}^M \mathrm{d}x_{j\nu} \, \phi^t_{j\nu \to \blacksquare}(x_{j\nu}) \right) \left( \prod_{\nu(\neq\mu)} \mathrm{d}x_{i\nu} \, \phi^t_{i\nu \to \blacksquare}(x_{i\nu}) \right) (\cdots). \tag{129}$$

Note that the statistical weight used in the average is completely factorized into the product of the messages $\phi^t_{j\nu \to \blacksquare}(x_{j\nu})$ from different $(j, \nu)$s. Thus we have for any observables $\langle A(x_{j_1\nu_1}) B(x_{j_2\nu_2}) \rangle^{\backslash i\mu}_t = \langle A(x_{j_1\nu_1}) \rangle^{\backslash i\mu}_t \langle B(x_{j_2\nu_2}) \rangle^{\backslash i\mu}_t$ unless $(j_1, \nu_1) = (j_2, \nu_2)$.

## 4.2 From BP to r-BP

In the regime $M \gg 1$ which we are interested in, the BP equations become intractable. In the following we derive r-BP (relaxed BP) [8, 58] which is suitable for $M \gg 1$.

### 4.2.1 The 1st BP equation

We first consider the 1st BP equation Eq. (127). Let us introduce a Fourier representation of $P_{\text{out}}(y_\blacksquare | \pi_\blacksquare)$

$$P_{\text{out}}(y_\blacksquare | \pi_\blacksquare) = \int \tfrac{\mathrm{d}k}{2\pi} \, \mathrm{d}z \, e^{ik(\pi_\blacksquare - z)} P_{\text{out}}(y_\blacksquare | z). \tag{130}$$

Then we can rewrite Eq. (127) as,

$$\tilde{\phi}^t_{\blacksquare \to i\mu}(x_{i\mu}) = \tfrac{1}{\tilde{z}^t_{\blacksquare \to i\mu}} \int \tfrac{\mathrm{d}k}{2\pi} \, \mathrm{d}z \, \left\langle e^{ik\pi_\blacksquare} \right\rangle^{\backslash i\mu}_t e^{-ikz} P_{\text{out}}(y_\blacksquare | z). \tag{131}$$

Now we evaluate $\left\langle e^{ik\pi_\blacksquare} \right\rangle^{\backslash i\mu}_t$ by a cumulant expansion,

$$\ln \left\langle e^{ik\pi_\blacksquare} \right\rangle^{\backslash i\mu}_t = ik O^{(1),t}_{\blacksquare \to i\mu} - \tfrac{1}{2}k^2 O^{(2),t}_{\blacksquare \to i\mu} - \tfrac{1}{6}ik^3 O^{(3),t}_{\blacksquare \to i\mu} + \ldots. \tag{132}$$

Using Eq. (129) and the comment mentioned there, we find the first few moments as,

$$
\begin{aligned}
O^{(1),t}_{\blacksquare\to i\mu} &= \langle \pi_\blacksquare \rangle^{\setminus i\mu}_t \\
&= \frac{\lambda}{\sqrt{M}} F_{\blacksquare,\mu} x_{i\mu} \prod_{j\in\partial\blacksquare\setminus i} m^t_{j\mu\to\blacksquare} + \sum_{\nu(\neq\mu)} \frac{\lambda}{\sqrt{M}} F_{\blacksquare\nu} \prod_{j\in\partial\blacksquare} m^t_{j\nu\to\blacksquare} \\
&= \frac{\lambda}{\sqrt{M}} F_{\blacksquare\mu} \rho^t_{\blacksquare\setminus i,\mu} x_{i\mu} + \omega^t_{\blacksquare\to\mu},
\end{aligned}
\tag{133}
$$

$$
\begin{aligned}
O^{(2),t}_{\blacksquare\to i\mu} &= \left\langle (\pi_\blacksquare)^2 \right\rangle^{\setminus i\mu}_t - \left( \langle \pi_\blacksquare \rangle^{\setminus i\mu}_t \right)^2 \\
&= \left( \frac{\lambda}{\sqrt{M}} F_{\blacksquare\mu} \right)^2 x_{i\mu}^2 \left( \prod_{j\in\partial\blacksquare\setminus i} v^t_{j\mu\to\blacksquare} - \prod_{j\in\partial\blacksquare\setminus i} \left( m^t_{j\mu\to\blacksquare} \right)^2 \right) \\
&\quad + \sum_{\nu(\neq\mu)} \left( \frac{\lambda}{\sqrt{M}} F_{\blacksquare\nu} \right)^2 \left( \prod_{j\in\partial\blacksquare} v^t_{j\nu\to\blacksquare} - \prod_{j\in\partial\blacksquare} \left( m^t_{j\nu\to\blacksquare} \right)^2 \right)
\end{aligned}
\tag{134}
$$

$$
= V^t_{\blacksquare\to\mu} + \mathcal{O}\left( M^{-1} \right),
\tag{135}
$$

$$
\begin{aligned}
O^{(3),t}_{\blacksquare\to i\mu} &= \left\langle (\pi_\blacksquare)^3 \right\rangle^{\setminus i\mu}_t - 3 \langle \pi_\blacksquare \rangle^{\setminus i\mu}_t \left\langle (\pi_\blacksquare)^2 \right\rangle^{\setminus i\mu}_t + 2 \left( \langle \pi_\blacksquare \rangle^{\setminus i\mu}_t \right)^3 \\
&= \sum_{\nu(\neq\mu)} \left( \frac{\lambda}{\sqrt{M}} F_{\blacksquare\nu} \right)^3 \left( \prod_{j\in\partial\blacksquare} t^t_{j\nu\to\blacksquare} - 3 \prod_{j\in\partial\blacksquare} m^t_{j\nu\to\blacksquare} v^t_{j\nu\to\blacksquare} + 2 \prod_{j\in\partial\blacksquare} \left( m^t_{j\nu\to\blacksquare} \right)^3 \right) + \mathcal{O}\left( M^{-1} \right)
\end{aligned}
$$

$$
= \mathcal{O}\left( M^{-1/2} \right).
\tag{136}
$$

Here we introduced moments of $x_{i\mu}$ associated with the distribution function $\phi^t_{i\mu\to\blacksquare}(x_{i\mu})$ (128),

$$
m^t_{i\mu\to\blacksquare} = \int \mathrm{d}x_{i\mu}\, \phi^t_{i\mu\to\blacksquare}(x_{i\mu})\, x_{i\mu},
\tag{137}
$$

$$
v^t_{i\mu\to\blacksquare} = \int \mathrm{d}x_{i\mu}\, \phi^t_{i\mu\to\blacksquare}(x_{i\mu})\, x_{i\mu}^2,
\tag{138}
$$

$$
t^t_{i\mu\to\blacksquare} = \int \mathrm{d}x_{i\mu}\, \phi^t_{i\mu\to\blacksquare}(x_{i\mu})\, x_{i\mu}^3.
\tag{139}
$$

We call these also as messages. Furthermore we also introduced

$$
\rho^t_{\blacksquare\setminus i,\mu} = \prod_{j\in\partial\blacksquare\setminus i} m^t_{j\mu\to\blacksquare},
\tag{140}
$$

$$
\omega^t_{\blacksquare\to\mu} = \sum_{\nu(\neq\mu)} \frac{\lambda}{\sqrt{M}} F_{\blacksquare\nu} \prod_{j\in\partial\blacksquare} m^t_{j\nu\to\blacksquare},
\tag{141}
$$

$$
V^t_{\blacksquare\to\mu} = \sum_{\nu(\neq\mu)} \left( \frac{\lambda}{\sqrt{M}} F_{\blacksquare\nu} \right)^2 \left( \prod_{j\in\partial\blacksquare} v^t_{j\nu\to\blacksquare} - \prod_{j\in\partial\blacksquare} \left( m^t_{j\nu\to\blacksquare} \right)^2 \right).
\tag{142}
$$

For $M \gg 1$, we can drop terms of $\mathcal{O}\left( M^{-1/2} \right)$ so that the BP equation (127) becomes,

$$
\begin{aligned}
\tilde{\phi}^t_{\blacksquare\to i\mu}(x_{i\mu}) &\propto \int \mathrm{d}k\, \mathrm{d}z \exp\left[ ik\left( \frac{\lambda}{\sqrt{M}} F_{\blacksquare\mu} \rho^t_{\blacksquare\setminus i,\mu} x_{i\mu} + \omega^t_{\blacksquare\to\mu} - z \right) - \frac{1}{2} k^2 V^t_{\blacksquare\to\mu} \right] P_{\mathrm{out}}(y_\blacksquare | z) \\
&\propto \int \mathrm{d}z \exp\left[ -\frac{1}{2 V^t_{\blacksquare\to\mu}} \left( \frac{\lambda}{\sqrt{M}} F_{\blacksquare\mu} \rho^t_{\blacksquare\setminus i,\mu} x_{i\mu} + \omega^t_{\blacksquare\to\mu} - z \right)^2 \right] P_{\mathrm{out}}(y_\blacksquare | z) \\
&\propto \exp\left[ -\frac{1}{2 V^t_{\blacksquare\to\mu}} \left( \frac{\lambda}{\sqrt{M}} F_{\blacksquare\mu} \rho^t_{\blacksquare\setminus i,\mu} \right)^2 x_{i\mu}^2 \right] \left\langle e^{\frac{1}{V^t_{\blacksquare\to\mu}} \left( z - \omega^t_{\blacksquare\to\mu} \right) \frac{\lambda}{\sqrt{M}} F_{\blacksquare\mu} \rho^t_{\blacksquare\setminus i,\mu} x_{i\mu}}; \omega^t_{\blacksquare\to\mu}, y_\blacksquare, V^t_{\blacksquare\to\mu} \right\rangle_{\mathrm{out}}
\end{aligned}
\tag{143}
$$

Here we defined $\langle \cdots ; \omega, y, V \rangle_{\text{out}}$,

$$\langle \cdots ; \omega, y, V \rangle_{\text{out}} \quad = \frac{\int \mathrm{d}z\, P_{\text{out}}(y|z)\, (\cdots)\, \exp\left(-\frac{1}{2V}(z-\omega)^2\right)}{\int \mathrm{d}z\, P_{\text{out}}(y|z)\, \exp\left(-\frac{1}{2V}(z-\omega)^2\right)}. \tag{144}$$

Again we can evaluate $\langle \cdots ; \omega, y, V \rangle_{\text{out}}$ in Eq. (143) by a cumulant expansion,

$$\ln \left\langle e^{\frac{1}{V}(z-\omega)\frac{\lambda}{\sqrt{M}}X\rho x}; \omega, y, V \right\rangle_{\text{out}}$$

$$= g_{\text{out}}(\omega, y, V)\frac{\lambda}{\sqrt{M}}X\rho x + \frac{1}{2}\left(g_{\text{out,II}}(\omega, y, V) - g_{\text{out}}^2(\omega, y, V)\right)\left(\frac{\lambda}{\sqrt{M}}X\right)^2 \rho^2 x^2$$

$$+\mathcal{O}\left(M^{-3/2}\right), \tag{145}$$

where we defined 'output functions'

$$g_{\text{out}}(\omega, y, V) \quad = \quad \frac{\int \mathrm{d}z\, P_{\text{out}}(y|z)(z-\omega)e^{-(z-\omega)^2/2V}}{V \int \mathrm{d}z\, P_{\text{out}}(y|z)e^{-(z-\omega)^2/2V}}, \tag{146}$$

$$g_{\text{out,II}}(\omega, y, V) \quad = \quad \frac{\int \mathrm{d}z\, P_{\text{out}}(y|z)(z-\omega)^2 e^{-(z-\omega)^2/2V}}{V^2 \int \mathrm{d}z\, P_{\text{out}}(y|z)e^{-(z-\omega)^2/2V}}, \tag{147}$$

which verify an identity

$$g_{\text{out,II}}(\omega, y, V) \quad = \frac{1}{V} + \frac{\partial}{\partial \omega}g_{\text{out}}(\omega, y, V) + g_{\text{out}}^2(\omega, y, V). \tag{148}$$

Using Eq. (145) and Eq. (148) we find Eq. (143) becomes,

$$\tilde{\phi}^t_{\blacksquare \to i\mu}(x_{i\mu}) \quad \propto \quad \exp\left(-\frac{1}{2}A^t_{\blacksquare \to i\mu}x_{i\mu}^2 + B^t_{\blacksquare \to i\mu}x_{i\mu}\right), \tag{149}$$

where we defined,

$$A^t_{\blacksquare \to i\mu} \quad = \quad -\left(\frac{\lambda}{\sqrt{M}}F_{\blacksquare\mu}\rho^t_{\blacksquare\backslash i,\mu}\right)^2 \frac{\partial}{\partial \omega}g_{\text{out}}\left(\omega^t_{\blacksquare \to \mu}, y_\blacksquare, V^t_{\blacksquare \to \mu}\right), \tag{150}$$

$$B^t_{\blacksquare \to i\mu} \quad = \quad \frac{\lambda}{\sqrt{M}}F_{\blacksquare\mu}\rho^t_{\blacksquare\backslash i,\mu}\, g_{\text{out}}\left(\omega^t_{\blacksquare \to \mu}, y_\blacksquare, V^t_{\blacksquare \to \mu}\right). \tag{151}$$

As shown in Eq. (140), Eq. (141) and Eq. (142), the parameters $\rho, \omega$ and $V$ depend on the messages $m^t_{i\mu \to \blacksquare}$ and $v^t_{i\mu \to \blacksquare}$ defined in Eq. (137) and Eq. (138).

### 4.2.2 The 2nd BP equation

Using the above result, we can write the 2nd BP equation (128) as,

$$\phi^{t+1}_{i\mu \to \blacksquare}(x_{i\mu}) \quad \propto P_{\text{pri.}}(x_{i\mu})\prod_{\blacksquare' \in \partial i \backslash \blacksquare}\exp\left(-\frac{1}{2}A^t_{\blacksquare' \to i\mu}x_{i\mu}^2 + B^t_{\blacksquare' \to i\mu}x_{i\mu}\right)$$

$$\propto P_{\text{pri.}}(x_{i\mu})\exp\left(-\frac{1}{2\,\Sigma^{t+1}_{i\mu \to \blacksquare}}\left(x_{i\mu} - T^{t+1}_{i\mu \to \blacksquare}\right)^2\right). \tag{152}$$

Here we introduced

$$\Sigma^{t+1}_{i\mu \to \blacksquare} \quad = \frac{1}{\sum_{\blacksquare' \in \partial i \backslash \blacksquare}A^t_{\blacksquare' \to i\mu}}, \qquad T^{t+1}_{i\mu \to \blacksquare} = \Sigma^{t+1}_{i\mu \to \blacksquare} \times \sum_{\blacksquare' \in \partial i \backslash \blacksquare}B^t_{\blacksquare' \to i\mu}. \tag{153}$$

### 4.2.3  Back to the 1st BP equation

When we analyzed the 1st BP equation in sec. 4.2.1, we found messages $m_{i\mu\to\blacksquare}^t$ and $v_{i\mu\to\blacksquare}^t$ defined in Eq. (137) and Eq. (138), are needed to be evaluated using $\phi_{i\mu\to\blacksquare}^t$. Using the results in sec 4.2.2 we can write these using $\Sigma_{i\mu\to\blacksquare}^t$ and $T_{i\mu\to\blacksquare}^t$ as,

$$
\begin{aligned}
m_{i\mu\to\blacksquare}^t &= \int \mathrm{d}x_{i\mu}\, \phi_{i\mu\to\blacksquare}^t(x_{i\mu})\, x_{i\mu} = f_{\text{input}}\Big(\Sigma_{i\mu\to\blacksquare}^t, T_{i\mu\to\blacksquare}^t\Big), & (154)
\end{aligned}
$$

$$
\begin{aligned}
v_{i\mu\to\blacksquare}^t &= \int \mathrm{d}x_{i\mu}\, \phi_{i\mu\to\blacksquare}^t(x_{i\mu})\, x_{i\mu}^2 = f_{\text{input,II}}\Big(\Sigma_{i\mu\to\blacksquare}^t, T_{i\mu\to\blacksquare}^t\Big) \\
&= \Sigma_{i\mu\to\blacksquare}^t \times \frac{\partial}{\partial T} f_{\text{input}}\Big(\Sigma_{i\mu\to\blacksquare}^t, T_{i\mu\to\blacksquare}^t\Big) + f_{\text{input}}^2\Big(\Sigma_{i\mu\to\blacksquare}^t, T_{i\mu\to\blacksquare}^t\Big), & (155)
\end{aligned}
$$

where we defined

$$
f_{\text{input}}(\Sigma, T) = \frac{\int \mathrm{d}x\, P_{\text{pri.}}(x)\, x\, e^{-(x-T)^2/2\Sigma}}{\int \mathrm{d}x\, P_{\text{pri.}}(x) e^{-(x-T)^2/2\Sigma}}, \tag{156}
$$

$$
f_{\text{input,II}}(\Sigma, T) = \frac{\int \mathrm{d}x\, P_{\text{pri.}}(x)\, x^2\, e^{-(x-T)^2/2\Sigma}}{\int \mathrm{d}x\, P_{\text{pri.}}(x) e^{-(x-T)^2/2\Sigma}}. \tag{157}
$$

Note that an identity

$$
f_{\text{input,II}}(\Sigma, T) = \Sigma \times \tfrac{\partial}{\partial T} f_{\text{input}}(\Sigma, T) + f_{\text{input}}^2(\Sigma, T) \tag{158}
$$

holds.

### 4.2.4  Marginal distribution and the moments

The marginal distribution is obtained as,

$$
\begin{aligned}
\phi_{i\mu}^{t+1}(x_{i\mu}) &= \frac{1}{z_{i\mu}^{t+1}} P_{\text{pri.}}(x_{i\mu}) \prod_{\blacksquare'\in\partial i} \tilde{\phi}_{\blacksquare'\to i\mu}^t(x_{i\mu}) \\
&\propto P_{\text{pri.}}(x_{i\mu}) \exp\left(-\frac{1}{2\,\Sigma_{i\mu}^{t+1}}\Big(x_{i\mu} - T_{i\mu}^{t+1}\Big)^2\right), & (159)
\end{aligned}
$$

with

$$
(\Sigma_{i\mu}^{t+1})^{-1} = -\sum_{\blacksquare\in\partial i}\left(\frac{\lambda}{\sqrt{M}} F_{\blacksquare\mu}\rho_{\blacksquare\backslash i,\mu}^t\right)^2 \frac{\partial}{\partial\omega} g_{\text{out}}\Big(\omega_{\blacksquare\to\mu}^t, y_{\blacksquare}, V_{\blacksquare\to\mu}^t\Big), \tag{160}
$$

$$
\frac{T_{i\mu}^{t+1}}{\Sigma_{i\mu}^{t+1}} = \sum_{\blacksquare\in\partial i}\frac{\lambda}{\sqrt{M}} F_{\blacksquare\mu}\rho_{\blacksquare\backslash i,\mu}^t\, g_{\text{out}}\Big(\omega_{\blacksquare\to\mu}^t, y_{\blacksquare}, V_{\blacksquare\to\mu}^t\Big). \tag{161}
$$

The 1st and 2nd moments of the marginal distribution function (159) are obtained as,

$$
m_{i\mu}^{t+1} = \int \mathrm{d}x_{i\mu}\, \phi_{i\mu}^{t+1}(x_{i\mu})\, x_{i\mu} = f_{\text{input}}\Big(\Sigma_{i\mu}^{t+1}, T_{i\mu}^{t+1}\Big), \tag{162}
$$

$$
v_{i\mu}^{t+1} = \int \mathrm{d}x_{i\mu}\, \phi_{i\mu}^{t+1}(x_{i\mu})\, x_{i\mu}^2 = f_{\text{input,II}}\Big(\Sigma_{i\mu}^{t+1}, T_{i\mu}^{t+1}\Big). \tag{163}
$$

Collecting the results we obtain the r-BP algorithm 1.

---

**Algorithm 1** r-BP algorithm

---

**Input:** $\mathbb{G} = (\mathtt{V}, \mathtt{E})$, matrix $F$, input function $f_{\text{input}}$, output function $g_{\text{out}}$

1: **initialization:**    Initialize messages (see sec. D) $m_{i\mu\to\blacksquare}^t$ and $v_{i\mu\to\blacksquare}^t$.

2: **repeat**

3:     Update $\omega_{\blacksquare\to\mu}^t$ and $V_{\blacksquare\to\mu}^t$.

$$\omega_{\blacksquare\to\mu}^t \leftarrow \sum_{\nu(\neq\mu)} \frac{\lambda}{\sqrt{M}} F_{\blacksquare\nu} \prod_{j\in\partial\blacksquare} m_{j\nu\to\blacksquare}^t,$$

$$V_{\blacksquare\to\mu}^t \leftarrow \sum_{\nu(\neq\mu)} \left(\frac{\lambda}{\sqrt{M}} F_{\blacksquare\nu}\right)^2 \left(\prod_{j\in\partial\blacksquare} v_{j\nu\to\blacksquare}^t - \prod_{j\in\partial\blacksquare} \left(m_{j\nu\to\blacksquare}^t\right)^2\right). \tag{Alg1.1}$$

4:     Update the values of the output functions.

$$g_{\blacksquare\to\mu}^t \leftarrow g_{\text{out}}\left(\omega_{\blacksquare\to\mu}^t, y_\blacksquare, V_{\blacksquare\to\mu}^t\right), \qquad \partial_\omega g_{\blacksquare\to\mu}^t \leftarrow \frac{\partial}{\partial\omega} g_{\text{out}}\left(\omega_{\blacksquare\to\mu}^t, y_\blacksquare, V_{\blacksquare\to\mu}^t\right). \tag{Alg1.2}$$

5:     Update $\Sigma_{i\mu\to\blacksquare}^{t+1}$ and $T_{i\mu\to\blacksquare}^{t+1}$.

$$\frac{1}{\Sigma_{i\mu\to\blacksquare}^{t+1}} \leftarrow \sum_{\blacksquare'\in\partial i\setminus\blacksquare} \left(\frac{\lambda}{\sqrt{M}} F_{\blacksquare'\mu}\right)^2 \left(-\partial_\omega g_{\blacksquare'\to\mu}^t\right) \prod_{j\in\partial\blacksquare'\setminus i} \left(m_{j\mu\to\blacksquare'}^t\right)^2,$$

$$\frac{T_{i\mu\to\blacksquare}^{t+1}}{\Sigma_{i\mu\to\blacksquare}^{t+1}} \leftarrow \sum_{\blacksquare'\in\partial i\setminus\blacksquare} \frac{\lambda}{\sqrt{M}} F_{\blacksquare'\mu} g_{\blacksquare'\to\mu}^t \prod_{j\in\partial\blacksquare'\setminus i} m_{j\mu\to\blacksquare'}^t. \tag{Alg1.3}$$

6:     Update messages $m_{i\mu\to\blacksquare}^{t+1}$ and $v_{i\mu\to\blacksquare}^{t+1}$.

$$m_{i\mu\to\blacksquare}^{t+1} \leftarrow f_{\text{input}}\left(\Sigma_{i\mu\to\blacksquare}^{t+1}, T_{i\mu\to\blacksquare}^{t+1}\right), \qquad v_{i\mu\to\blacksquare}^{t+1} \leftarrow f_{\text{input,II}}\left(\Sigma_{i\mu\to\blacksquare}^{t+1}, T_{i\mu\to\blacksquare}^{t+1}\right). \tag{Alg1.4}$$

7: **until** $m_{i\mu\to\blacksquare}^t$ and $v_{i\mu\to\blacksquare}^t$ converge.

**Output:** Messages $m_{i\mu}, v_{i\mu}$ which are the average and variance of the variables.

$$\frac{1}{\Sigma_{i\mu}} \leftarrow \sum_{\blacksquare\in\partial i} \left(\frac{\lambda}{\sqrt{M}} F_{\blacksquare\mu}\right)^2 \left(-\partial_\omega g_{\blacksquare\to\mu}\right) \prod_{j\in\partial\blacksquare\setminus i} \left(m_{j\mu\to\blacksquare}\right)^2,$$

$$\frac{T_{i\mu}}{\Sigma_{i\mu}} \leftarrow \sum_{\blacksquare\in\partial i} \frac{\lambda}{\sqrt{M}} F_{\blacksquare\mu} g_{\blacksquare\to\mu} \prod_{j\in\partial\blacksquare\setminus i} m_{j\mu\to\blacksquare},$$

$$m_{i\mu} \leftarrow f_{\text{input}}\left(\Sigma_{i\mu}, T_{i\mu}\right), \qquad v_{i\mu} \leftarrow f_{\text{input,II}}\left(\Sigma_{i\mu}, T_{i\mu}\right). \tag{Alg1.5}$$

---

## 4.3   From r-BP to G-AMP

In the limit $M \gg 1$, we can further simplify the r-BP obtained in the previous section to derive AMP (approximate message passing) algorithm. While BP defines recursion formulae for 'messages' of the form $A_{\blacksquare\to i\mu}, B_{i\mu\to\blacksquare}$ between the factor and variable nodes, AMP defines a set of recursion formulae for completely local quantities like $A_\blacksquare, B_{i\mu}$. The AMP equations are derived from the r-BP ones through a perturbative manner which is essentially the same technique to derive the so called TAP (Thouless-Anderson-Palmer) equations in mean-field spin-glass models with global couplings.

From Eq. (141) and (142) we find,

$$\omega^t_{\blacksquare\to\mu} = \omega^t_{\blacksquare} - \frac{\lambda}{\sqrt{M}} F_{\blacksquare\mu} \prod_{j\in\partial\blacksquare} m^t_{j\mu\to\blacksquare}, \tag{164}$$

$$V^t_{\blacksquare\to\mu} = V^t_{\blacksquare} + \mathcal{O}\left(M^{-1}\right), \tag{165}$$

where we introduced,

$$\omega^t_{\blacksquare} = \sum_{\nu=1}^M \frac{\lambda}{\sqrt{M}} F_{\blacksquare\nu} \prod_{j\in\partial\blacksquare} m^t_{j\nu\to\blacksquare}, \qquad V^t_{\blacksquare} = \sum_{\nu=1}^M \left(\frac{\lambda}{\sqrt{M}} F_{\blacksquare\nu}\right)^2 \left(\prod_{j\in\partial\blacksquare} v^t_{j\nu\to\blacksquare} - \prod_{j\in\partial\blacksquare} \left(m^t_{j\nu\to\blacksquare}\right)^2\right) \tag{166}$$

From Eq. (153) and Eq. (151), we find,

$$\Sigma^{t+1}_{i\mu\to\blacksquare} = \Sigma^{t+1}_{i\mu} + \mathcal{O}\left(M^{-1}\right), \tag{167}$$

$$T^{t+1}_{i\mu\to\blacksquare} = T^{t+1}_{i\mu} - \Sigma^{t+1}_{i\mu} \frac{\lambda}{\sqrt{M}} F_{\blacksquare\mu} g_{\text{out}}\left(\omega^t_{\blacksquare}, y_{\blacksquare}, V^t_{\blacksquare}\right) \prod_{j\in\partial\blacksquare\backslash i} m^t_{j\mu\to\blacksquare} + \mathcal{O}\left(M^{-1}\right), \tag{168}$$

where we used an expansion of the output function (146),

$$g_{\text{out}}\left(\omega^t_{\blacksquare\to\mu}, y_{\blacksquare}, V^t_{\blacksquare\to\mu}\right) = g_{\text{out}}\left(\omega^t_{\blacksquare}, y_{\blacksquare}, V^t_{\blacksquare}\right) - \frac{\lambda}{\sqrt{M}} F_{\blacksquare\mu} \left(\prod_{j\in\partial\blacksquare} m^t_{j\mu\to\blacksquare}\right) \frac{\partial}{\partial\omega} g_{\text{out}}\left(\omega^t_{\blacksquare}, y_{\blacksquare}, V^t_{\blacksquare}\right) + \mathcal{O}\left(M^{-1}\right), \tag{169}$$

which follows from Eq. (165).

From Eq. (154) and (155) we find,

$$
\begin{aligned}
m^{t+1}_{i\mu\to\blacksquare} &= m^{t+1}_{i\mu} \\
&\quad - \frac{\partial}{\partial T} f_{\text{input}}\left(\Sigma^{t+1}_{i\mu}, T^{t+1}_{i\mu}\right) \Sigma^{t+1}_{i\mu} \frac{\lambda}{\sqrt{M}} F_{\blacksquare\mu} g_{\text{out}}\left(\omega^t_{\blacksquare}, y_{\blacksquare}, V^t_{\blacksquare}\right) \prod_{j\in\partial\blacksquare\backslash i} m^t_{j\mu\to\blacksquare} + \mathcal{O}\left(M^{-1}\right) \\
&= m^{t+1}_{i\mu} \\
&\quad - \left(f_{\text{input,II}}\left(\Sigma^{t+1}_{i\mu}, T^{t+1}_{i\mu}\right) - f^2_{\text{input}}\left(\Sigma^{t+1}_{i\mu}, T^{t+1}_{i\mu}\right)\right) \frac{\lambda}{\sqrt{M}} F_{\blacksquare\mu} g_{\text{out}}\left(\omega^t_{\blacksquare}, y_{\blacksquare}, V^t_{\blacksquare}\right) \prod_{j\in\partial\blacksquare\backslash i} m^t_{j\mu\to\blacksquare} \\
&\quad + \mathcal{O}\left(M^{-1}\right) \\
&= m^{t+1}_{i\mu} - \left(v^{t+1}_{i\mu} - \left(m^{t+1}_{i\mu}\right)^2\right) \frac{\lambda}{\sqrt{M}} F_{\blacksquare\mu} g_{\text{out}}\left(\omega^t_{\blacksquare}, y_{\blacksquare}, V^t_{\blacksquare}\right) \prod_{j\in\partial\blacksquare\backslash i} m^t_{j\mu} + \mathcal{O}\left(M^{-1}\right), \tag{170}
\end{aligned}
$$

$$
\begin{aligned}
v^{t+1}_{i\mu\to\blacksquare} &= v^{t+1}_{i\mu} - \frac{\partial}{\partial T} f_{\text{input,II}}\left(\Sigma^{t+1}_{i\mu}, T^{t+1}_{i\mu}\right) \Sigma^{t+1}_{i\mu} \frac{\lambda}{\sqrt{M}} F_{\blacksquare\mu} g_{\text{out}}\left(\omega^t_{\blacksquare}, y_{\blacksquare}, V^t_{\blacksquare}\right) \prod_{j\in\partial\blacksquare\backslash i} m^t_{j\mu} \\
&\quad + \mathcal{O}\left(M^{-1}\right), \tag{171}
\end{aligned}
$$

where we used an expansion of the input function (156),

$$
\begin{aligned}
f_{\text{input}}\left(\Sigma^t_{i\mu\to\blacksquare}, T^t_{i\mu\to\blacksquare}\right) &= f_{\text{input}}\left(\Sigma^t_{i\mu}, T^t_{i\mu}\right) \\
&\quad - \frac{\partial}{\partial T} f_{\text{input}}\left(\Sigma^t_{i\mu}, T^t_{i\mu}\right) \Sigma^t_{i\mu} \frac{\lambda}{\sqrt{M}} F_{\blacksquare\mu} g_{\text{out}}\left(\omega^{t-1}_{\blacksquare}, y_{\blacksquare}, V^{t-1}_{\blacksquare}\right) \prod_{j\in\partial\blacksquare\backslash i} m^{t-1}_{j\mu\to\blacksquare} \\
&\quad + \mathcal{O}\left(M^{-1}\right), \tag{172}
\end{aligned}
$$

which follows from Eq. (168).

Now collecting the above results we obtain the following, dropping $O(M^{-1})$ corrections,

$$
\omega_{\blacksquare}^t = \sum_{\nu=1}^{M} \frac{\lambda}{\sqrt{M}} F_{\blacksquare\nu} \left( \prod_{j\in\partial\blacksquare} m_{j\nu}^t \right.
$$

$$
\left. - \frac{\lambda}{\sqrt{M}} F_{\blacksquare\nu} g_{\text{out}}\left(\omega_{\blacksquare}^{t-1}, y_{\blacksquare}, V_{\blacksquare}^{t-1}\right) \sum_{j\in\partial\blacksquare} \left(v_{j\nu}^t - \left(m_{j\nu}^t\right)^2\right) \prod_{k\in\partial\blacksquare\backslash j} m_{k\nu}^t m_{k\nu}^{t-1} \right),
\tag{173}
$$

$$
V_{\blacksquare}^t = \sum_{\nu=1}^{M} \left(\frac{\lambda}{\sqrt{M}} F_{\blacksquare\nu}\right)^2 \left( \prod_{j\in\partial\blacksquare} v_{j\nu}^t - \prod_{j\in\partial\blacksquare} \left(m_{j\nu}^t\right)^2 \right),
\tag{174}
$$

$$
\Sigma_{i\mu}^{t+1} = -\frac{1}{\sum_{\blacksquare\in\partial i} \left(\frac{\lambda}{\sqrt{M}} F_{\blacksquare\mu}\right)^2 \frac{\partial}{\partial\omega} g_{\text{out}}\left(\omega_{\blacksquare}^t, y_{\blacksquare}, V_{\blacksquare}^t\right) \prod_{j\in\partial\blacksquare\backslash i} \left(m_{j\mu}^t\right)^2},
\tag{175}
$$

$$
\frac{T_{i\mu}^{t+1}}{\Sigma_{i\mu}^{t+1}} = \frac{m_{i\mu}^t}{\Sigma_{i\mu}^{t+1}} + \sum_{\blacksquare\partial i} \frac{\lambda}{\sqrt{M}} F_{\blacksquare\mu} g_{\text{out}}\left(\omega_{\blacksquare}^t, y_{\blacksquare}, V_{\blacksquare}^t\right) \left( \prod_{j\in\partial\blacksquare} m_{j\mu}^t \right.
$$

$$
\left. - \frac{\lambda}{\sqrt{M}} F_{\blacksquare\mu} g_{\text{out}}\left(\omega_{\blacksquare}^{t-1}, y_{\blacksquare}, V_{\blacksquare}^{t-1}\right) m_{i\mu}^{t-1} \sum_{j\in\partial\blacksquare} \left(v_{j\mu}^t - \left(m_{j\mu}^t\right)^2\right) \prod_{k\in\partial\blacksquare\backslash i,j} m_{k\mu}^t m_{k\mu}^{t-1} \right)
\tag{176}
$$

This set of equations gives the G-AMP algorithm 2.

---

**Algorithm 2** G-AMP algorithm

---

**Input:** graph $\mathbb{G} = (\mathtt{V}, \mathtt{E})$, matrix $F$, input function $f_{\text{input}}$, output function $g_{\text{out}}$.

1: **initialization:** Initialize messages (see sec. D) $m_{i\mu}^t$, $m_{i\mu}^{t-1}$, $v_{i\mu}^t$ and $g_{\blacksquare}^{t-1}$.

2: **repeat**

3: Update $\omega_{\blacksquare \to \mu}^t$ and $V_{\blacksquare \to \mu}^t$.

$$\omega_{\blacksquare}^t \leftarrow \sum_{\nu=1}^{M} \frac{\lambda}{\sqrt{M}} F_{\blacksquare\nu} \left( \prod_{j \in \partial\blacksquare} m_{j\nu}^t - \frac{\lambda}{\sqrt{M}} F_{\blacksquare\nu} g_{\blacksquare}^{t-1} \sum_{j \in \partial\blacksquare} \left( v_{j\nu}^t - \left( m_{j\nu}^t \right)^2 \right) \prod_{k \in \partial\blacksquare \setminus j} m_{k\nu}^t m_{k\nu}^{t-1} \right),$$

$$V_{\blacksquare}^t \leftarrow \sum_{\nu=1}^{M} \left( \frac{\lambda}{\sqrt{M}} F_{\blacksquare\nu} \right)^2 \left( \prod_{j \in \partial\blacksquare} v_{j\nu}^t - \prod_{j \in \partial\blacksquare} \left( m_{j\nu}^t \right)^2 \right).$$

(Alg2.1)

4: Update the values of the output function

$$g_{\blacksquare}^t \leftarrow g_{\text{out}} \left( \omega_{\blacksquare}^t, y_{\blacksquare}, V_{\blacksquare}^t \right), \qquad \partial_\omega g_{\blacksquare}^t \leftarrow \frac{\partial}{\partial \omega} g_{\text{out}} \left( \omega_{\blacksquare}^t, y_{\blacksquare}, V_{\blacksquare}^t \right).$$

(Alg2.2)

5: Update $\Sigma_{i\mu \to \blacksquare}^t$ and $T_{i\mu \to \blacksquare}^t$.

$$\frac{1}{\Sigma_{i\mu}^{t+1}} \leftarrow \sum_{\blacksquare \in \partial i} \left( \frac{\lambda}{\sqrt{M}} F_{\blacksquare\mu} \right)^2 \left( -\partial_\omega g_{\blacksquare}^t \right) \prod_{j \in \partial\blacksquare \setminus i} \left( m_{j\mu}^t \right)^2,$$

$$\frac{T_{i\mu}^{t+1}}{\Sigma_{i\mu}^{t+1}} \leftarrow \frac{m_{i\mu}^t}{\Sigma_{i\mu}^{t+1}} + \sum_{\blacksquare \in \partial i} \frac{\lambda}{\sqrt{M}} F_{\blacksquare\mu} g_{\blacksquare}^t \left( \prod_{k \in \partial\blacksquare \setminus i} m_{k\mu}^t \right.$$

$$\left. - \frac{\lambda}{\sqrt{M}} F_{\blacksquare\mu} g_{\blacksquare}^{t-1} m_{i\mu}^{t-1} \sum_{j \in \partial\blacksquare \setminus i} \left( v_{j\mu}^t - \left( m_{j\mu}^t \right)^2 \right) \prod_{k \in \partial\blacksquare \setminus i,j} m_{k\mu}^t m_{k\mu}^{t-1} \right).$$

(Alg2.3)

6: Update messages $m_{i\mu}^t$ and $v_{i\mu}^t$.

$$m_{i\mu}^{t+1} \leftarrow f_{\text{input}} \left( \Sigma_{i\mu}^{t+1}, T_{i\mu}^{t+1} \right), \qquad v_{i\mu}^{t+1} \leftarrow f_{\text{input,II}} \left( \Sigma_{i\mu}^{t+1}, T_{i\mu}^{t+1} \right).$$

(Alg2.4)

7: **until** $m_{i\mu}^t$ and $v_{i\mu}^t$ converge.

**Output:** Messages $m_{i\mu}$, $v_{i\mu}$ which are the average and variance of the variables.

---

## 4.4 Computational costs

In r-BP we have $NM \times c = \alpha NM^2$ messages and in Eq. (Alg1.3) we do summation over $c = \alpha M$ terms so that the computational cost of r-BP is order $O(NMc^2) = O(NM^3)$. On the other hand, in G-AMP, we have $NM$ messages and the computational cost is reduced to $O(NMc) = O(NM^2)$. In both cases, the computational cost does not depend on $p$, which is in contrast to the fully connected case.

## 4.5 Some prior distributions

### 4.5.1 Ising model

Let us consider the case that the ground truth data takes Ising values Eq. (11). In this case we find the input functions Eq. (156) as,

$$f_{\text{input}}(\Sigma, T) = \tanh\left(\frac{T}{\Sigma}\right), \qquad f_{\text{input,II}}(\Sigma, T) = 1. \tag{177}$$

### 4.5.2 Gaussian model

We also consider the case that data takes real values which obey Gaussian distribution Eq. (12). In this case we find the input functions as,

$$f_{\text{input}}(\Sigma, T) = \frac{T}{\Sigma+1}, \qquad f_{\text{input,II}}(\Sigma, T) = \frac{\Sigma}{\Sigma+1} + \frac{T^2}{(\Sigma+1)^2}. \tag{178}$$

## 4.6 Some likelihood functions

### 4.6.1 Additive Noise

Let us consider the case that the observed value $y$ contains additive noise $w$ to $\pi_*$ (see Eq. (13)),

$$y = \pi_* + w. \tag{179}$$

This means the likelihood function is

$$P_{\text{out}}(y|\pi) = \int dw W(w)\delta(y - (\pi + w)) = W(y - \pi), \tag{180}$$

with $W(w)$ being the distribution function of the noise. This yields the output functions Eq. (146)

$$g_{\text{out}}(\omega, y, V) = \frac{\int \frac{dz}{\sqrt{2\pi}} e^{-z^2/2}(-\frac{\partial}{\partial y}) W(y-\omega-\sqrt{V}z)}{\int \frac{dz}{\sqrt{2\pi}} e^{-z^2/2} W(\omega-y-\sqrt{V}z)}. \tag{181}$$

In particular, in the case of Gaussian noise

$$W(w) = \mathcal{N}(0, \Delta_*^2), \tag{182}$$

we find

$$g_{\text{out}}(\omega, y, V) = \frac{y-\omega}{V+\Delta_*^2}, \qquad -\frac{\partial}{\partial\omega} g_{\text{out}}(\omega, y, V) = \frac{1}{V+\Delta_*^2}. \tag{183}$$

### 4.6.2 Sign output

We also consider that the observed value $y$ is just the sign of $\pi_*$, (see Eq. (16))

$$y = \text{sgn}(\pi_*). \tag{184}$$

This means

$$P_{\text{out}}(y|\pi) = \delta(y - 1)\theta(\pi) + \delta(y + 1)\theta(-\pi). \tag{185}$$

This yields the output function Eq. (146)

$$g_{\text{out}}(\omega, y, V) = \sum_{\sigma=-1,1} \delta(y - \sigma)\tilde{g}_{\text{out}}(\omega, \sigma, V), \qquad \tilde{g}_{\text{out}}(\omega, \sigma, V) = \frac{-\sigma}{\sqrt{V}} \frac{H'(x)}{H(x)}\bigg|_{x=-\sigma\omega/\sqrt{V}}. \tag{186}$$

Here we introduced

$$H(x) = \int_x^\infty \frac{dt}{\sqrt{2\pi}} e^{-\frac{1}{2}t^2} = \frac{1}{2}\text{erfc}\left(\frac{x}{\sqrt{2}}\right), \tag{187}$$

with $\text{erfc}(x) = (2/\sqrt{\pi}) \int_x^\infty dy e^{-y^2}$ being the complementary error function.

## 4.7 State Evolution

In order to analyze the performance of the algorithms explained above at macroscopic scales $N, M \to \infty$ we now turn to analyze the so called state evolution (SE). The equations governing SE will be compared with the equations of states obtained by the replica approach.

Let us introduce the order parameters which characterize the macroscopic behavior of the algorithms,

$$m^t \quad = \frac{1}{NM} \sum_{i=1}^{N} \sum_{\mu=1}^{M} x_{*,i\mu} m_{i\mu}^t, \tag{188}$$

$$q^t \quad = \frac{1}{NM} \sum_{i=1}^{N} \sum_{\mu=1}^{M} \left( m_{i\mu}^t \right)^2, \tag{189}$$

$$Q^t \quad = \frac{1}{NM} \sum_{i=1}^{N} \sum_{\mu=1}^{M} v_{i\mu}^t, \tag{190}$$

where $m_{i\mu}^t$ and $v_{i\mu}^t$ are given by (see Eq. (163) ),

$$m_{i\mu}^t \quad = f_{\text{input}} \left( \Sigma_{i\mu}^t, T_{i\mu}^t \right), \tag{191}$$

$$v_{i\mu}^t \quad = f_{\text{input,II}} \left( \Sigma_{i\mu}^t, T_{i\mu}^t \right). \tag{192}$$

The individual microscopic variables $m_{i\mu}^t$ and $v_{i\mu}^t$ fluctuate depending on realizations of the quenched random variables $F_{\blacksquare\mu}$, $x_{*,i\mu}$ and $w_{\blacksquare}$ and the realization of the graph $G$. However we expect the macroscopic observables $m^t$, $q^t$ and $Q^t$ are self-averaging such that they become independent of the realizations of the quenched random variables in the thermodynamic limit $N, M \to \infty$. Then we can evaluate them as,

$$m^t = \mathbb{E}_y[m_{i\mu}^t] = \mathbb{E}_y[x_{*,i\mu} f_{\text{input}} \left( \Sigma_{i\mu}^t, T_{i\mu}^t \right)], \tag{193}$$

$$q^t = \mathbb{E}_y[(m_{i\mu}^t)^2] = \mathbb{E}_y[f_{\text{input}}^2 \left( \Sigma_{i\mu}^t, T_{i\mu}^t \right)], \tag{194}$$

$$Q^t = \mathbb{E}_y[v_{i\mu}^t] = \mathbb{E}_y[f_{\text{input,II}} \left( \Sigma_{i\mu}^t, T_{i\mu}^t \right)]. \tag{195}$$

To evaluate the order parameters we need $\Sigma_{i\mu}^t$ given by Eq. (160) and $T_{i\mu}^t$ given by Eq. (161). Both involve $g_{\text{out}} \left( \omega_{\blacksquare\to\mu}^t, y_{\blacksquare}, V_{\blacksquare\to\mu}^t \right)$. So let us first consider $V_{\blacksquare\to\mu}^t$ defined in Eq. (142). Since it is given by a summation over a large number of terms in $M \to \infty$ limit we can evaluate it as,

$$
\begin{aligned}
V_{\blacksquare\to\mu}^t \quad &= \sum_{\nu(\neq\mu)} \left( \frac{\lambda}{\sqrt{M}} F_{\blacksquare\nu} \right)^2 \left( \prod_{j\in\partial\blacksquare} v_{j\nu\to\blacksquare}^t - \prod_{j\in\partial\blacksquare} \left( m_{j\nu\to\blacksquare}^t \right)^2 \right) \\
&= M \mathbb{E}_F \left[ \left( \frac{\lambda}{\sqrt{M}} F_{\blacksquare\nu} \right)^2 \right] \left( \left( \mathbb{E}_y[v_{i\nu}^t] \right)^p - \left( \mathbb{E}_y \left[ (m_{i\nu}^t)^2 \right] \right)^p \right) + \mathcal{O}(M^{-1}) \\
&= \lambda^2 \left( (Q^t)^p - (q^t)^p \right) + \mathcal{O}(M^{-1}) \simeq V^t,
\end{aligned}
\tag{196}
$$

with

$$V^t = \lambda^2 \left( \left( Q^t \right)^p - \left( q^t \right)^p \right). \tag{197}$$

Here we assumed that, in the BP algorithms, the messages $\left\{ v_{j\nu\to\blacksquare}^t \right\}$ and $\left\{ m_{j\nu\to\blacksquare}^t \right\}$ are independent from each other and from $F$ and that messages are independent. We also used $\mathbb{E}_F \left[ F_{\blacksquare\nu}^2 \right] = 1$ (see Eq. (10)) where $\mathbb{E}_F$ denotes the average over the linear coefficients $F$. In the following we write the observation that the student receives as

$$y_{\blacksquare} = h_{\text{out}}(\pi_{\blacksquare}, w_{\blacksquare}), \tag{198}$$

where

$$h_{\text{out}}(\pi, w) = \pi + w, \tag{199}$$

in the case of additive noise (see Eq. (13)) and

$$h_{\text{out}}(\pi, w) = \text{sgn}(\pi), \tag{200}$$

in the case of sign output (see Eq. (16)). Using $V^t$ defined in Eq. (197) let us introduce some disorder-averaged quantities,

$$\hat{\chi}^t \;\; = \mathbb{E}_y\left[-\tfrac{\partial}{\partial \omega} g_{\text{out}}\left(\omega^t_{\blacksquare \to \mu}, h_{\text{out}}\left(\pi_{*,\blacksquare \to \mu}, w_\blacksquare\right), V^t\right)\right], \tag{201}$$

$$\hat{m}^t \;\; = \mathbb{E}_y\left[\tfrac{\partial}{\partial \pi_*} g_{\text{out}}\left(\omega^t_{\blacksquare \to \mu}, h_{\text{out}}\left(\pi_{*,\blacksquare \to \mu}, w_\blacksquare\right), V^t\right)\right], \tag{202}$$

$$\hat{q}^t \;\;\;\; = \mathbb{E}_y\left[g^2_{\text{out}}\left(\omega^t_{\blacksquare \to \mu}, h_{\text{out}}\left(\pi_{*,\blacksquare \to \mu}, w_\blacksquare\right), V^t\right)\right], \tag{203}$$

which become useful in the following.

Now let us turn to evaluate $\Sigma^{t+1}_{i\mu}$ given by Eq. (160). Similarly to $V^t_{\blacksquare \to \mu}$ discussed above, it is given by a summation over large number of variables in $c(= \alpha M) \to \infty$ limit so that it can be evaluated as,

$$\left(\Sigma^{t+1}_{i\mu}\right)^{-1} = -\sum_{\blacksquare \in \partial i} \left(\frac{\lambda}{\sqrt{M}} F_{\blacksquare \mu}\right)^2 \left(\prod_{j \in \partial \blacksquare \setminus i} (m_{j\mu \to \blacksquare})^2\right) \frac{\partial}{\partial \omega} g_{\text{out}}\left(\omega^t_{\blacksquare \to \mu}, h_{\text{out}}\left(\pi_{*,\blacksquare}, w_\blacksquare\right), V^t_{\blacksquare \to \mu}\right)$$

$$= \alpha M \mathbb{E}_F\left[\left(\frac{\lambda}{\sqrt{M}} F_{\blacksquare \mu}\right)^2\right] \left(\mathbb{E}_y\left[\left(m^t_{i\mu}\right)^2\right]\right)^{p-1}$$

$$\mathbb{E}_y\left[-\frac{\partial}{\partial \omega} g_{\text{out}}\left(\omega^t_{\blacksquare \to \mu}, h_{\text{out}}\left(\pi_{*,\blacksquare \to \mu}, w_\blacksquare\right), V^t\right)\right] + \mathcal{O}\left(M^{-1}\right)$$

$$\simeq \alpha \lambda^2 \left(q^t\right)^{p-1} \hat{\chi}^t. \tag{204}$$

We also used the fact that $F_{\blacksquare \mu}$, which is not independent from $\pi_{*,\blacksquare}$, is independent from

$$\pi_{*,\blacksquare \to \mu} = \pi_{*,\blacksquare} - \frac{\lambda}{\sqrt{M}} F_{\blacksquare \mu} \prod_{j \in \partial \blacksquare} x_{*,j\mu}. \tag{205}$$

Next let us examine $T^{t+1}_{i\mu}$ defined by Eq. (161). It can be evaluated in $M \to \infty$ limit

as,

$$
\begin{aligned}
\frac{T_{i\mu}^{t+1}}{\Sigma_{i\mu}^{t+1}} &= \sum_{\blacksquare\in\partial i}\left(\frac{\lambda}{\sqrt{M}}F_{\blacksquare\mu}\right)\left(\prod_{j\in\partial\blacksquare\backslash i}m_{j\mu\to\blacksquare}^t\right)g_{\mathrm{out}}\left(\omega_{\blacksquare\to\mu}^t,h_{\mathrm{out}}\left(\pi_{*,\blacksquare},w_\blacksquare\right),V_{\blacksquare\to\mu}^t\right) \\
&= \sum_{\blacksquare\in\partial i}\left(\frac{\lambda}{\sqrt{M}}F_{\blacksquare\mu}\right)\left(\prod_{j\in\partial\blacksquare\backslash i}m_{j\mu\to\blacksquare}^t\right)\Bigg\{g_{\mathrm{out}}\left(\omega_{\blacksquare\to\mu}^t,h_{\mathrm{out}}\left(\pi_{*,\blacksquare\to\mu},w_\blacksquare\right),V^t\right) \\
&\qquad\qquad + \left(\frac{\lambda}{\sqrt{M}}F_{\blacksquare\mu}\right)\left(\prod_{j\in\partial\blacksquare}x_{*,j\mu}\right)\frac{\partial}{\partial\pi_*}g_{\mathrm{out}}\left(\omega_{\blacksquare\to\mu}^t,h_{\mathrm{out}}\left(\pi_{*,\blacksquare\to\mu},w_\blacksquare\right),V^t\right)\Bigg\} \\
&\quad + \mathcal{O}\left(M^{-1}\right) \\
&= \sum_{\blacksquare\in\partial i}\left(\frac{\lambda}{\sqrt{M}}F_{\blacksquare\mu}\right)\left(\prod_{j\in\partial\blacksquare\backslash i}m_{j\mu\to\blacksquare}^t\right)g_{\mathrm{out}}\left(\omega_{\blacksquare\to\mu}^t,h_{\mathrm{out}}\left(\pi_{*,\blacksquare\to\mu},w_\blacksquare\right),V^t\right) \\
&\quad + \sum_{\blacksquare\in\partial i}\left(\frac{\lambda}{\sqrt{M}}F_{\blacksquare\mu}\right)^2\left(\prod_{j\in\partial\blacksquare\backslash i}m_{j\mu\to\blacksquare}^t x_{*,j\mu}\right)x_{*,i\mu}\frac{\partial}{\partial\pi_*}g_{\mathrm{out}}\left(\omega_{\blacksquare\to\mu}^t,h_{\mathrm{out}}\left(\pi_{*,\blacksquare\to\mu},w_\blacksquare\right),V^t\right) + \mathcal{O}\left(M^{-1}\right) \\
&\simeq \sqrt{\alpha\lambda^2(q^t)^{p-1}\hat{q}^t}\,\mathcal{N}(0,1) + \alpha\lambda^2\left(m^t\right)^{p-1}\hat{m}^t x_{*,i\mu}.
\end{aligned} \tag{206}
$$

The 1st term in the last equation on righthand side (rhs) is due to the 1st term in the 3rd equation on rhs, which is a summation over large number $c(=\alpha M)\gg 1$ of statistically independent variables. Here we are using again the fact that $F_{\blacksquare\mu}$ is independent from $\pi_{*,\blacksquare\to\mu}$. Then it can be considered as a Gaussian variable due to the central limit theorem. Its mean is evaluated as,

$$
\mathbb{E}_F\left[\sum_{\blacksquare\in\partial i}\left(\frac{\lambda}{\sqrt{M}}F_{\blacksquare\mu}\right)\right]\mathbb{E}_y\left[\left(\prod_{j\in\partial\blacksquare\backslash i}m_{j\mu\to\blacksquare}^t\right)\right]\mathbb{E}_y\left[g_{\mathrm{out}}\left(\omega_{\blacksquare\to\mu}^t,h_{\mathrm{out}}\left(\pi_{*,\blacksquare\to\mu},w_\blacksquare\right),V^t\right)\right] = 0 \tag{207}
$$

This holds because of the reflection symmetry of the prior distribution $P_{\mathrm{pri.}}(x)=P_{\mathrm{pri.}}(-x)$ (see sec. 2.2) which ensures $\mathbb{E}_y\left[\left(\prod_{j\in\partial\blacksquare\backslash i}m_{j\mu\to\blacksquare}^t\right)\right]=0$ for $p>1$. For the $p=1$ case (linear estimation), which we do not consider in the present paper, the similar mean vanishment happens thanks to the random spreading code yielding $\mathbb{E}_F[F_{\blacksquare,\mu}]=0$ in some earlier contexts such as CDMA and compressed sensing. On the other hand, similarly to $V_{\blacksquare\to\mu}^t$ (see Eq. (196)) and $\Sigma_{i\mu}^{t+1}$ (see Eq. (204)), the variance is evaluated as

$$
\mathbb{E}_y\left[\left(\left(\sum_{\blacksquare\in\partial i}\left(\frac{\lambda}{\sqrt{M}}F_{\blacksquare\mu}\right)\left(\prod_{j\in\partial\blacksquare\backslash i}m_{j\mu\to\blacksquare}^t\right)g_{\mathrm{out}}\left(\omega_{\blacksquare\to\mu}^t,h_{\mathrm{out}}\left(\pi_{*,\blacksquare\to\mu},w_\blacksquare\right),V^t\right)\right)\right)^2\right]
$$
$$
\simeq \alpha\lambda^2\left(q^t\right)^{p-1}\hat{q}^t. \tag{208}
$$

The 2nd term in the last equation on rhs of Eq. (206) is derived from the 2nd term in the 3rd equation on rhs similarly.

To sum up, we now have expressions of the two quantities $\left(\Sigma_{i\mu}^t\right)^{-1}$ Eq. (204) and $\frac{T_{i\mu}^t}{\Sigma_{i\mu}^t}$ Eq. (206) needed to evaluate the order parameters $m^t$, $q^t$, $Q^t$ as given by Eq. (193)-Eq. (195) expressed in terms of $m^t$, $q^t$, $Q^t$ and $\hat{\chi}^t$, $\hat{m}^t$ and $\hat{q}^t$ given by Eq. (201)-Eq. (203).

Now we have to examine the average over the quenched randomness $\mathbb{E}_y$ which appear in these equations. The average $\mathbb{E}_y$ over $F_{\blacksquare\mu}$, $x_{*,i\mu}$ and $w_{\blacksquare}$ can be regarded as the average over $\omega^t_{\blacksquare\to\mu}$, $\pi_{*,\blacksquare\to\mu}$ and $w_{\blacksquare}$. They are obtained as

$$\mathbb{E}_y\left[\omega^t_{\blacksquare\to\mu}\right] \qquad = \mathbb{E}_{x_*}\left[\pi_{*,\blacksquare}\right] = 0, \tag{209}$$

$$\mathbb{E}_y\left[\left(\omega^t_{\blacksquare\to\mu}\right)^2\right] \quad = \lambda^2\left(q^t\right)^p + \mathcal{O}\left(M^{-1}\right), \tag{210}$$

$$\mathbb{E}_y\left[\omega^t_{\blacksquare\to\mu}\pi_{*,\blacksquare}\right] \quad = \lambda^2\left(m^t\right)^p + \mathcal{O}\left(M^{-1}\right), \tag{211}$$

$$\mathbb{E}_y\left[\pi^2_{*\blacksquare}\right] \quad = \lambda^2\left(\mathbb{E}_{x_*}\left[x^2_{*,i\mu}\right]\right)^p = 1. \tag{212}$$

Introducing the bivariate normal distribution function $\mathcal{N}[x_1, x_2; C]$ of variables $x_1$ and $x_2$ with 0 mean and covariance matrix $C = (C^t)$,

$$\mathcal{N}[x_1, x_2; C] = \frac{1}{2\pi\sqrt{\det C}} \exp\left[-\frac{1}{2}((C^{-1})_{11}x_1^2 + 2(C^{-1})_{12}x_1x_2 + (C^{-1})_{22}x_2^2)\right], \tag{213}$$

with

$$C = \begin{pmatrix} C_{11} & C_{12} \\ C_{21} & C_{22} \end{pmatrix} = \lambda^2 \begin{pmatrix} (q^t)^p & (m^t)^p \\ (m^t)^p & 1 \end{pmatrix}, \qquad \det C = \lambda^4\left((q^t)^p - (m^t)^{2p}\right). \tag{214}$$

Eq. (201)-(203) can be expressed as

$$\hat{\chi}^t = \int \mathrm{d}w\, W(w) \int \mathrm{d}\xi\, \mathrm{d}z\, \mathcal{N}[\xi, z; C] \left(-\tfrac{\partial}{\partial\xi}g_{\mathrm{out}}\left(\xi, h_{\mathrm{out}}(z, w), V^t\right)\right), \tag{215}$$

$$\hat{m}^t = \int \mathrm{d}w\, W(w) \int \mathrm{d}\xi\, \mathrm{d}z\, \mathcal{N}[\xi, z; C] \ \tfrac{\partial}{\partial z}g_{\mathrm{out}}\left(\xi, h_{\mathrm{out}}(z, w), V^t\right), \tag{216}$$

$$\hat{q}^t = \int \mathrm{d}w\, W(w) \int \mathrm{d}\xi\, \mathrm{d}z\, \mathcal{N}[\xi, z; C] \ g^2_{\mathrm{out}}\left(\xi, h_{\mathrm{out}}(z, w), V^t\right). \tag{217}$$

Finally we find the order parameters (188)-(190) satisfy the following self-consistent equations,

$$m^{t+1} = \int \mathrm{d}x_*\, P_{\mathrm{pri.}}(x_*) \int \mathcal{D}z\, x_* f_{\mathrm{input}}\left(\frac{1}{\alpha\lambda^2(q^t)^{p-1}\hat{\chi}^t}, \frac{\alpha\lambda^2(m^t)^{p-1}\hat{m}^t x_* + z\sqrt{\alpha\lambda^2(q^t)^{p-1}\hat{q}^t}}{\alpha\lambda^2(q^t)^{p-1}\hat{\chi}^t}\right) \tag{218}$$

$$q^{t+1} = \int \mathrm{d}x_*\, P_{\mathrm{pri.}}(x_*) \int \mathcal{D}z\, f^2_{\mathrm{input}}\left(\frac{1}{\alpha\lambda^2(q^t)^{p-1}\hat{\chi}^t}, \frac{\alpha\lambda^2(m^t)^{p-1}\hat{m}^t x_* + z\sqrt{\alpha\lambda^2(q^t)^{p-1}\hat{q}^t}}{\alpha\lambda^2(q^t)^{p-1}\hat{\chi}^t}\right), \tag{219}$$

$$Q^{t+1} = \int \mathrm{d}x_*\, P_{\mathrm{pri.}}(x_*) \int \mathcal{D}z\, f_{\mathrm{input,II}}\left(\frac{1}{\alpha\lambda^2(q^t)^{p-1}\hat{\chi}^t}, \frac{\alpha\lambda^2(m^t)^{p-1}\hat{m}^t x_* + z\sqrt{\alpha\lambda^2(q^t)^{p-1}\hat{q}^t}}{\alpha\lambda^2(q^t)^{p-1}\hat{\chi}^t}\right) \tag{220}$$

These are the SE equations for the current system. The corresponding algorithm is given in algorithm 3.

---

**Algorithm 3** SE algorithm

---

**Input:** input function $f_{\text{input}}$, output function $g_{\text{out}}$, maximum time step $t_{\max}$.

1: **initialization:**    Set initial values of $m^t$, $q^t$ and $Q^t$.

2: **repeat**

3:     Update $V^t$ and the matrix $C$

$$V^t \leftarrow \lambda^2 \left( \left( Q^t \right)^p - \left( q^t \right)^p \right) \qquad C \leftarrow \lambda^2 \begin{pmatrix} (q^t)^p & (m^t)^p \\ (m^t)^p & 1 \end{pmatrix} \tag{Alg3.1}$$

4:     Update $\hat{\chi}^t$, $\hat{m}^t$ and $\hat{q}^t$

$$\begin{aligned}
\hat{\chi}^t &\leftarrow \int \mathrm{d}w\, W(w) \int \mathrm{d}\xi\, \mathrm{d}z\, \mathcal{N}[\xi, z; C] \left( -\frac{\partial}{\partial \xi} g_{\text{out}} \left( \xi, h_{\text{out}}(z, w), V^t \right) \right), \\
\hat{m}^t &\leftarrow \int \mathrm{d}w\, W(w) \int \mathrm{d}\xi\, \mathrm{d}z\, \mathcal{N}[\xi, z; C]\; \frac{\partial}{\partial z} g_{\text{out}} \left( \xi, h_{\text{out}}(z, w), V^t \right), \\
\hat{q}^t &\leftarrow \int \mathrm{d}w\, W(w) \int \mathrm{d}\xi\, \mathrm{d}z\, \mathcal{N}[\xi, z; C]\; g_{\text{out}}^2 \left( \xi, h_{\text{out}}(z, w), V^t \right)
\end{aligned} \tag{Alg3.2}$$

5:     Update $m^t$, $q^t$ and $Q^t$

$$\begin{aligned}
m^{t+1} &\leftarrow \int \mathrm{d}x_*\, P_{\text{pri.}}(x_*) \int \mathcal{D}z\; x_* f_{\text{input}} \left( \frac{1}{\alpha\lambda^2 (q^t)^{p-1}\hat{\chi}^t}, \frac{\alpha\lambda^2 (m^t)^{p-1}\hat{m}^t x_* + z\sqrt{\alpha\lambda^2(q^t)^{p-1}\hat{q}^t}}{\alpha\lambda^2 (q^t)^{p-1}\hat{\chi}^t} \right), \\
q^{t+1} &\leftarrow \int \mathrm{d}x_*\, P_{\text{pri.}}(x_*) \int \mathcal{D}z\; f_{\text{input}}^2 \left( \frac{1}{\alpha\lambda^2 (q^t)^{p-1}\hat{\chi}^t}, \frac{\alpha\lambda^2 (m^t)^{p-1}\hat{m}^t x_* + z\sqrt{\alpha\lambda^2(q^t)^{p-1}\hat{q}^t}}{\alpha\lambda^2 (q^t)^{p-1}\hat{\chi}^t} \right), \\
Q^{t+1} &\leftarrow \int \mathrm{d}x_*\, P_{\text{pri.}}(x_*) \int \mathcal{D}z\; f_{\text{input,II}} \left( \frac{1}{\alpha\lambda^2 (q^t)^{p-1}\hat{\chi}^t}, \frac{\alpha\lambda^2 (m^t)^{p-1}\hat{m}^t x_* + z\sqrt{\alpha\lambda^2(q^t)^{p-1}\hat{q}^t}}{\alpha\lambda^2 (q^t)^{p-1}\hat{\chi}^t} \right).
\end{aligned} \tag{Alg3.3}$$

6: **until** $m^t$, $q^t$ and $Q^t$ converge or $t$ reaches the maximum step $t_{\max}$.

**Output:** Time sequences of the order parameters $\{(m^t, q^t, Q^t)\}_t$.

---

### 4.7.1   Ising model

In the case of the Ising input (see sec. 4.5.1), we find using Eq. (177) in Eq. (218)-Eq. (220),

$$\begin{aligned}
m^{t+1} &= \int \mathcal{D}z\; \tanh \left( \alpha\lambda^2 \left( m^t \right)^{p-1} \hat{m}^t + z\sqrt{\alpha\lambda^2 (q^t)^{p-1}\hat{q}^t} \right), & (221) \\
q^{t+1} &= \int \mathcal{D}z\; \tanh^2 \left( \alpha\lambda^2 \left( m^t \right)^{p-1} \hat{m}^t + z\sqrt{\alpha\lambda^2 (q^t)^{p-1}\hat{q}^t} \right), & (222) \\
Q^{t+1} &= 1. & (223)
\end{aligned}$$

In Bayes optimal case we expect $m^t = q^t$ holds at the fixed point. Later we will find that $\hat{m}^t = \hat{q}^t$ holds if $m^t = q^t$ and $Q^t = 1$. Then one can show that $m^{t+1} = q^{t+1}$ holds using Eq. (221), Eq. (222) and Lemma A.

### 4.7.2 Gaussian model

In the case of the Gaussian input (see sec. 4.5.2), we find using Eq. (178) in Eq. (218)-Eq. (220),

$$m^{t+1} = \int dx_* \, P_{\text{pri.}}(x_*) \int \mathcal{D}z \, x_* \frac{\alpha\lambda^2(m^t)^{p-1}\hat{m}^t x_* + z\sqrt{\alpha\lambda^2(q^t)^{p-1}\hat{q}^t}}{1+\alpha\lambda^2(q^t)^{p-1}\hat{\chi}^t} = \frac{\alpha\lambda^2(m^t)^{p-1}\hat{m}^t}{1+\alpha\lambda^2(q^t)^{p-1}\hat{\chi}^t}, \quad (224)$$

$$q^{t+1} = \int dx_* \, P_{\text{pri.}}(x_*) \int \mathcal{D}z \left[ \left( \frac{\alpha\lambda^2(m^t)^{p-1}\hat{m}^t}{1+\alpha\lambda^2(q^t)^{p-1}\hat{\chi}^t} \right)^2 (x_*)^2 + z^2 \frac{\alpha\lambda^2(q^t)^{p-1}\hat{q}^t}{(1+\alpha\lambda^2(q^t)^{p-1}\hat{\chi}^t)^2} + 2zx_* \cdots \right]$$

$$= (m^{t+1})^2 + \frac{\alpha\lambda^2(q^t)^{p-1}\hat{q}^t}{(1+\alpha\lambda^2(q^t)^{p-1}\hat{\chi}^t)^2}, \quad (225)$$

$$Q^{t+1} = q^{t+1} + \frac{1}{1+\alpha\lambda^2(q^t)^{p-1}\hat{\chi}^t}. \quad (226)$$

Assuming the spin normalization $Q^{t+1} = 1$, the last equation implies,

$$\frac{1}{1-q^{t+1}} = 1 + \alpha\lambda^2(q^t)^{p-1}\hat{\chi}^t. \quad (227)$$

Using this we find

$$\frac{m^{t+1}}{1-q^{t+1}} = \alpha\lambda^2(m^t)^{p-1}\hat{m}^t, \quad (228)$$

$$\frac{q^{t+1}-(m^{t+1})^2}{(1-q^{t+1})^2} = \alpha\lambda^2(q^t)^{p-1}\hat{q}^t. \quad (229)$$

In Bayes optimal case we expect $m^t = q^t$ holds at the fixed point. Later we will find that $\hat{\chi}^t = \hat{m}^t = \hat{q}^t$ holds if $m^t = q^t$ and $Q^t = 1$ hold. Then given these we find that $q^{t+1} = m^{t+1}$ holds using Eq. (228) and Eq. (229). We also find that $Q^{t+1} = 1$ holds if $\hat{\chi}^t = \hat{m}^t = \hat{q}^t$ using Eq. (224)-Eq. (226).

### 4.7.3 Additive noise

In the case of additive noise (see Eq. (13)) we have $h_{\text{out}}(\pi, w) = \pi + w$ as given by Eq. (199). In sec 4.6.1 we obtained Eq. (181)) which implies,

$$g_{\text{out}}(\xi, h_{\text{out}}(z, w), V) = g_{\text{out}}(w + (z - \xi), V). \quad (230)$$

The fact that the output function depends on $z$ and $\xi$ just through the difference $z - \xi$ leads to a simplification. For a generic function $f(z)$, we find the average weighted by the bivariate normal distribution Eq. (213) becomes

$$\int d\xi dz \mathcal{N}[\xi, z; C] f(z - \xi) = \int \mathcal{D}z_0 f(\sqrt{C_{11} + C_{22} - 2C_{12}} z_0). \quad (231)$$

Using this and the explicit form of the output function given by Eq. (181)) we find Eq. (215)-Eq. (217) become

$$\hat{\chi}^t = \hat{m}^t = \int dw \, W(w) \int \mathcal{D}z_0 \frac{\partial}{\partial w} g_{\text{out}}\left( w - \lambda\sqrt{(q^t)^p + 1 - 2(m^t)^p} z_0, V^t \right)$$

$$= \theta_0 - \theta_1, \quad (232)$$

$$\hat{q}^t = \int dw \, W(w) \int \mathcal{D}z_0 g_{\text{out}}^2\left( w - \lambda\sqrt{(q^t)^p + 1 - 2(m^t)^p} z_0, V^t \right)$$

$$= \theta_0, \quad (233)$$

where we introduced

$$\theta_0 \quad = \int dw\, W(w) \int \mathcal{D}z_0 \left( \frac{\int \mathcal{D}z_1 \frac{\partial}{\partial \omega} W(\Xi)}{\int \mathcal{D}z_1 W(\Xi)} \right)^2, \tag{234}$$

$$\theta_1 \quad = \int dw\, W(w) \int \mathcal{D}z_0 \frac{\int \mathcal{D}z_1 \frac{\partial^2}{\partial \omega^2} \frac{\partial}{\partial \omega} W(\Xi)}{\int \mathcal{D}z_1 W(\Xi)}, \tag{235}$$

with

$$\Xi = w - \lambda \sqrt{(q^t)^p + 1 - 2(m^t)^p]} z_0 - \sqrt{V^t} z_1. \tag{236}$$

In the Bayes optimal case we expect $m^t = q^t$ holds at the fixed point. Let us also assume $Q^t = 1$ holds. Then we find (see Eq. (197)) $\Xi = w - \lambda \sqrt{1 - (q^t)^p}(z_0 + z_1)$. Then it is possible to show that $\theta_1 = 0$ (Lemma B). This in turn implies $\hat{\chi}^t = \hat{q}^t = \hat{m}^t$.

### 4.7.4  Sign output

In the case of sign output (see Eq. (16)) we have $h_{\rm out}(\pi, w) = {\rm sgn}(\pi)$ as given by Eq. (200). The explicit form of the output function is given by Eq. (186) which reads,

$$g_{\rm out}(\omega, y, V) = \sum_{\sigma=-1,1} \delta(y - \sigma) \tilde{g}_{\rm out}(\xi, \sigma, V), \qquad \tilde{g}_{\rm out}(\xi, \sigma, V) = \frac{-\sigma}{\sqrt{V}} \frac{H'(x)}{H(x)}\bigg|_{x=-\sigma\xi/\sqrt{V}}, \tag{237}$$

where we find a useful relation $\tilde{g}_{\rm out}(\xi, -1, V) = -\tilde{g}_{\rm out}(-\xi, +1, V)$. Using these we find $\hat{\chi}^t$, $\hat{m}^t$ and $\hat{q}^t$ become the following.

$$\hat{\chi}^t \quad = \quad 2 \int d\xi \frac{e^{-\frac{\xi^2}{2C_{11}}}}{\sqrt{2\pi C_{11}}} H(-\gamma\xi) \left[ \frac{\xi}{V} \tilde{g}_{\rm out}(\xi, +1, V^t) + \tilde{g}_{\rm out}^2(\xi, +1, V^t) \right], \tag{238}$$

$$\hat{m}^t \quad = \quad 2 \int d\xi \frac{e^{-\frac{(C^{-1})_{11}\xi^2}{2}}}{2\pi\sqrt{\det C}} \tilde{g}_{\rm out}(\xi, +1, \sigma, V), \tag{239}$$

$$\hat{q}^t \quad = \quad 2 \int d\xi \frac{e^{-\frac{\xi^2}{2C_{11}}}}{\sqrt{2\pi C_{11}}} H(-\gamma\xi) \tilde{g}_{\rm out}^2(\xi, +1, \sigma, V), \tag{240}$$

with

$$\gamma = \frac{C_{12}}{\sqrt{\det C}} \frac{1}{\sqrt{C_{11}}} = \frac{1}{\left\{ \lambda^2 \left[ \left( \frac{q^t}{m^t} \right)^{2p} - (q^t)^p \right] \right\}^{1/2}}, \tag{241}$$

for the covariance matrix $C$ (see Eq. (214)). Since we assume the Bayes optimal setting we expect $q^t = m^t$ holds at the fixed point. Assuming also $Q^t = 1$ which must hold due to the normalization of spins we find (see Eq. (197)),

$$\gamma = \frac{1}{\sqrt{\lambda^2(1 - q^t)^p}} = \frac{1}{\sqrt{V^t}}, \tag{242}$$

which implies

$$\hat{\chi}^t = \hat{q}^t = \hat{m}^t = 2 \int d\xi \frac{e^{-\frac{\xi^2}{2\lambda^2(q^t)^p}}}{\sqrt{2\pi\lambda^2(q^t)^p}} \frac{1}{2\pi V^t} \frac{e^{-\frac{\xi^2}{V^t}}}{H\left(-\frac{\xi}{\sqrt{V^t}}\right)}. \tag{243}$$

## 4.8 Comparison with the replica theory

Here let us compare the SE equations with the equation of states obtained by the replica approach. Schematically the SE equations look like $A_{t+1} = F_A(A_t, B_t, ...)$, $B_{t+1} = F_B(A_t, B_t, ...)$, ... while the equations of states by the replica approach look like $A = \tilde{F}_A(A, B, ...)$, $B = \tilde{F}_B(A, B, ...)$, ..... The two approaches agree if $F_A = \tilde{F}_A$, $F_B = \tilde{F}_B$, .....

For clarity let us focus on the explicit expressions for the three specific cases. In the case of Ising prior and additive noise, we find that the SE equations using Eq. (221) and Eq. (222) combined with Eq. (232)-Eq. (236) agree with the two equations of states obtained by the replica approach given by Eq. (101). Similarly in the case of Gaussian prior and additive noise, the SE equations (Eq. (228) and Eq. (229) combined with Eq. (233)-Eq. (236)) again agree with the replica result given by Eq. (107). For the case of Gaussian prior and sign output in the Bayes optimal setting, the SE equation (Eq. (228) with Eq. (243)) matches with the replica result Eq. (112). In all the cases, the equivalence between the SE and replica results is confirmed.

## 4.9 Discussions

In the replica approach we have shown that loop corrections vanish in the dense limit $\lim_{c\to\infty} \lim_{N\to\infty}$. Unfortunately such considerations are absent in the message passing approach discussed in this section. We believe that the arguments used to derive the algorithms (r-BP and G-AMP) are valid only in the dense limit $\lim_{c\to\infty} \lim_{N\to\infty}$. Indeed the analysis of TAP equation through PGY expansion [30] on the $p = 2$ system (matrix factorization) have shown that loop corrections are inevitable in the globally coupled system.

In the results presented above one can notice that expressions for the macroscopic quantities like the order-parameters, equation of states are the same for the deterministic model with $F = 1$ Eq. (9) and the disordered model with random $F$ Eq. (10). Let us recall that we found the same in the replica approach discussed in sec 3. However we found that convergence of the algorithms (r-BP and G-AMP) can be very different between the two models in some specific cases as we explain in sec 5.

# 5 Analysis of specific models

Now let us examine some representative cases closely. In the following we limit ourselves to the Bayes optimal setting for which $m = q$ and $Q_0 = Q = 1$ holds. We discuss the cases of Ising prior and additive Gaussian noise in sec. 5.1, Gaussian prior and additive Gaussian noise in sec. 5.2, Gaussian prior and sign output in sec. 5.3. In each of the cases, we first examine the equation of states for the order parameter $m$ (obtained by the replica and message passing approaches which agree) and establish phase diagrams. Then we examine the performance of message passing algorithms. For the details on some numerical prescriptions for the message passing algorithms see appendix D. Some detailed comparisons between different algorithms are presented in appendix E.

## 5.1 Ising prior and additive Gaussian noise

Here we examine the case of the Bayes optimal additive Gaussian noise. The corresponding equation of state and free energy in the replica approach are given by Eq. (103) and Eq. (104). For the message passing approach, we will examine the r-BP algorithm (Algorithm 1) and G-AMP algorithm (Algorithm 2) with the input function Eq. (177) and output function Eq. (183). The SE equation can be found as Eq. (221) and Eq. (222) combined with Eq. (232)-Eq. (236), showing a consistency with the replica formula.

### 5.1.1 $p = 2$ case

Let us first examine the case $p = 2$. By solving the equation of state Eq. (103) numerically we obtained results as shown in Fig. 3. Note that $m = 0$ is always a solution. At large enough $\alpha$ we find another solution $m > 0$ whose magnitude grows continuously increasing $\lambda$ passing a critical point $\lambda^*(\alpha) = \frac{1}{\sqrt{\alpha-1}}$ (black line in the right panel of Fig 3), which coincides with the stability condition of the paramagnetic solution computed in appekdix C.1. Here one observes a continuous transition by increasing $\lambda$ between a state with $m = 0$ and a magnetized state with $m > 0$. At small enough $\alpha$, the transition in $\lambda$ becomes first-order (with the paramagnetic state remaining stable up to $\lambda = \infty$ for $\alpha < 1$). For intermediate values of $\alpha$, the phase behavior is more complex: the system exhibits first a continuous transition from paramagnet to a non-trivial low-overlap state, followed by a first-order transition to a higher-overlap state.

The consequences of this phase diagram on the performance of inference are as follows (see the right panel Fig 3). In Region I inference is hard due to the stability of the paramagnetic solution, in the sense that G-AMP, and presumably any polynomial time algorithms, will not be able to find a solution correlated with the signal when starting from uninformative (low overlap) initialization. Regions II and V can also be considered hard, but in a slightly more complicated way. In fact, it is easy to find a solution correlated with the signal more than a random guess, since the paramagnetic phase loses its stability. However, this solution is not optimal, since it corresponds to the low-overlap branch, while the high-overlap branch remains inaccessible in polynomial time. This means that the perfect reconstruction of the signal is impossible even in the $\lambda \to \infty$ limit when starting from the uninformative initialization. By solving the equation of state (Eq. (103)) for $\lambda \to \infty$, we obtained that the green line separating Regions II and III asymptotically approaches $\alpha \approx 1.30$. This value defines the easy-to-hard threshold $\alpha_P$: in the whole region $\alpha < \alpha_P$ the perfect reconstruction of the signal is not possible even in the $\lambda \to \infty$ limit (noiseless limit), see also the last panel of Fig. 4. Conversely, in regions III and IV inference becomes easy, as there is a unique solution which has a large overlap with the signal, which converges to the perfect reconstruction solution ($m = 1$) in the limit

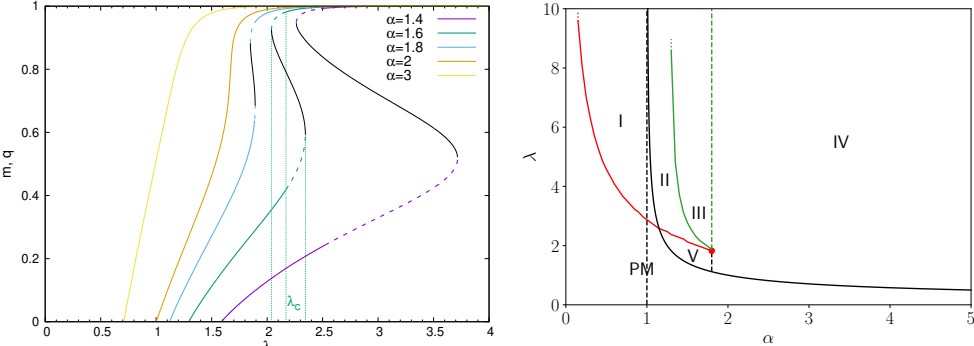

Figure 3: Left: Order parameter $m = q$ in the Bayes optimal case for Ising prior, additive Gaussian noise and $p = 2$. Dashed lines indicates a metastable magnetized state associated to a first-order transition. For $\alpha = 1.6$ three vertical lines are shown: the central one $\lambda_c$ is the thermodynamic first-order transition where the difference in free energy between the two branches of solutions changes sign; the leftmost one is the spinodal point $\lambda_d(\alpha)$ for the high-magnetization solution (corresponding to the red line in the right panel); the rightmost one is the spinodal point for the low-magnetization solution (green line in the right panel). Right: phase diagram in the $\alpha - \lambda$ plane. The black solid line is the spinodal line (stability limit) of the paramagnetic (PM) solution $\lambda^*(\alpha)$; for $\alpha < 1$ the PM solution is stable $\forall \lambda$. Region I: coexistence of PM and a high-magnetization solution, separated by a first-order transition (not shown for clarity). Region II: coexistence of low and high-magnetization phases, separated by a first-order transition (not shown). Region III: only the high-magnetization phase survives beyond the spinodal point for the low-magnetization solution. Regions IV and V: as in region III, there is a unique magnetized phase, but decreasing $\lambda$ it transitions continuously to the PM phase.

$\lambda \to \infty$. Interestingly, another threshold $\alpha_s$ called possible-to-impossible threshold, which represents the boundary for the existence of the perfect recovery solution ($m = 1$), is here given by $\alpha_s = 0$ since the $m = 1$ solution always exists in the $\lambda \to \infty$ limit. This is very in contrast to the Gaussian prior case as shown below.

Next let us examine the performance of message passing algorithms. In Fig. 5 we show time evolution of the overlaps $m^t$ and $q^t$ computed using G-AMP. For the cases $\alpha = 1.6$ and $\alpha = 1.2$, uninformative initialization is used (see sec. D.1) so that the initial configuration is chosen to be very far from the teacher's configuration. The results are compared with the numerical solution of the SE equation. On the left plot, the case $\alpha = 1.6$ is considered: for large enough values of $\lambda$ the system falls in Region III in which inference is easy. On the right panel, the case $\alpha = 1.2$ is considered. This case falls in Region II, that is a hard phase for which the perfect reconstruction is not possible, when starting from an uninformative initialization. In the third panel, the case $\alpha = 0.9 < 1$ is considered with the informative initial condition (the initialization parameter in sec. D.1 is set to be $a = 0.9$): this is for examining the predicted possible-to-impossible threshold $\alpha_s = 0$ in the limit $\lambda \to \infty$, and actually both the SE and GAMP results converge to the $m \approx 1$ solution when $\lambda$ is large ($\lambda = 10$ is examined). For comparison, the case $\lambda = 4$ is also examined, exhibiting the convergence to the paramagnetic solution.

The results of G-AMP compare well with the solution of the SE equation. However, for small values of the overlap, it can be seen that the algorithm slows down. Discrepancies in this case may be attributed to finite $M$ corrections. See also Fig. 21 in the appendix for a similar case. The convergence parameter $D_t$ is defined in Eq. (281) in appendix E. Finally, in Fig. 18 in appendix E, we also show results of r-BP algorithm compared

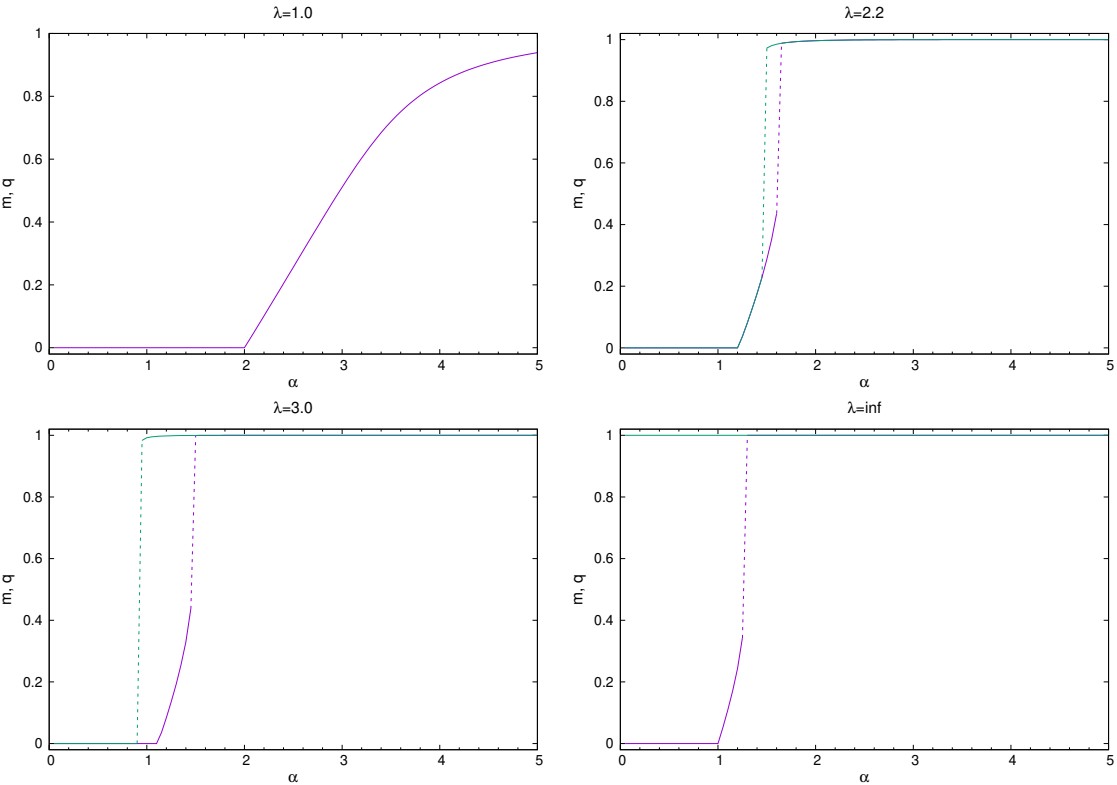

Figure 4: Order parameter $m = q$ in the Bayes optimal case for Ising prior, additive Gaussian noise and $p = 2$. The order parameter is presented as a function of $\alpha$ for different values of $\lambda$. These figures present the same information as in Fig. 3, but shown in a different, complementary way. The first panel shows a unique branch of solutions, transitioning continuously from the paramagnetic one $m = 0$. Increasing $\lambda$, in the second and third panels, also a high overlap branch (shown in green) appears discontinuously. Last panel shows the behaviour of the order parameter in the $\lambda \to \infty$ limit. The vertical dashed line coincides with the easy-to-hard threshold $\alpha_P$ for perfect reconstruction in the $\lambda \to \infty$ limit, while the boundary for the existence of the perfect recovery solution is given by $\alpha_s = 0$.

with G-AMP. For smaller $N$ r-BP is more accurate, but increasing $N$ G-AMP drastically improves.

### 5.1.2  $p = 3$ case

In this case, a discontinuous transition between the paramagnetic $m = 0$ and the magnetized phase $m > 0$ is universally found. We expect this is the case for $p > 2$. In Fig. 6 we show the phase diagram obtained by analyzing the equation of states. We define $\lambda_d(\alpha)$ the spinodal point of the non-trivial solution, that coexists with the trivial paramagnetic one since the transition is of first-order type. The true thermodynamic transition point $\lambda_c(\alpha)$ is defined as the point where the free energy of the highly magnetized state becomes equal to the one of the paramagnetic state.

The magnetization becomes very large in the magnetized phase meaning that the quality of the inference is good. However, unlike some sparse systems with $c = O(1)$ [14], the paramagnetic state $m = 0$ is always stable (see sec. C.1 ) in the dense limit $c \gg 1$

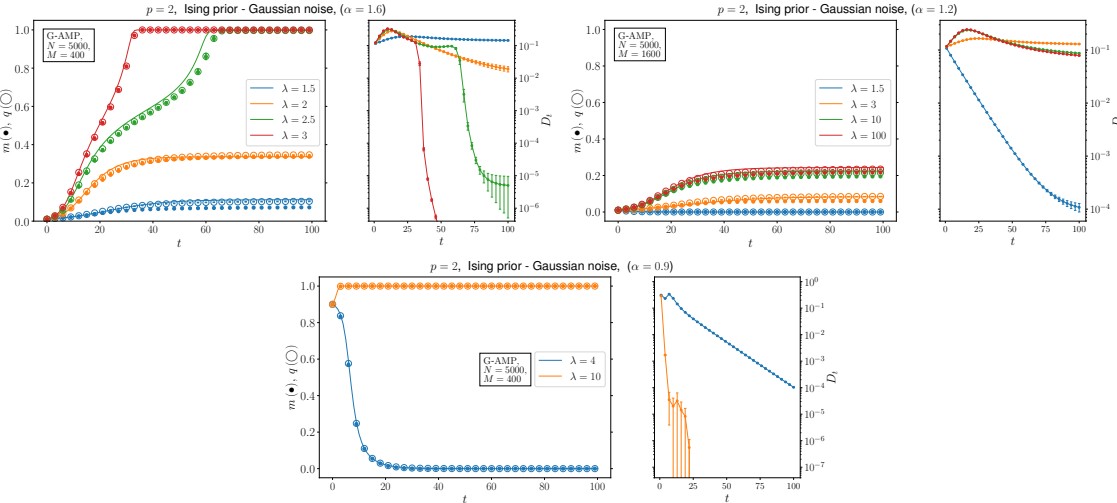

Figure 5: Evolution of the order parameters in the G-AMP algorithm for $\alpha = 1.6$, $\alpha = 1.2$ and $\alpha = 0.9$ on random instances of the problem. The horizontal axis $t$ is for the time step. Data is averaged over 5 instances. Solid lines represent the State Evolution predictions. On the smaller panels, the convergence parameter is plotted.

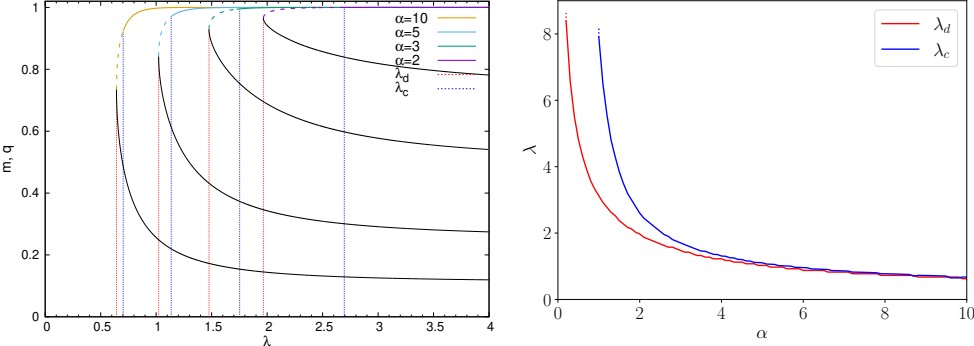

Figure 6: Left: order parameter $m = q$ in the Bayes optimal case for Ising prior, Gaussian noise and $p = 3$. Dashed lines indicate the values of $\lambda$ for which the magnetized phase is metastable with respect to the paramagnetic one. Right: phase diagram in the $\alpha - \lambda$ plane. The paramagnetic state $m = 0$ is always locally stable.

for $p > 2$ [11]. This is problematic from the algorithmic point of view, as it constitutes a hard phase for inference, meaning that there is no easy-to-hard threshold at finite $\alpha$. Meanwhile, the possible-to-impossible threshold is $\alpha_s = 0$ since the $m = 1$ solution always exists at finite $\alpha$. Hence, the computational gap is quite serious in this case. As a way to circumvent this, we propose in sec. 5.2.3 to mix $p = 2$ interactions to destabilize the paramagnetic state.

We report in Fig. 7 the behavior of the G-AMP using the informative (large overlap with the signal) initial condition (see sec. D.1). The results compare well with the solution of the SE equations. In Fig. 19 in the appendix we show results of r-BP algorithm which compare well with G-AMP.

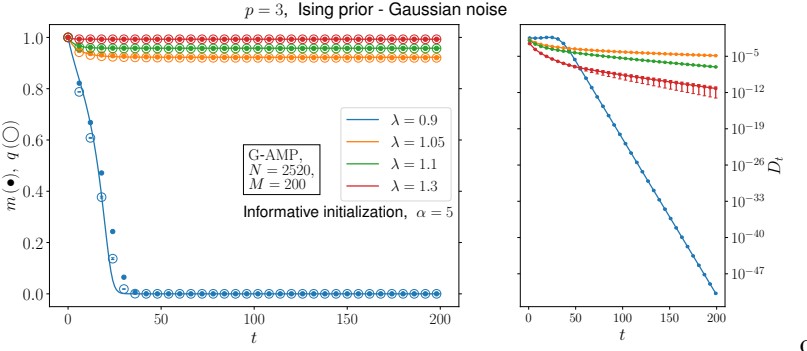

Figure 7: Evolution of the order parameters in the G-AMP algorithm for $\alpha = 5$ on random instances of the problem with $N = 2520$, $M = 200$. Data is averaged over 10 instances. Solid lines represent the SE predictions. On the right panel, the convergence parameter is plotted.

## 5.2 Gaussian prior and additive Gaussian noise

Let us turn to the case of Gaussian prior. In sec. 3.4.3 we obtained results for the case of Gaussian prior and additive noise by the replica approach. We focus on the Bayes optimal case, and the corresponding equation of state and the free energy expression are given by Eq. (108) and Eq. (109), respectively. We also checked in sec. C.2 the stability of the paramagnetic solution by computing the second derivative of the free energy. It turns out that for $p > 2$ the paramagnetic state is stable for any $\alpha, \lambda > 0$.

For the message passing approach, we examine the r-BP algorithm (Algorithm 1) and G-AMP algorithm (Algorithm 2) with the input function Eq. (178) and the output function Eq. (183). The SE equations are Eq. (224)-Eq. (226) combined with Eq. (232)-Eq. (236), showing a consistency with the replica formula.

### 5.2.1 $p = 2$ case

The value $m = 0$ is always a solution to (108). Two other solutions arise by solving the remaining second order equation in $m$, to find

$$m_{\pm} = \frac{\alpha}{2} \pm \gamma(\alpha, \lambda), \qquad \gamma(\alpha, \lambda) \equiv \frac{1}{2}\sqrt{(\alpha - 2)^2 + \frac{4}{\lambda^2}}. \tag{244}$$

Using $\gamma(\alpha, \lambda) \geq \frac{1}{2}\sqrt{(\alpha - 2)^2} = \frac{|\alpha - 2|}{2}$, one can easily show that $m_+ \geq 1 \; \forall \alpha$, meaning that it represents an unphysical solution. The correct solution is thus $m = \frac{\alpha}{2} - \gamma(\alpha, \lambda)$. It satisfies, as a function of $\lambda$, $m \leq 1$ for $\alpha \geq 2$ and $m \leq \alpha - 1$ for $\alpha < 2$, implying that the perfect reconstruction could be possible only when $\alpha \geq 2$ and only the paramagnetic solution exists for $\alpha < 1$. In the limit $\lambda \to \infty$, one has $m = 0$ for $\alpha < 1$, $m = \alpha - 1$ for $1 \leq \alpha < 2$ and $m = 1$ for $\alpha \geq 2$. The critical value of $\lambda$ above which the non-trivial solution $m > 0$ appears can also be obtained by assuming the continuity of the solution: the critical point $\lambda_c(\alpha)$ is thus derived from the equation $m = \frac{\alpha}{2} - \gamma(\alpha, \lambda_c) = 0$, leading to

$$\lambda_c(\alpha) = \frac{1}{\sqrt{\alpha - 1}}. \tag{245}$$

This also coincides with the stability condition for the paramagnetic solution obtained in sec. C.2. The result of this analysis is displayed in Fig. 8.

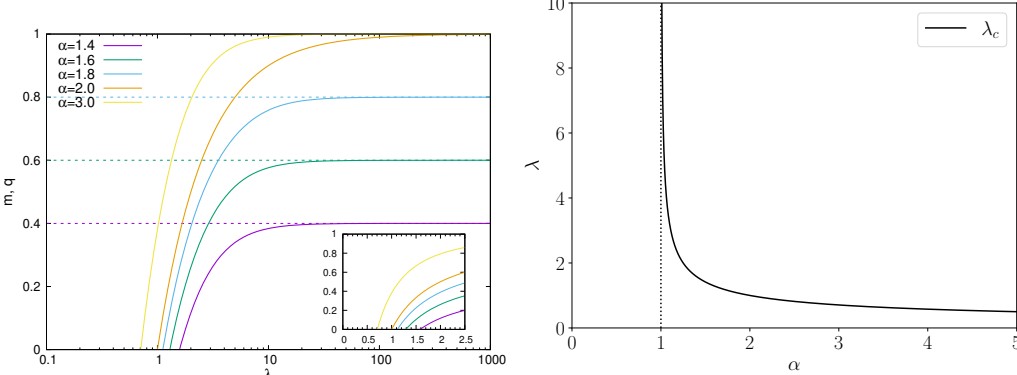

Figure 8: Left: Order parameter $m = q$ in the Bayes optimal case for the Gaussian prior, Gaussian noise and $p = 2$. Inset: zoom on the region around $\lambda_c(\alpha)$. Right: phase diagram in the $\alpha - \lambda$ plane.

From the algorithmic point of view, one observes a phase where inference is impossible for $\lambda < \lambda_c(\alpha)$, and a phase where inference is easy for $\lambda > \lambda_c(\alpha)$. This yields the possible-to-impossible threshold $\alpha_s$ in the limit $\lambda \to \infty$ as $\alpha_s = 1$, which is in contrast to the Ising prior case where the perfect reconstruction solution exists for any finite $\alpha$. Meanwhile, the easy-to-hard threshold for perfect reconstruction in the limit $\lambda \to \infty$ is given by $\alpha_P = 2$ from the above discussion.

In Fig. 9 we show time evolution of the overlaps $m^t$ and $q^t$ (points) computed using G-AMP, with informative and uninformative initialization, at $N = 2500, M = 400, \alpha = 1.6$. The results compare well with the solution of the SE equation (solid lines). However for

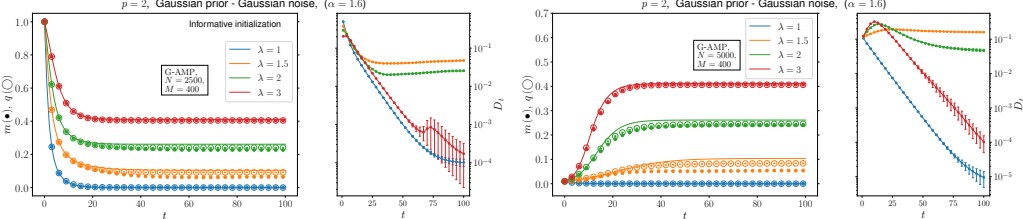

Figure 9: Evolution of the order parameters in the G-AMP algorithm for $\alpha = 1.6$ on random instances of the problem. Two different initializations (informative, uninformative) are considered. Solid lines represent the State Evolution prediction.

small $\lambda$, it can be seen that the convergence parameter $D_t$ does not vanish. This is a general feature that we observe throughout the numerical simulation. We presume this is related to finite size corrections becoming particularly relevant close to the transition point. See also the discussion in sec. E.

In Fig. 20 in the appendix we show results of r-BP algorithm for the informative initialization which can be compared well with G-AMP.

### 5.2.2  $p = 3$ case

Again, the value $m = 0$ is always a solution to (108). A non-trivial $m \neq 0$ solution has to satisfy $1 + \lambda^2(1 - m^3) + \alpha\lambda^2 m^2 = \alpha\lambda^2 m$, from which

$$\lambda(m, \alpha) = \frac{1}{\sqrt{m^3 - \alpha m^2 + \alpha m - 1}}. \tag{246}$$

The plot of $m$ as a function of $\lambda$ can be obtained by inverting (246), as shown in Fig. 10. Notice that the radicand at denominator of (246) can be written as $(m-1)(m^2 - (\alpha - $

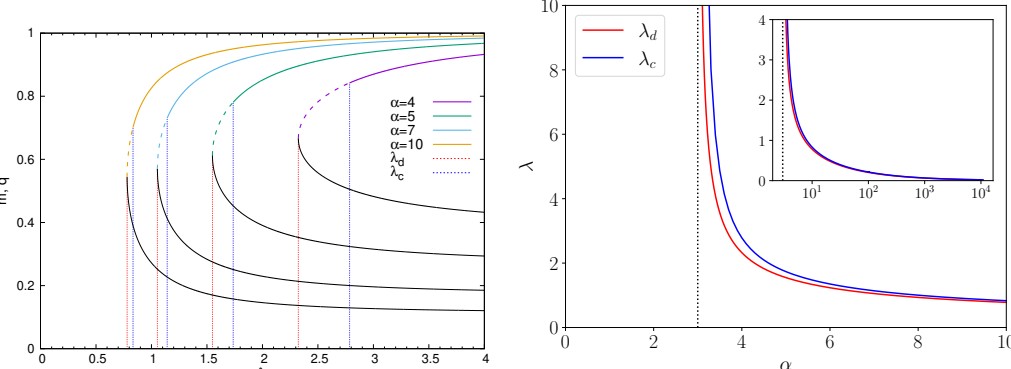

Figure 10: Left: Order parameter $m = q$ in the Bayes optimal case for the Gaussian prior, Gaussian noise and $p = 3$. Right: phase diagram in the $\alpha - \lambda$ plane.

$1)m + 1)$. This implies that when $\Delta = (\alpha - 1)^2 - 4 = (\alpha + 1)(\alpha - 3) < 0$, that is for $\alpha < 3$, the radicand is negative, and the non-trivial solution cannot exist, implying that the possible-to-impossible threshold is $\alpha_s = 3$ in this case.

For $\alpha > \alpha_s$ we can obtain the spinodal point $\lambda_d(\alpha)$ of the non-trivial solution by imposing $\frac{\partial}{\partial m}\lambda(m, \alpha) = 0$, that results in the second order equation $3m^2 - 2\alpha m + \alpha = 0$. Of the two branches of solutions, we keep the one compatible with the physical constraint $0 \le m \le 1$, which gives for the value of the order parameter at the spinodal point

$$m_d(\alpha) = \frac{\alpha - \sqrt{\alpha(\alpha - 3)}}{3}. \tag{247}$$

The dynamical transition line is then obtained as $\lambda_d(\alpha) = \lambda(m_d(\alpha), \alpha)$. In the large $\alpha$ limit $m_d(\alpha) \to \frac{1}{2}$, and one can observe that $\lambda_d(\alpha)$ is actually well approximated by $\lambda_d(\alpha) \sim \sqrt{\frac{8}{2\alpha - 7}}$, being the relative error of the order of 0.5% already for $\alpha = 10$.

The point $\lambda_c(\alpha)$ signals a first order thermodynamic transition where the ferromagnetic state becomes dominant with respect to the paramagnetic one. We have the difference in free energy as

$$f_{\mathrm{RS}}(m = 0) - f_{\mathrm{RS}}(m_*) = \frac{1}{2}\ln(1 - m_*) + \frac{m_*}{2} - \frac{\alpha}{p}\ln\left(1 - \frac{\lambda^2}{1 + \lambda^2}m_*^p\right), \tag{248}$$

where $m_*$ is given by the non-trivial solution of Eq. (246). The critical point $\lambda_c(\alpha)$ is computed from the zero point of this free energy difference. The two lines $\lambda_c(\alpha), \lambda_d(\alpha)$ are plotted in the right panel of Fig. 10. Since the paramagnetic solution is stable for the whole $\alpha > \alpha_s$ region, the easy-to-hard threshold does not exist at finite $\alpha$ as in sec. 5.1.2.

In Fig. 11 we show the time evolution of the overlaps $m^t$ and $q^t$ computed using G-AMP at $N = 5040, M = 400, \alpha = 5$ with several values of $\lambda$. Here the informative initialization is used. The results of G-AMP compare well with the solution of the SE equation (Eq. (224)-Eq. (226) combined with with Eq. (232)-Eq. (236)).

In Fig. 22 in the appendix we show results of r-BP algorithm for the informative initialization which can be compared well with G-AMP.

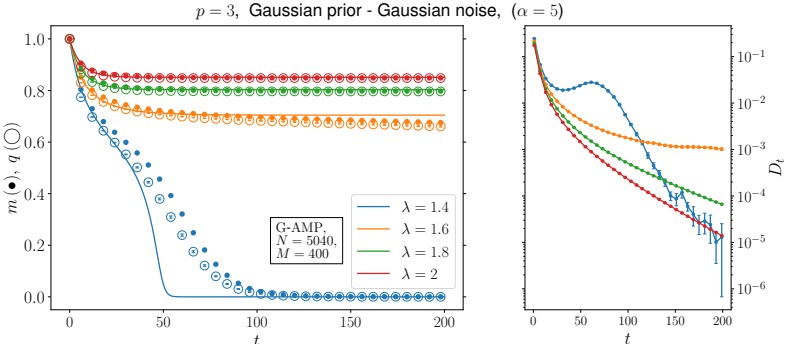

Figure 11: Evolution of the order parameters in the G-AMP algorithm for $\alpha = 5$ on random instances of the problem with $N = 5040$, $M = 400$. Data is averaged over 10 instances. Solid lines represent the State Evolution predictions. On the right panel, the convergence parameter is plotted.

### 5.2.3 mixed case: $p = 2 + 3$

As we already pointed out in the previous sections (see also appendix C), the $m = 0$ solution is always stable for $p > 2$ in the dense limit $c \gg 1$, which is problematic in the context of inference. As a way to circumvent this, we propose a mixed model in which two species of interactions ($p_1$ and $p_2$) are present. The total number of interactions, or measurements, is $N_{\blacksquare} = N_1 + N_2$. The connectivity of each species is defined as $c_i = \alpha_i M$, then $N_i = \alpha_i M N / p_i$. The idea behind this is to contaminate a $p_2 = 3$ system with some fraction of $p_1 = 2$ measurements in order to have a first order transition in the magnetization and an instability of the paramagnetic solution at finite $\lambda$ at the same time.

The only modification to the pure $p$ case is that the summation over $\blacksquare$ in the interaction part of the free energy Eq. (76) can be split according to the two species $p_1$ and $p_2$. This produces the following equation of state

$$\frac{m}{1-m} = \frac{\alpha_1 \lambda^2 m^{p_1 - 1}}{1 + \lambda^2 (1 - m^{p_1})} + \frac{\alpha_2 \lambda^2 m^{p_2 - 1}}{1 + \lambda^2 (1 - m^{p_2})} \equiv D_{p_1, p_2}(\alpha_1, \alpha_2, \lambda, m), \qquad (249)$$

under the homogeneous, Bayes optimal and replica symmetric ansatz.

We specialize in the following to the case $p_1 = 2$, $p_2 = 3$ and we fix $\alpha_1 = 2$. This choice is due to the fact that this allows us to have an instability of the paramagnetic state for finite $\lambda_c = (\sqrt{\alpha_2 - 1})^{-1} = 1$. The phase diagram in the $\alpha_2 - \lambda$ plane is depicted in Fig. 12. When $\alpha_2 \leq 2$ (which coincides to the condition $\alpha_2 \leq \alpha_1$) the transition in $\lambda$ is second order, and it is first order otherwise. Interestingly, the discontinuity of the order parameter emerges continuously when moving away from $\alpha_2 = 2$. This means that for values of $\alpha_2$ bigger than but very close to 2 the first order transition resembles very much a continuous one, see also Fig. 13.

In Fig. 14 we report the result of G-AMP compared with the solution of the SE equation. Deviations close to the transition can be possibly ascribed to finite $N, M$ corrections.

## 5.3 Gaussian prior and sign output

Here the case of Gaussian prior and sign output is treated. The equation of states and the free energy expression in the replica approach are given by Eq. (112) and Eq. (113), respectively, where $H(x)$ is defined in Eq. (91). This formula shows the consistency with

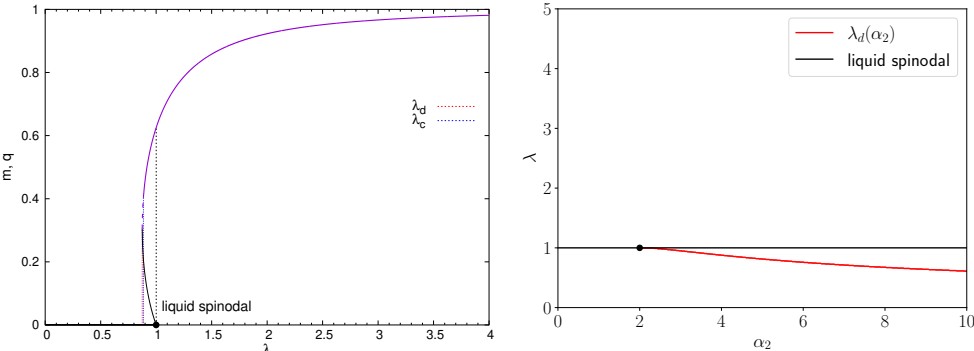

Figure 12: Left: order parameter $m = q$ for the Gaussian prior, Gaussian noise and mixed $p = 2 + 3$. The parameter $\alpha$ of each species is $\alpha_1 = 2$ and $\alpha_2 = 4$. Right: phase diagram in the $\alpha_2 - \lambda$ plane, for fixed $\alpha_1 = 2$.

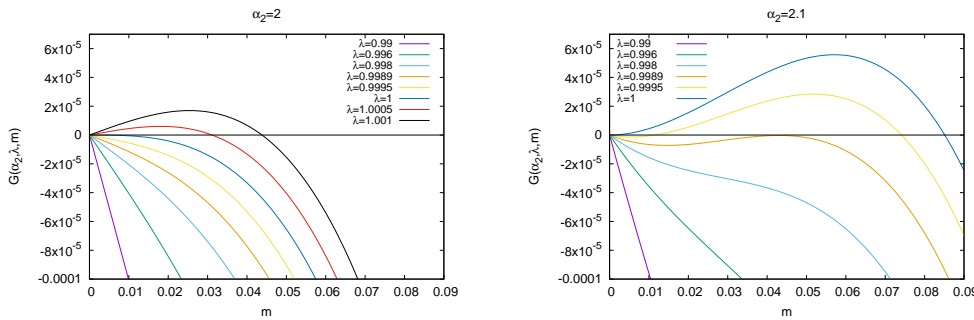

Figure 13: For the case $p_1 = 2$, $p_2 = 3$ and $\alpha_1 = 2$, we plot $G(\alpha_2, \lambda, m) \equiv (1 - m)D(2, \alpha_2, \lambda, m) - m$. Left: for $\alpha_2 \leq 2$ the transition in $\lambda$ is second order. The derivative of $G(m)$ at $m = 0$ becomes 0 exactly at $\lambda = 1$. Right: for $\alpha_2 > 2$ there is a discontinuous appearance of a non-trivial solution already for $\lambda < 1$. For $\alpha_2$ close to 2, however, the range of $\lambda$ and $m$ values interested is very small and the transition can resemble a continuous one.

the SE equation (Eq. (224)-Eq. (226) combined with Eq. (243)) in the message passing approach.

For the message passing approach, we examine the r-BP algorithm (Algorithm 1) and G-AMP algorithm (Algorithm 2) with the input function Eq. (178) and output function Eq. (186)-Eq. (187). Again, we limit ourselves to the Bayes optimal case. The only control parameter is in this case $\alpha$. The transition is second order for $p = 2$ and first order for $p = 3$. See Fig. 15. In this case, in the context of perfect reconstruction, there are no meaningful threshold at finite $\alpha$ due to the mismatch between the prior and the noise model.

We report in Fig. 16 and Fig. 17 the behavior of the G-AMP using the informative/uninformative initial conditions for $p = 2$ and $p = 3$ case respectively. The results again compare well with the solution of the SE equations.

In Fig. 23 in the appendix we show results of the r-BP algorithm for the $p = 2$ case. The comparison with SE equations is good up to $\alpha = 3$, though for $\alpha = 6$ we observe strong deviations; we also notice that the observable $Q_t = \langle v_{i\mu}^t \rangle$ is in this case consistently different from unity. This is presumably due to strong finite $M$ corrections, since some

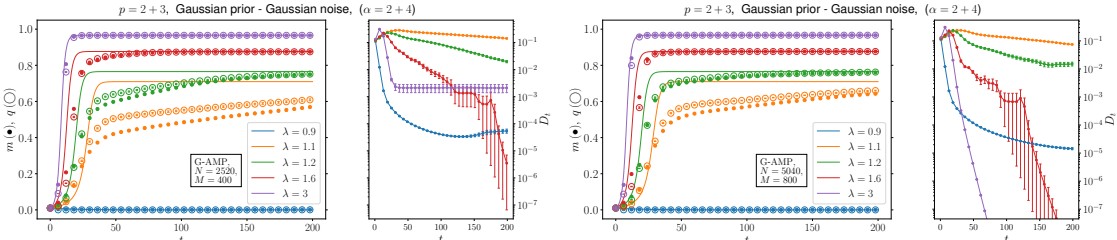

Figure 14: Evolution of the order parameters in the G-AMP algorithm for the mixed $p = 2+3$ case with $\alpha_1 = 2$, $\alpha_2 = 4$. Even starting from an uninformative initialization, partial recovery of the planted solution is possible thanks to the instability of the paramagnetic phase. Solid lines represent the State Evolution predictions. On the right panel, the convergence parameter is plotted.

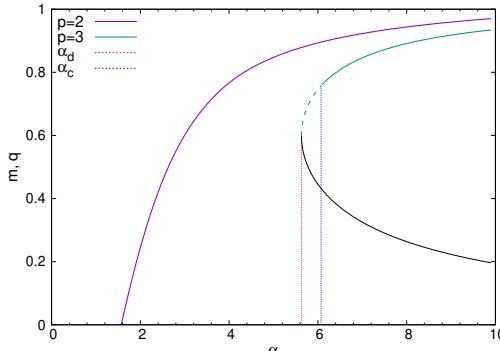

Figure 15: Order parameter m=q for the Gaussian prior and sign output. The transition in $\alpha$ is second order for $p = 2$ and first order for $p = 3$.

improvement is present when increasing $M$ to $M = 300$ as shown in Fig. 24. In Fig. 16, we have shown that for $M = 400$ the G-AMP algorithm does not suffer from this instability. See also Fig 26 discussed in appendix concerning uninformative initialization and truly random initialization.

## 5.4 Dependence of the convergence of algorithms on the choice of $F$

As discussed in previous sections we found the macroscopic quantities like the order-parameters, equation of states and the free energy are the same for the deterministic and disordered factors $F_{\blacksquare\mu}$ from the theoretical perspective in the dense limit. However behavior of the message passing algorithms applied on systems with finite $N, c$, and $M$ can be different between two cases. For the $p = 2$ case (both with Ising/Gaussian priors) we found the message passing algorithms (r-BP and G-AMP) do not converge properly with the deterministic model $F_{\blacksquare,\mu} = 1$ Eq. (9) and we had to resort to the model with random $F_{\blacksquare,\mu}$ Eq. (10) in numerical analysis. On the other hand we did not encounter such problems in the $p = 3$ cases.

One might expect that the residual global symmetries of the problems may affect convergence of the algorithms. Among the global symmetries listed in sec 2.2, the rotational symmetry exists only in the $p = 2$ model with Gaussian prior and $F_{\blacksquare,\mu} = 1$. Such a symmetry does not exist with the Ising prior. Thus we cannot attribute the reason for the failure of convergence in the $p = 2$ model, which happen with both the Ising/Gaussian priors, entirely to the rotational symmetry. The absence of the convergence problem in the $p = 3$ case with the deterministic model $F_{\blacksquare,\mu} = 1$ suggests that the permutation

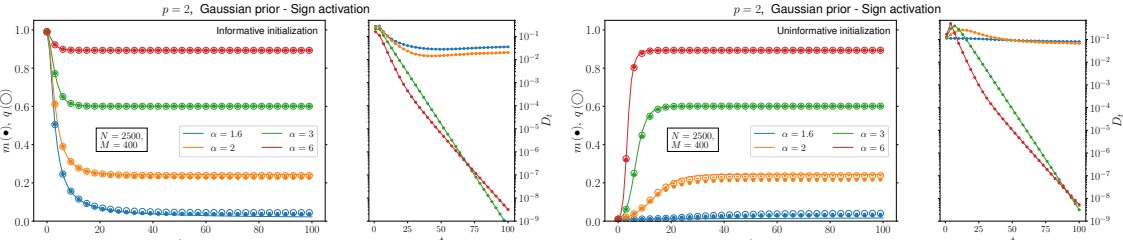

Figure 16: Evolution of the order parameters in the G-AMP algorithm averaged over 10 random instances and for two different initializations. Solid lines represent the State Evolution prediction

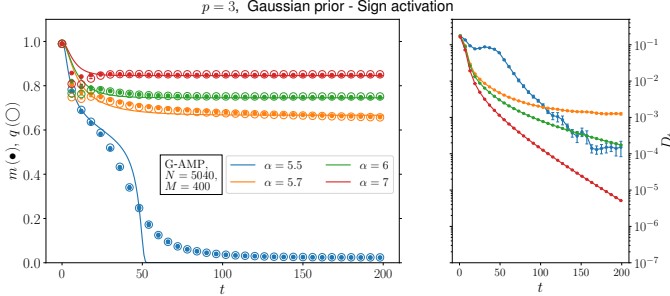

Figure 17: Evolution of the order parameters in the G-AMP algorithm on random instances of the problem with $N = 5040$, $M = 400$. Data is averaged over 5 instances. Solid lines represent the State Evolution predictions. On the right panel, the convergence parameter is plotted.

symmetry can be broken dynamically by the message passing algorithms. Let us note also that we performed some Monte Carlo simulations (not reported in the present paper) on the $p = 2$ Ising/Gaussian models with $F_{\blacksquare,\mu} = 1$ and found that statistical inferences are indeed possible modulo the rotational/permutation symmetries (these symmetries are not broken in the Monte Carlo simulations of finite size systems in equilibrium).

We speculate that the reason for the difference between the convergence of the message passing algorithms between the $p = 2$ and $p = 3$ cases may be attributed to some differences in the abundance of short loops. In general, message passing algorithms assume that the system is locally tree-like. Presence of short loops can hamper the convergence of the algorithm. In the $p = 2$ cases, the shortest loops are the triangular loops (See Fig. 2 a)) while triangular loops are absent and the shortest loops are rectangular in the $p = 3$ model. In the idealized dense limit $N \gg c \gg 1$ the presence of such short loops become negligible but may be the values of $N$ and $c$ used in our analysis reported below are not separated enough.

# 6 Conclusions

To summarize, we proposed a scheme of tensor factorization based on sparse measurements of tensor components represented by a graphical model which is dense but not globally coupled. We expect that the setup will be useful in cases where substantial amount of data is missing as in recommendation systems heavily used in social network services. We studied the Bayesian inference of the tensor factorization by statistical mechanics approaches: the replica method and message passing algorithms.

We considered the dense limit $N \gg M \gg 1$ [11, 12] which amounts to consider dense but not fully connected graph. This limit is useful from theoretical point of view in two aspects: 1) it allows exact theoretical analysis since loop corrections vanish and 2) the r-BP (relaxed belief propagation) and G-AMP (generalized approximate message passing) become valid in this limit.

From practical point of views, our result on the dense limit will be useful to consider situations where effective rank of tensors is not quite low. As mentioned in sec. 2, this can happen for instance in the case of facial images [27] and in recommendation systems [29]. Conventional algorithm like alternating least-square, gradient based methods can also be applied on our problem. It will be very interesting to test our setup for real-world data using various algorithms including the message passing algorithms developed in the present paper.

There are numerous possibilities to extend our present work. In the present paper we considered just one specie of vectors $x_{i\mu}$. The theory can be straightforwardly extended to the case of multi-species $x_{i\mu}, y_{i\mu}, z_{i\mu}, \ldots$. This will be useful for instance in the context of recommendation systems and dictionary learning $Y = DX$ where one usually consider two species, 'dictionary' $D$ and 'sparse representation' $X$.

## Acknowledgments

We thank Yoshiyuki Kabashima, Sakata Ayaka, Takashi Takahashi, Koki Okajima and Lenka Zdeborová, for useful discussions. This work was supported by KANENHI (No. 19H01812), (No. 21K18146), (No. 22H05117), (No. 18K11463), (No. 17H00764), (No. 22K12179) from MEXT, Japan.

# A  Some useful formulas

In the following we collect some useful formulas. We will use the short-hand notation Eq. (63), which reads as,

$$\int \mathcal{D}z \ldots = \int_{-\infty}^{\infty} \frac{dz}{\sqrt{2\pi}} e^{-\frac{z^2}{2}} \ldots. \tag{250}$$

**Lemma 1.** *For any analytic function $f(h)$ and any constant $C$,*

$$f(h + C) = e^{C\frac{\partial}{\partial h}} f(h). \tag{251}$$

**Lemma 2.** *For any analytic function $f(h)$ and any constant $C \geq 0$,*

$$e^{\frac{C}{2}\frac{\partial^2}{\partial h^2}} f(h) = \int \mathcal{D}z f(h \pm \sqrt{C}z). \tag{252}$$

**Lemma 3.** *For any $m$-times differentiable functions $F(h_1, h_2, \ldots, h_n)$ and any positive integer $m$*

$$\left(\sum_{a=1}^{n} \frac{\partial}{\partial h_a}\right)^m F(h_1, h_2, \ldots, h_n)\Bigg|_{\{h_a=h\}} = \frac{\partial^m}{\partial h^m} F(h, h, \ldots, h). \tag{253}$$

**Corollary 1.** *For any analytic functions $\{f_a(h)\}_{a=1,\ldots,n}$ and any constant $C \geq 0$,*

$$e^{\frac{C}{2}\left(\sum_{a=1}^{n} \frac{\partial}{\partial h_a}\right)^2} \prod_{a=1}^{n} f_a(h_a)\Bigg|_{\{h_a=0\}} = \int \mathcal{D}z \prod_{a=1}^{n} f_a(\pm\sqrt{C}z). \tag{254}$$

**Lemma 4.** *For any positive integer $n$,*

$$\int dw W(w) \mathcal{D}z_0 \frac{\int \mathcal{D}z_1 W^{(n)}(\Xi)}{\int \mathcal{D}z_1 W(\Xi)} = 0, \tag{255}$$

*where*

$$\Xi \equiv w - Az_0 - Az_1. \tag{256}$$

**Lemma 5.** *If $\beta = 1$ and $m = q$, for any positive integer $n$,*

$$\int dw_* W_*(w_*) \int \mathcal{D}z_0 \frac{\int \mathcal{D}z_1 W_*^{(n)}(\Xi)}{\int \mathcal{D}z_1 W_*(\Xi)} = 0.$$

*Proof.* Change variables from $z_1$ to $\zeta_1 = w_* - Az_0 - Az_1$.

$$(\text{l.h.s.}) = \int dw_* W_*(w_*) \int \mathcal{D}z_0 \frac{\int d\zeta_1 e^{-\frac{1}{2A^2}\zeta_1^2 + \frac{1}{A^2}(w_*-Az_0)\zeta_1} W_*^{(n)}(\zeta_1)}{\int d\zeta_1 e^{-\frac{1}{2A^2}\zeta_1^2 + \frac{1}{A^2}(w_*-Az_0)\zeta_1} W_*(\zeta_1)}.$$

Change variables again from $z_0$ to $\zeta_0 = \frac{w_*}{A} - z_0$.

$$(\text{l.h.s.}) = \int \frac{\mathrm{d}\zeta_0}{\sqrt{2\pi}} e^{-\frac{1}{2}\zeta_0^2} \left( \int \mathrm{d}w_* \, e^{-\frac{1}{2A^2}w_*^2 + \frac{1}{A}\zeta_0 w_*} W_*(w_*) \right)$$

$$\times \frac{\int \mathrm{d}\zeta_1 \, e^{-\frac{1}{2A^2}\zeta_1^2 + \frac{1}{A}\zeta_0\zeta_1} W_*^{(n)}(\zeta_1)}{\int \mathrm{d}\zeta_1 \, e^{-\frac{1}{2A^2}\zeta_1^2 + \frac{1}{A}\zeta_0\zeta_1} W_*(\zeta_1)}.$$

The denominator can be reduced and it becomes

$$(\text{l.h.s.}) = \int \frac{\mathrm{d}\zeta_0}{\sqrt{2\pi}} e^{-\frac{1}{2}\zeta_0^2} \int \mathrm{d}\zeta_1 \, e^{-\frac{1}{2A^2}\zeta_1^2 + \frac{1}{A}\zeta_0\zeta_1} W_*^{(n)}(\zeta_1)$$

$$= \int \mathrm{d}\zeta_1 \, W_*^{(n)}(\zeta_1) \int \frac{\mathrm{d}\zeta_0}{\sqrt{2\pi}} e^{-\frac{1}{2}\left(\zeta_0 - \frac{1}{A}\zeta_1\right)^2}$$

$$= \int \mathrm{d}\zeta_1 \, W_*^{(n)}(\zeta_1)$$

$$= \left[ W_*^{(n-1)}(\zeta_1) \right]_{\zeta_1 = -\infty}^{\infty}.$$

We expect that $W_*(w_*)$ is a well-behaved probability distribution, and its derivatives converge to zero at infinity. Therefore

$$(\text{l.h.s.}) = 0.$$

$\square$

**Lemma 6.** *Given a matrix of size $n \times n$ in the form $M_{ab} = M_1\delta_{ab} + M_2$, its determinant is obtained as,*

$$\det M = M_1^n \det(\delta_{ab} + M_2/M_1) = M_1^n(1 + nM_2/M_1) \tag{257}$$

**Lemma 7.** *With $A$ and $C$ being certain square (sub)matrices, we have the following formula for the determinant of a square matrix.*

$$\det \begin{pmatrix} A & B \\ B^t & C \end{pmatrix} = \det A \det(C - BA^{-1}B^t) \tag{258}$$

# B    Analysis of cumulant expansion

In the following we explain the cumulant expansion discussed in sec. 3.3 which is used to extract the interaction part of the replicated free-energy.

## B.1    Order $\lambda$

At the lowest order the contribution can be represented by the diagrams shown in Fig.2 a). We readily find using Eq. (44).

$$\langle \pi_{\blacksquare}^a \rangle_{\epsilon,0} = \sum_{\mu=1}^{M} \frac{\lambda}{\sqrt{M}} F_{\blacksquare,\mu} \langle \prod_{j \in bs_1} x_{j\mu}^a \rangle_{\epsilon,0} = 0 \tag{259}$$

because of the reflection symmetry $P_{\text{pri.}}(x) = P_{\text{pri.}}(-x)$ which holds in both the Ising and Gaussian priors that we consider in the present paper (see sec 2.2).

## B.2   Order $\lambda^2$

At the next order in the expansion Eq. (72) we find

$$\langle \pi^a_{\blacksquare_1} \pi^b_{\blacksquare_2} \rangle_{\epsilon,0} - \langle \pi^a_{\blacksquare_1} \rangle_{\epsilon,0} \langle \pi^b_{\blacksquare_2} \rangle_{\epsilon,0}$$

$$= \frac{\lambda^2}{M} \sum_{\mu_1=1}^{M} \sum_{\mu_2=1}^{M} F_{\blacksquare_1,\mu_1} F_{\blacksquare_2,\mu_2} \underbrace{\left[ \langle \prod_{j_1 \in \blacksquare_1} x^a_{j_1 \mu_1} \prod_{j_2 \in \blacksquare_1} x^b_{j_2 \mu_2} \rangle_{\epsilon,0} - \langle \prod_{j_1 \in \blacksquare_1} x^a_{j_1 \mu_1} \rangle_{\epsilon,0} \langle \prod_{j_2 \in \blacksquare_1} x^b_{j_2 \mu_2} \rangle_{\epsilon,0} \right]}_{\delta_{\blacksquare_1,\blacksquare_2} \delta_{\mu_1,\mu_2} \prod_{i \in \blacksquare_1} Q^{ab}_i}$$

$$= \delta_{\blacksquare_1,\blacksquare_2} \frac{\lambda^2}{M} \sum_{\mu=1}^{M} F^2_{\blacksquare_1,\mu} \prod_{i \in \partial \blacksquare_1} Q^{ab}_i = \delta_{\blacksquare_1,\blacksquare_2} \lambda^2 \prod_{i \in \partial \blacksquare_1} Q^{ab}_i \tag{260}$$

To derive the above equation we used

$$\langle x^a_{i\mu} x^b_{j\nu} \rangle_{\epsilon,0} = \delta_{ij} \delta_{\mu\nu} Q^{ab}_i \tag{261}$$

which follows from Eq. (43), Eq. (44) and Eq. (50). We also assumed

$$\frac{1}{M} \sum_{\mu=1}^{M} F^2_{\blacksquare,\mu} = 1 \tag{262}$$

for the factor $F_{\blacksquare,\mu}$ introduced in sec 2.1.1. This trivially holds for the 'deterministic model' $F_{\blacksquare,\mu} = 1$ but also holds for the 'random model' where $F_{\blacksquare,\mu}$ is an iid random variable with unit variance.

   This is a type (A) contribution discussed in sec. 3.3 which can be represented by doubling the diagrams shown in Fig.2 a) : two diagrams associated with function nodes $\blacksquare_1$ and $\blacksquare_2$ which are identical but carrying two replicas $a$ and $b$. Equivalently this also corresponds to diagrams Fig.2 b) with two function nodes carrying replicas $a$ and $b$ under the condition that the variable nodes 'j' and 'k' are identical (otherwise the contribution becomes zero by the same symmetry reason as for the vanishing of $O(\lambda)$ terms).

   So far we found $O(\lambda)$ term in the expansion Eq. (72) is zero while $O(\lambda^2)$ is non-zero. Up to this order, it is natural to choose $A = \lambda^2$ in the Plefka expansion Eq. (47) and Eq. (48). Using the above results in Eq. (71), Eq. (72) and Eq. (49)) we readily find,

$$-\beta \hat{F}_1[Q][\partial/\partial h^a_{\blacksquare}] = \frac{1}{2} \sum_{\blacksquare} \sum_{a,b=0}^{n} \prod_{j \in \partial \blacksquare} Q^{ab}_j \frac{\partial^2}{\partial h^a_{\blacksquare} \partial h^b_{\blacksquare}} \tag{263}$$

## B.3   Order $\lambda^3$

At $O(\lambda^3)$ we find terms which can be represented as the 'triangular' diagrams shown in Fig. 2 c) with the three function nodes '1', '2' and '3' carrying three replicas $a$, $b$ and $c$. Notice that the function nodes adjacent to each other (joined by the variable nodes in between) exhibit a 'triangle' which is the shortest loop. Each function node is associated with a $\pi$ and carries a factor $1/\sqrt{M}$. There is one common running index $\mu$ for the vector component which produces a factor $M$. For example the triangular diagram for $p = 2$ and $p = 4$ contains three and six variable nodes so that they yield

$$\left( \frac{1}{\sqrt{M}} \right)^3 M Q^{ab}_i Q^{bc}_j Q^{ca}_k \qquad \left( \frac{1}{\sqrt{M}} \right)^3 M Q^{ab}_i Q^{bc}_j Q^{ca}_k Q^{ab}_l Q^{bc}_m Q^{ca}_n \tag{264}$$

with $M = c/\alpha$ with $\alpha = O(1)$.

An important feature is that all function nodes carry a replica while all variable nodes carry two replicas so that a variable node participating the triangle carries a replica overlap like $Q^{ab}$. For this to happen with 3 function nodes, $p$ has to be an even number. Note that for odd $p$ models, including $p = 3$, similar single loop diagrams in which each variable node carries a replica overlap like $Q^{ab}$ appear first at $O(\lambda^4)$ (rectangle). Generalization of the following discussion to such cases is straightforward.

Now let us consider whether the triangular diagrams (assuming $p$ is even) give relevant contributions to the free-energy. To this end let us consider the set of such diagrams containing a function node, say '1'. It is adjacent to $p$ variable nodes. Take one of them and call it as 'i'. The variable node 'i' has $c - 1$ adjacent function nodes other than '1'. Choose one of them and call it as '2'. The function node '2' has $p - 1$ variable nodes adjacent to it other than 'i'. Choose one of them and call it as 'j'. The variable node 'j' has $c - 1$ adjacent function nodes other than '2'. Choose one of them and call it as '3'. The function node '3' has $p - 1$ variable nodes adjacent to it other than 'j'. Choose one of them and call it as 'k'. Now what is the probability that 'k' is adjacent to the function node '1' to close a loop? Let us denote it as $p_{\text{connect}}$. To sum up the total number of such diagrams will scale as $p(p-1)^2(c-1)^2 p_{\text{connect}}$ so that their contribution to the free-energy per function node scale as

$$M \left( \frac{1}{\sqrt{M}} \right)^3 p(p-1)^2(c-1)^2 p_{\text{connect}} \sim c^{3/2} p_{\text{connect}} \tag{265}$$

for $c \gg 1$ assuming $\alpha$ and $p$ are finite. In the globally coupled case ($c \propto N^{p-1}$) surely we have $p_{\text{connect}} = 1$ so that the contribution *cannot* be neglected. This is in agreement with the recent observation in the matrix factorization problem [30] which corresponds to $p = 2$ case with global coupling. On the other hand in our dense limit $N \gg c \gg 1$, we expect $p_{\text{connect}} = O(c/N)$ so that it can be neglected by taking $N \to \infty$ limit first.

Note that in the presence of the random spreading code $F_\blacksquare^\mu$ the contribution vanishes whatever the connectivity by taking average over the realization of $F_\blacksquare^\mu$ (see sec. 2.1.1).

## B.4   Order $\lambda^4$

At order $O(\lambda^4)$ we find terms which contribute to $\hat{G}_2$. They consist of 4 function nodes carrying replicas $a$,$b$,$c$ and $d$.

- The simplest possibility is to use again diagrams shown in Fig. 2 a) with $\blacksquare$ carrying four replicas. In each of the $p$ variable nodes we attach a four point connected correlation function like $\langle x_{i\mu}^a x_{i\mu}^a x_{i\mu}^c x_{i\mu}^d \rangle_{\epsilon,0}^c$. Each function node carries $1/\sqrt{M}$ factor and there is one running global index $\mu$ so that the contribution of such a term to the free-energy per function node scales as $M(1/\sqrt{M})^4 \sim M^{-1}$ which vanish in the limit $M \to \infty$.

- (One particle reducible diagram) Another possibility is to use the diagrams shown in Fig. 2 b) where $\blacksquare_1$ and $\blacksquare_2$ carry replica pairs $(a, b)$ and $(c, d)$ respectively. At the variable node $i$ we attach a four point connected correlation function $\langle x_{i\mu}^a x_{i\mu}^a x_{i\mu}^c x_{i\mu}^d \rangle_{\epsilon,0}^c$ while on other variable nodes we attach $Q^{ab}$ or $Q^{cd}$. This case contributes to $\hat{G}_2$ but disappears in $\hat{F}_2$ by the Legendre transformation Eq. (46). Indeed one can show that they become canceled by $\beta \frac{(\hat{G}_0'[\epsilon_0])^2}{\hat{G}_0''[\epsilon_0]}$ which appear in Eq. (49). Note that the diagram can be disconnected into two parts by cutting the diagram at $i$. Such diagrams are examples of the so called one particle reducible diagrams and disappear in $\hat{F}$ which consists only of one particle irreducible (1PI) diagrams [51, 52].

## B.5 Higher orders

Let us consider a generic closed one-loop diagram of length $l(\geq 3)$ which goes through $l$ function nodes and $l$ variable nodes and carries $k$ replicas on each function node. Repeating the argument in sec B.3, we find that the contribution to the free-energy per function node can be estimated as the following:

- Running $\mu$ index gives a factor $M$

- A function node yields a factor $(\lambda/\sqrt{M})^k$

- The number of closed loop starting from a function node is $(c-1)^{l-1}p(p-1)^{l-1}p_{\text{connect}}$

- A variable node caries a $2k$ point connected correlation function of $x_{i\mu}^a$s.

- A function node caries also a factor $F^k$.

Combing these we find a contribution of order

$$M\left(\frac{\lambda}{\sqrt{M}}\right)^{lk}(c-1)^{l-1}p(p-1)^{l-1}p_{\text{connect}} \propto c^{l(1-\frac{k}{2})}p_{\text{connect}} \tag{266}$$

For $k=1$ we find it scales as $c^{l/2}p_{\text{connect}}$ so that such a contribution is relevant in globally coupled system for which $p_{\text{connect}}=1$. However, in the dense limit $p_{\text{connect}}\sim O(c/N)$ so that it vanishes by taking $N\to\infty$ limit first. The above analysis implies $N$ must grow faster than any power law of $c$ in order to get rid of contribution from loop diagrams.

For $k>2$ we find the contribution vanishes in $c\to\infty$ limit either in the dense limit or in globally coupled systems.

The contributions of diagrams with larger number of closed loops (2-loop, 3-loops,...) will vanish faster in $c\to\infty$ limit.

To summarize the relevant loop diagrams which can contribute to the free-energy in the $c\to\infty$ limit are those with $k=1$ in the case of global coupling. Using the dense coupling $N\gg c\gg 1$ we can avoid this. Such contributions of loops (with $k=1$) disappear also in the presence of the random spreading code $F_{\blacksquare}^\mu$ after averaging over $F_{\blacksquare}^\mu$. Indeed this is the situation in the standard fully connected $p$-spin spin-glass models ($p=2$ case is the Sherrington-Kirkpatrick model [59], more general $p$-spin model introduced first in [55]).

# C  Stability of the paramagnetic solution

The instability of the $m=0$ solution can be detected by studying the change of sign of the second derivative of the RS free-energy (we consider here the Bayes Optimal case $\beta=1$ and $q=m$) at $m=0$.

## C.1  Ising prior and additive gaussian noise

Let us begin with the entropic part of the free-energy. Our goal is to compute $\left.\frac{d^2}{dm^2}\frac{\partial}{\partial n}\frac{-\beta F_0(m,m)}{NM}\right|_{n=0}$. We start from the definition of the entropic part of the free-energy as given in the first line of Eq. (62), equipped with Eq. (100). One finds

$$\left.\frac{\partial}{\partial n}\frac{-\beta F_0(q,m)}{NM}\right|_{n=0,q=m} = -\frac{1}{2}(1+m)A(m,\alpha,\lambda)+e^{\frac{A(m,\alpha,\lambda)}{2}\frac{\partial^2}{\partial h^2}}\log 2\cosh\left(h+A(m,\alpha,\lambda)\right)\Big|_{h=0}, \tag{267}$$

where $A(m, \alpha, \lambda)$ is defined, for the additive gaussian noise case, in Eq. (103). Taking two derivatives with respect to $m$ we obtain

$$-\frac{\partial A}{\partial m} + B\left(\frac{\partial A}{\partial m}\right)^2 + \left(C - \frac{1}{2}(1+m)\right)\frac{\partial^2 A}{\partial m^2}, \tag{268}$$

where

$$C = e^{\frac{A}{2}\frac{\partial^2}{\partial h^2}}\frac{1}{2}\frac{\partial}{\partial h}\tanh(h+A)\Big|_{h=0} + e^{\frac{A}{2}\frac{\partial^2}{\partial h^2}}\tanh(h+A)\Big|_{h=0},$$
$$B = e^{\frac{A}{2}\frac{\partial^2}{\partial h^2}}\left[\frac{1}{4}\frac{\partial^3}{\partial h^3} + \frac{\partial^2}{\partial h^2} + \frac{\partial}{\partial h}\right]\tanh(h+A)\Big|_{h=0}, \tag{269}$$

which evaluated at $m=0$ gives $C_{m=0} = B_{m=0} = \frac{1}{2}$.

Let us turn now to the interaction part of the free-energy and compute $\frac{d^2}{dm^2}\frac{\partial}{\partial n}\frac{-\beta F_{\text{ex}}(m,m)}{NM}\Big|_{n=0}$. From Eq. (80), equipped with Eq. (84) in the case of gaussian noise, we obtain

$$\frac{d}{dm}\frac{\partial}{\partial n}\frac{-\beta F_{\text{ex}}(m,m)}{NM}\Big|_{n=0} = \partial_n\frac{\partial}{\partial q}\frac{-\beta F_{\text{ex}}(q,m)}{NM}\Big|_{n=0,q=m} + \partial_n\frac{\partial}{\partial m}\frac{-\beta F_{\text{ex}}(q,m)}{NM}\Big|_{n=0,q=m} =$$
$$= \frac{A}{2}. \tag{270}$$

Putting the pieces together

$$\frac{d^2}{dm^2}(-f(m)) = \left(B\frac{\partial A}{\partial m} - \frac{1}{2}\right)\frac{\partial A}{\partial m} + \left(C - \frac{1}{2}(1+m)\right)\frac{\partial^2 A}{\partial m^2},$$
$$\frac{d^2}{dm^2}(-f(m))\Big|_{m=0} = \frac{1}{2}\left(\frac{\partial A}{\partial m}\Big|_{m=0} - 1\right)\frac{\partial A}{\partial m}\Big|_{m=0}. \tag{271}$$

For $p=2$, we have that $\frac{\partial}{\partial m}A|_{m=0} = \frac{\alpha\lambda^2}{1+\lambda^2} > 0$. The condition of stability then implies $\frac{\partial A}{\partial m}\Big|_{m=0} < 1$ or

$$\begin{cases} 0 < \alpha < 1, & \forall \lambda > 0, \\ \alpha > 1, & \lambda < \frac{1}{\sqrt{\alpha-1}}. \end{cases} \tag{272}$$

For $p > 2$, we have instead $\frac{\partial}{\partial m}A|_{m=0} = 0$ and the second derivative is not enough to assess the stability of the paramagnetic phase. We can go on and compute the third derivative

$$\frac{d^3}{dm^3}(-f(m)) = \left(\frac{\partial B}{\partial m}\frac{\partial A}{\partial m} + B\frac{\partial^2 A}{\partial m^2}\right)\frac{\partial A}{\partial m} + \left(2B\frac{\partial A}{\partial m} - 1\right)\frac{\partial^2 A}{\partial m^2} + \left(C - \frac{1}{2}(1+m)\right)\frac{\partial^3 A}{\partial m^3}, \tag{273}$$

where we used the fact that $\frac{\partial C}{\partial m} = B\frac{\partial A}{\partial m}$. For $p=3$, the only term that survives at $m=0$ is the one proportional to $\frac{\partial^2 A}{\partial m^2}$, i.e. $\frac{d^3}{dm^3}(-f(m))|_{m=0} = -\frac{\partial^2 A}{\partial m^2}|_{m=0} = \frac{2\alpha\lambda^2}{1+\lambda^2} < 0 \ \forall \alpha, \lambda > 0$. For $p=4$, the third derivative is zero since $C_{m=0} = \frac{1}{2}$, and we expect a non-trivial contribution appearing in $\frac{d^4}{dm^4}(-f(m))|_{m=0}$ associated to the $\frac{\partial^3 A}{\partial m^3}$ term.

## C.2 Gaussian prior and additive gaussian noise

In the case of gaussian prior, from the expression of the entropic term Eq. 70 one obtains

$$\frac{d}{dm}\frac{\partial}{\partial n}\frac{-\beta F_0(m,m)}{NM}\bigg|_{n=0} = \partial_n\frac{\partial}{\partial q}\frac{-\beta F_0(q,m)}{NM}\bigg|_{n=0,q=m} + \partial_n\frac{\partial}{\partial m}\frac{-\beta F_0(q,m)}{NM}\bigg|_{n=0,q=m} =$$
$$= -\frac{1}{2}\frac{m}{1-m}. \tag{274}$$

Putting the pieces together (the interaction term is the same as in the Ising case)

$$\frac{d^2}{dm^2}(-f(m)) = -\frac{1}{2}\frac{1-2m}{(1-m)^2} + \frac{1}{2}\frac{\partial A}{\partial m},$$
$$\frac{d^2}{dm^2}(-f(m))\bigg|_{m=0} = \frac{1}{2}\left(\frac{\partial A}{\partial m}\bigg|_{m=0} - 1\right). \tag{275}$$

s For $p=2$, this gives the same stability condition as in the Ising case. For $p>2$, one obtains $\frac{d^2}{dm^2}(-f(m))\big|_{m=0} = -\frac{1}{2} < 0 \ \forall\alpha,\lambda > 0$.

# D  Details on numerical analysis of the message passings

## D.1  Initialization

In both r-BP and G-AMP experiments we consider two main kinds of initializations: informative initialization and uninformative initialization.

- In the **informative initialization**, we set the values of the messages $m_{i\mu\to\blacksquare}^{t=0}$, $v_{i\mu\to\blacksquare}^{t=0}$ (or equivalently $m_{i\mu}^{t=0}$, $v_{i\mu}^{t=0}$ in the G-AMP case) to

$$\begin{cases} m_{i\mu\to\blacksquare}^{t=0} = x_{i\mu}^* \\ v_{i\mu\to\blacksquare}^{t=0} = (x_{i\mu}^*)^2 \end{cases} \quad \text{(Gaussian prior)}. \tag{276}$$

The starting point of SE equations is then $m_0 \equiv \langle x_{i\mu}^* m_{i\mu}^{t=0}\rangle = q_0 \equiv \langle (m_{i\mu}^{t=0})^2\rangle = 1$ and $Q_0 = \langle v_{i\mu}^{t=0} = 1\rangle$. In the case of sign output, this definition is slightly modified

$$\begin{cases} m_{i\mu\to\blacksquare}^{t=0} = x_{i\mu}^*(a + \sqrt{a-a^2}\mathcal{N}(0,1)), \quad a = 0.99 \\ v_{i\mu\to\blacksquare}^{t=0} = a^{-1}(m_{i\mu\to\blacksquare}^{t=0})^2. \end{cases} \tag{277}$$

In this way, $m_0 = q_0 = a$ and $Q_0 = 1$. Furthermore, $v_{i\mu\to\blacksquare}^{t=0} > (m_{i\mu\to\blacksquare}^{t=0})^2 \ \forall i,\mu$, which makes the quantity under square root $V_{\blacksquare\to\mu}^{t=0}$ strictly positive.

- In the **uninformative initialization**, messages are set to

$$\begin{cases} m_{i\mu\to\blacksquare}^{t=0} = ax_{i\mu}^* + \sqrt{a-a^2}\mathcal{N}(0,1), \quad a = 0.01 \\ v_{i\mu\to\blacksquare}^{t=0} = 1 \quad \text{(Gaussian prior)}. \end{cases} \tag{278}$$

Also in this case we have $m_0 = q_0 = a$ and $Q_0 = 1$.

The main advantage of the 'uninformative' initialization with respect to a truly random one is twofold: it satisfies the Bayes-optimal condition $m = q$ from $t = 0$, and it helps selecting the 'correct' state with respect to a remaining global symmetry of the system

when $p$ is even. This symmetry is realized under the change of sign of a given $\mu$ component of all of the $N$ vectors, and we may refer to it as the symmetry under inversion on the $\mu$-plane. This symmetry remains even after the introduction of the random spreading factor for $p = 2$, but being global and discrete it happens to be spontaneously broken by the message passing algorithms. We show it in Fig. 26, by comparing the result of uninformative and **truly random initialization**

$$\begin{cases} m_{i\mu\to\blacksquare}^{t=0} = \sqrt{a}\mathcal{N}(0,1), & a = 0.01 \\ v_{i\mu\to\blacksquare}^{t=0} = 1 & (\text{Gaussian prior}). \end{cases} \tag{279}$$

With this choice, $m_0 = 0$, $Q_0 = 1$ but $q_0 = a$, so the algorithm is initialized slightly off the Bayes Optimal ansatz. In general, without any prior knowledge of the planted solution, we are able to infer it only up to a global sign on each $\mu$ component. In order to get non-zero magnetization, one has to modify its definition according to $m = \frac{1}{M} \sum_{\mu=1}^{M} \left| \frac{1}{N} \sum_{i=1}^{N} x_{i\mu}^* m_{i\mu} \right|$.

## D.2   Spreading factor

For $p = 2$, we used the 'disordered model' with random spreading factor $F_{\blacksquare\mu}$ (see sec 2.1.1) in order to eliminate the rotational symmetry and help the convergence of the algorithms. The spreading factor was also beneficial to the mixed case $p = 2+3$. Otherwise, for $p = 3$, we found the algorithms converge well with the 'disorder-free model' $F_{\blacksquare\mu} = 1$.

## D.3   Update of messages and check of convergence

We introduce a parameter $\epsilon$ to control the magnitude of the updates

$$m^{t+1} = m^t + \epsilon(f_{\text{input}}^{t+1} - m^t), \tag{280}$$
$$v^{t+1} = v^t + \epsilon(f_{\text{input,II}}^{t+1} - v^t).$$

Usually values of $\epsilon$ between 0.5 and 1 work well; in some cases, such as the mixed $p = 2+3$ Gaussian model, an accurate tuning of the parameter is crucial in order to get a good convergence of the algorithms.

For each set of parameters, the data is averaged over 10 random instances. Convergence is tracked via the parameter

$$D_t \equiv \langle |f_{\text{input}}^{t+1} - m^t| \rangle, \tag{281}$$

where $\langle \dots \rangle$ is an average over all the messages.

# E   Comparison between r-BP and G-AMP, finite size corrections

In this section we show more extensive numerical results, providing comparisons between the r-BP and G-AMP algorithms. In general, both algorithms give comparable performances. There are cases, such as the $p = 2$ Ising prior or the $p = 3$ Gaussian prior equipped with Gaussian noise, where at relatively small $N, M$ sizes the r-BP algorithm performs slightly better. And cases such as the $p = 2$ sign output, where instead the better scalability of G-AMP allows to easily reach bigger $M$ sizes and get rid of some strong finite $M$ corrections.

We also provide comparisons for different sizes of $N, M$. We found in general that when the transition is continuous, as in the $p = 2$ cases, stronger deviations are observed

for values of $\lambda$ ($\alpha$ in the sign output case) closer to the transition. Also the convergence rate of the algorithm deteriorates. The closer to the transition, the more relevant the finite $M$ corrections seem to become, see for example Fig. 21. Finite $M$ corrections can arise both as loops corrections and at the level of the saddle point evaluation of the replicated free-energy. Further away from the transition, also finite $N$ effects become more evident.

When the transition is discontinuous, as in the $p = 3$ cases, finite $N$ effects can destabilize the system close to the transition, see in particular Figs. 22 and 25.

## E.1   Ising prior - Gaussian noise

### E.1.1   $p = 2$

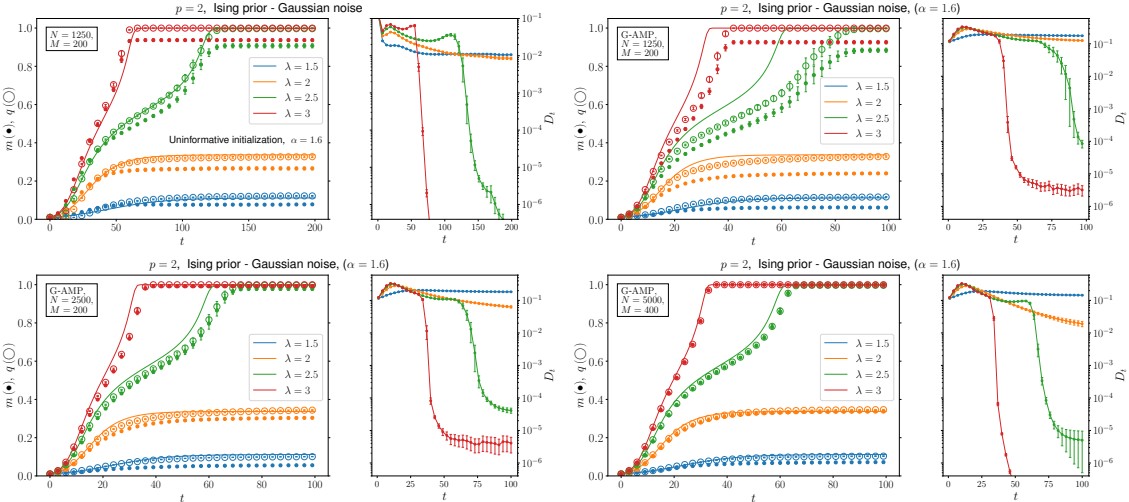

Figure 18: Comparison between r-BP algorithm (first panel) and G-AMP. For smaller sizes r-BP is more accurate, but increasing the size G-AMP drastically improves (last panel is the same as in Fig. 5). Solid lines represent the predictions from State Evolution. The values of $\epsilon$ used are $\epsilon = 0.5$ with r-BP and $\epsilon = 1$ with G-AMP.

### E.1.2   $p = 3$

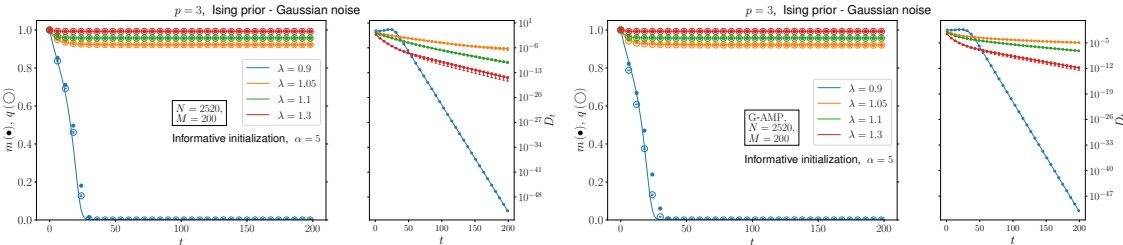

Figure 19: Comparison between r-BP algorithm (left panel) and G-AMP (right panel, from Fig. 7). In both cases $\epsilon = 0.5$.

## E.2 Gaussian prior - Gaussian noise

### E.2.1 $p = 2$

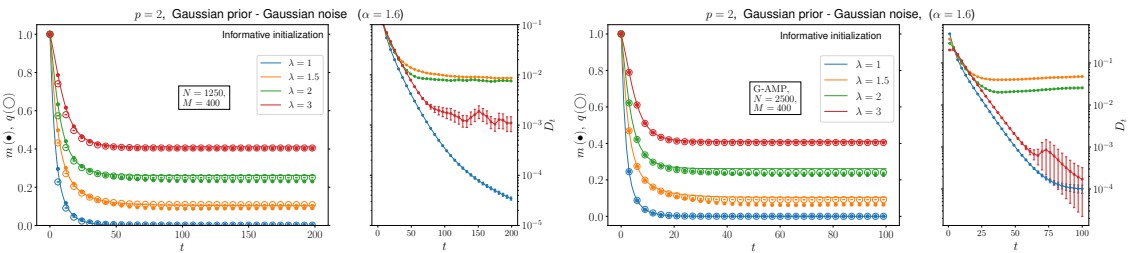

Figure 20: Comparison between r-BP algorithm (left panel) and G-AMP (right panel, the same as in Fig. 9). The values of $\epsilon$ in this case are respectively $\epsilon = 0.5, 1$

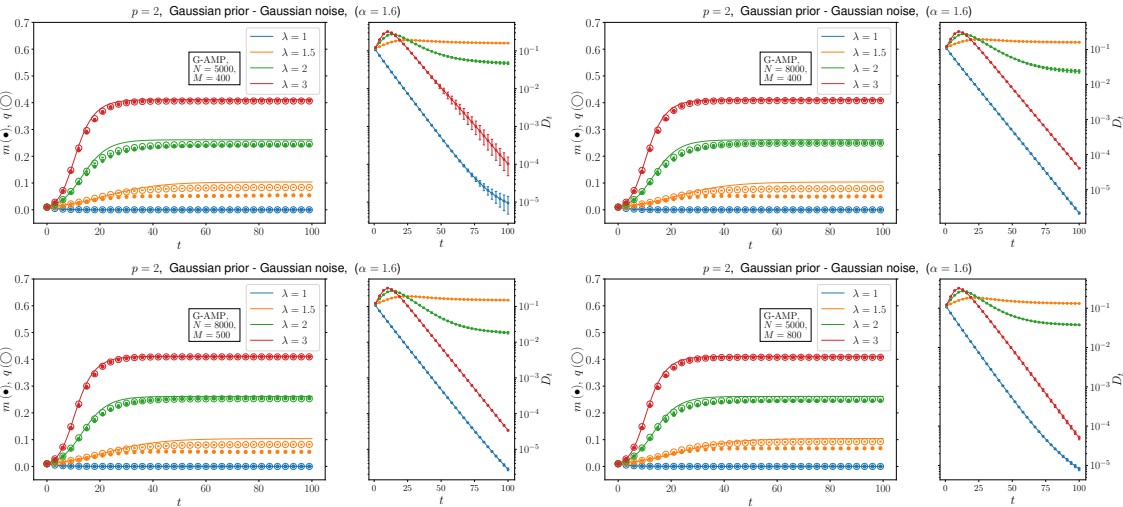

Figure 21: Finite $N, M$ corrections. Closer to the continuous transition, for $\lambda = 1.5$, convergence is the worst and deviations are present. Some slight improvement is observed when increasing $M$, in the last panel. Further away from the transition, as for $\lambda = 2$, also some finite $N$ effect becomes evident.

### E.2.2   $p = 3$

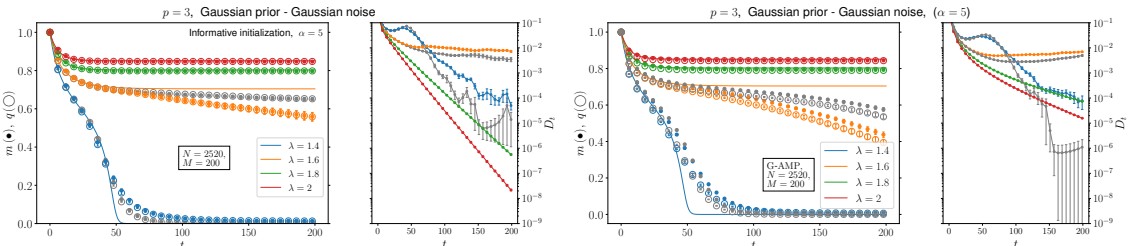

Figure 22: Comparison between r-BP algorithm (left panel) and G-AMP (right panel). Grey data points are obtained considering a bigger system size, $N = 5040$, while keeping fixed $M = 200$, for $\lambda = 1.4$ and 1.6. In both cases $\epsilon = 0.5$. Compare also to Fig. 11 in the main text for bigger values of $N, M$ in the G-AMP case.

## E.3   Gaussian prior - SignOutput

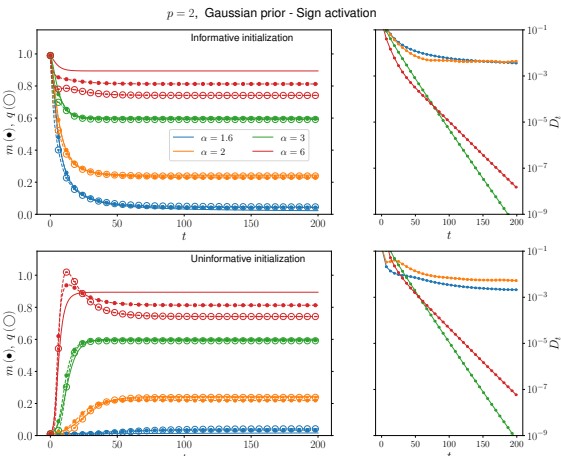

Figure 23: Results from the r-BP algorithm, $N = 2500$ and $M = 200$. For the largest value of $\alpha$ there are strong finite $M$ corrections. Solid lines represent the prediction from SE equations.

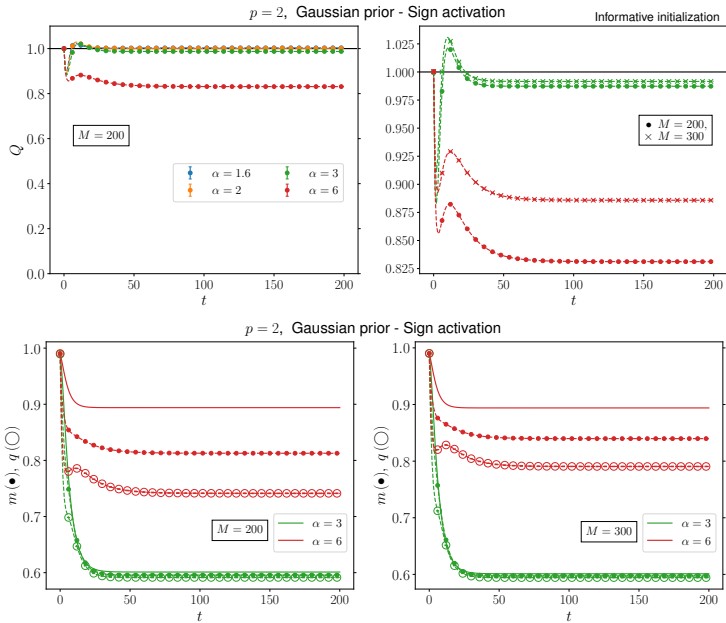

Figure 24: Results from the r-BP algorithm. Top panels: the finite $M$ corrections for $\alpha = 6$ in Fig. 23 are associated to a value of $Q_t$ consistently different from the Bayes Optimal expectation $Q_t = 1$. In the right panel, $Q_t$ for $\alpha = 3$ and $6$ is shown to improve when increasing $M$. Bottom panels: also the observables $q$ and $m$ show an improvement for $\alpha = 6$ when increasing $M$.

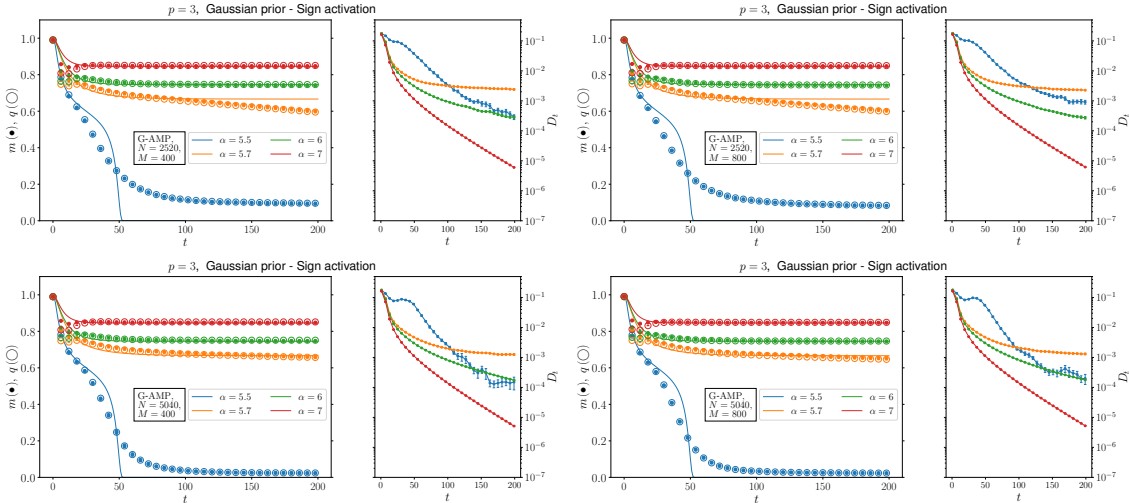

Figure 25: Finite $N, M$ corrections. Closer to the discontinuous transition, for $\alpha = 5.5$ and $\alpha = 5.7$, convergence is the worst and strong deviations due to finite size $N$ are present.

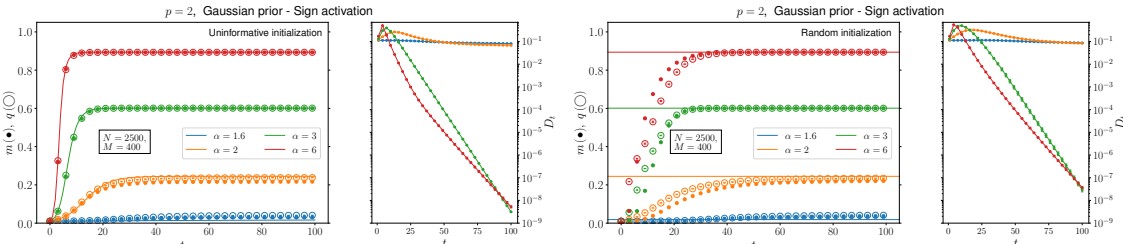

Figure 26: Comparison between the 'uninformative' (left) and 'truly random' (right) initializations. Right panel: the long time limit of the SE equations (coincident with the replica result) is shown as solid lines. The definition of the magnetization is corrected (see sec. D.1) in order to take into account the impossibility of recovering a global sign on each $\mu$-plane.

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
