# Peer review of "Graphical model for tensor factorization by sparse sampling"

_SciPost Physics_

## Round 1 · Referee Report · Anonymous (Referee 1) · 2025-11-25

Strengths

1- The authors study a nice and timely computational model with statistical physics tools, characterising the phase diagram of several sub-cases, finding first and second order phase transitions, and clarifying why their analysis breaks down if $M$ (tensor rank) is too big.

Weaknesses

1- While the presentation is clear and linear, maybe a presentation highlighting more the results would help the reader to parse the long and technical, but standard, computations. See comment below. 2- Lack of code. In this kind of papers, I find publishing code for full reproducibility a must, as the phase diagrams are all coming from numerical solutions of non-trivial equations.

Report

Summary

Model. The authors study a model of rank-M order-p tensor completion, where the observed tensor lives in $R^{N^p}$ and is the sum of M rank-1 spikes. Each spike may be multiplied by a known random observation factor F or not (F=1 case). Only a subset of the tensor is observed (hence the naming "completion" in my review), specifically only $\gamma \times (NM)$, where $NM$ is the total number of scalars to determine, under a uniformity assumption (namely that each of the $N$ tensor dimension is observed uniformly $\alpha M$ times, ensuring that the measurements are distributed uniformly enough). The authors work in the "dense limit" $1 \ll M \ll N$, where the rank of the tensor to retrieve is big, but much smaller than the ambient space dimension $N$.

Results. - The authors fully characterize the Bayes optimal estimation problem for generic priors and observation channels by providing a replica computation and the associated message passing algorithm. - The authors study in detail the phase diagram of the problem in specific settings, finding first and second order phase transitions reminiscent (but not equal) to what happens in rank-1 matrix factorization. - As a more technical remark, the authors argue by a cumulant expansion that some local fields that appear in the computation are Gaussian in the dense limit, and show that this would not be the case if $M = O(N^{p-1})$. They claim (app B, and discussion around eq. 73) that $c$ should be much smaller than any polynomial in $N$.

Comments

Presentation. The paper is quite clearly written, going sequentially first through the derivation and later on to the results. Nonetheless, I think the paper would benefit from a different presentation style. I find: - that the replica computation and message passing derivation are standard for a knowledgeable reader, while very technical for a less knowledgeable one. They could be a bit more "hidden", they do not provide much intuition. - The crucial non-trivial point of the replica computation is the dense limit and the associated cumulant expansion. This should be highlighted much more, and possibly discussed and referenced directly in the introduction when introducing the dense limit itself. - The phase diagrams are the actual main result, and they come only after a lot of technical computations. I do not have a precise suggestion here for the authors, I just wanted to make these points and let them assess whether and how to implement them.

Reproducibility. I think that in papers like this one, access to the code used both to solve the equations and to run AMP is a crucial part of the paper. I strongly encourage the authors to publish the code used both for solving the state equations (and find the subdominant branches / phase transition of the various problems) and for running AMP.

Completion vs factorization. The authors call repeatedly the model they study "tensor factorization", which is fine, but I would suggest the nomenclature "tensor completion". In usual studies of low-rank tensor factorization, the tensor is observed in full (contrary to here where the observation is partial and very sparse) while the spike strength is $O(N^{-1/2})$ (contrary to here where by the lack of observations one needs larger signal-to-noise ratio). This is just nomenclature, let me be clear on this, but I think it would help the reader to frame the results more easily. Additionally, I would invite the authors to: - explore the matrix completion literature and to compare their findings with known results there. A point of entry that I am aware of is [1], but I am sure this is not the only relevant work. - Compare their results in Section 5 with usual tensor factorization. For e.g., the phenomenology of Section 5.2.3 can be observed already in 2+3 rank-1 factorization. I think that heuristically, one could take the limit $\lambda \to 0$ and $\alpha \to \infty$ with appropriate scaling to access the usual tensor factorization regime (still for rank $1 \ll M \ll N$).

Cumulant expansion in BP. In Section 4.9 you say that the cumulant expansion found in replicas is not present. I have a strong feeling that it should be morally the same as the cumulant expansion in 4.2.1, and that while after 142 some terms are dropped as sub-leading, maybe such terms would actually re-sum to $O(1)$ contributions in the message passing scheme. But I do not have a precise argument, just a vibe.

Questions

  • A bit after eq 24: it is not clear to me why you say the non-numbered equation is a "generalisation error". Can you explain?

  • Eq 46: here you seem to be taking a saddle point over $N$ order parameters $Q_i$ as $M$ goes to infinity, but $M \ll N$. This is ill-posed. I think the computation is still correct under the ansatz eq 55. Could you comment on this?

  • Around Eq 73: here you write that $c \ll N$, but say in words and argue in App. B that $c$ should be much smaller than any polynomial in $N$. The notation $c \ll N$ suggests to me that $c = \sqrt{N}$ would be a good choice for $c$, but the argument in words would not allow for this. Can you clarify?

  • After Figure 6: "Meanwhile, the possible-to-impossible threshold is αs = 0 since the m= 1 solution always exists at finite" here this is not clear from Figure 6. In Figure 6 the solution m=1 does not appear to exist for lambda 0.5 and any value of alpha. Can you clarify?

  • Page 10: "and presumably any polynomial time algorithms" is missing citations to at the very minimum [2].

  • Section 5.2.1: is the transition here related to a BBP transition in the spectrum of the observed matrix?

  • In Section 3.6 bullet point 1, if I take p=3 and a=1, then the potential is effectively linear. Doe this mean that one has a reduction to low-rank $M$? If not, can you try to explain me what is going on here?

Minor points. - Eq 36: for the non-physicists reading the paper, I find it would be nice to explicitly tell that the identity can be proven by formally expanding the exponential in power series. - Eq 42: here and in surrounding equations pi depends on x and F. It could be stressed, even just in words, to clarify that the averages on x and F are not trivial. - Eq 51: is the differential for M (magnetization) missing? - Remark after Eq 122: this is very opaque for the reader not well-versed in error correcting codes. Maybe you could consider trying to better explain this? - Figure 3 left: maybe an inset zooming around m=1 would help to see that is going on there. - The discussion in Section 3.6 is difficult to follow without knowing very well [11], but seem to be important. It would be very nice if you could make an additional effort to clarify it further. - Figure 3,6,10 have the black lines undefined. I think they are subdominant branches, but that is not super clear, and the caption should probably mention this.

References

[1] A non-backtracking method for long matrix and tensor completion. L Stephan, Y Zhu. 2024 [2] The estimation error of general first order methods. Michael Celentano, Andrea Montanari, Yuchen Wu. 2020

Requested changes

I do not have specific requests for the authors. I just invite them to engage with the points I raised in the Report if they find it useful.

Recommendation

Publish (meets expectations and criteria for this Journal)

---

## Round 1 · Referee Report · Anonymous (Referee 2) · 2025-12-10

Strengths

See Report below

Weaknesses

Only minor points, see Requested changes below

Report

The paper considers the problem of low-rank matrix/tensor factorization under sparse measurements, when the number of tensor components observed sparsely at random by the Statistician is of the same order of the degrees of freedom of the problem. In this setting, a thorough statistical mechanics analysis is reported in the Bayes-optimal scenario, when the prior distributions and output channel generating the observations are known. On one hand, an equilibrium replica calculation is performed, giving access to the phase diagram of the problem for various choices of the prior distributions and output channels. On the other hand, message passing algorithms based on Belief Propagation and Approximate Message Passing are devised, as efficient methods to perform inference and alternative theoretical tools yielding results in agreement with the replica approach.

The setting and results are interesting both at the practical level, as making sparse measurements of a tensor and still being able to reconstruct it is a valuable observation, and at the theoretical one, as the sparsity of the problem allows to neglect intricate correlations among degrees of freedom, making the mean-field analysis much simpler and potentially exact. Moreover, the introduction is clear and previous literature properly referenced. For these reasons, I would like to recommend the paper for publications, once the following minor revisions will be addressed.

Requested changes

The following are all minor requests.

1- In the $p=2$, the inference strategy proposed by the paper (making sparse measurements, comparable in number to the degrees of freedom entering the low-rank signal) can be thought as alternative to the standard approach (measuring all the components, as in [10]). A comparison among the two strategies is only briefly mentioned at the end of section 2.3.2. It could be interesting for the reader to see a more explicit comparison, for example by showing plots of the MMSE from [10] side by side with the results of the paper. In this way, the trade off between measurement efficiency and accuracy would be more clear. 2- The use of the adjective “dense” to deem the limit considered by the authors is a bit confusing, in particular when first introduced in the main text, as it follows a sentence where the adjective “sparse” is used instead. I understand this use is partially supported by previous literature, to distinguish the finite-connectivity case to the sub-extensive connectivity limit the authors are considering. I suggest to make more explicit this fact by rephrasing a bit the sentence explaining the name. 3- The general ansatz (57), and all subsequent formulas reporting both $m$ and $q$, are not really needed in the Bayes-optimal setting, as the authors already point out (and actually show explicitly). Specializing formulas in the main text to the case $m=q$, and moving the more general ones and other details of the derivation to a dedicated appendix, could help readability and conciseness. 4- The control parameter $\Delta$ (variance of the additive Gaussian noise) is introduced in Eq. (15), but never made explicit in the figures/phase diagrams. It is only in Eq. (83) that the reader gets $\Delta=1$. This choice should be made more explicit. 5- What are the continuous black lines in Fig. 3 (left), Fig. 6 (left), Fig. 10 (left), Fig. 12 (left), Fig. 15? I assumed they were other branches for the order parameters, but the text and equations clearly show that the other solutions of the saddle point equations are unphysical, with $m,q>1$. 6- In the conclusions, the authors propose to test the setup with real-world data, possibly pushing the validity of the low-rank setting to its limits. Before doing that, I think it could be interesting to test the theoretical results they obtained, both from replicas and BP, with independent numerical methods not relying on the Gaussian assumption (74) (or on the simplifications taken in the BP approaches), such as Monte Carlo samplers. This should give indications on the size of the finite size corrections to the dense limit and to the relaxed BP already reported in the main. How hard could this be? Do the authors think that, by sampling at random $\alpha M N $ components of the fully-connected edge set, the constraints on the connectivity of each node would be approximately satisfied? And if this is not the case, how could one chose the edge set to be sparse and satisfy the constraints in practice? 7- A code should be provided to improve reproducibility of the results. 8- The mixed case is only briefly mentioned in section 5.2.3. Are there additional diagrams to control to make Eq. (74) exact, in this case? Such as cross correlations between the 2 species? 9- It is briefly mentioned after Eq. (20) that the randomness due to the specific instance of the graph chosen is irrelevant to the analysis. This is usually the case for random regular graphs. Could the authors add some references/elaborate a bit on this point? 10- Typos: - “Gaussian” should be capitalized everywhere (see Appendix C) - Eq. (66): the LHS should not have the indices $a,b$, and the ln det in Eq. (67) as well. The matrix $Q$ should be distinguished in some way from the $Q$ entering the RS ansatz (note that the latter $Q=1$ in the main text and could be simplified, see point 3- above). - Eq. (78), RHS: $Q\to Q^p$ - Few equations with wrong spacing, such as (105), (106), the ones on page 36...

Recommendation

Ask for minor revision

---

## Editorial Decision

awaiting_resubmission